# Finite-Time Analysis of Three-Timescale Constrained Actor-Critic and Constrained Natural Actor-Critic Algorithms

**Prashansa Panda**[1]                    **Shalabh Bhatnagar**[1]

[1]Department of Computer Science and Automation, Indian Institute of Science, Bangalore, India

## Abstract

Actor Critic methods have found immense applications on a wide range of Reinforcement Learning tasks especially when the state-action space is large. In this paper, we consider actor critic and natural actor critic algorithms with function approximation for constrained Markov decision processes (C-MDP) involving inequality constraints and carry out a non-asymptotic analysis for both of these algorithms in a non-i.i.d (Markovian) setting. We consider the long-run average cost criterion where both the objective and the constraint functions are suitable policy-dependent long-run averages of certain prescribed cost functions. We handle the inequality constraints using the Lagrange multiplier method. We prove that these algorithms are guaranteed to find a first-order stationary point (i.e., $\|\nabla L(\theta, \gamma)\|_2^2 \leq \epsilon$) of the performance (Lagrange) function $L(\theta, \gamma)$, with a sample complexity of $\tilde{\mathcal{O}}(\epsilon^{-2.5})$ in the case of both Constrained Actor Critic (C-AC) and Constrained Natural Actor Critic (C-NAC) algorithms. We also show the results of experiments on three different Safety-Gym environments.

## 1 INTRODUCTION

In recent times, there has been significant research activity on constrained reinforcement learning algorithms, motivated largely from applications in safe reinforcement learning (Safe-RL). Each state transition here not just receives a single-stage cost indicating the desirability of the action and the resulting next state but also receives additional constraint (single-stage) costs that may account for safety of the chosen action and the resulting next state. The goal then is to minimize a 'long-term cost' criterion while ensuring at the same time that the 'long-term constraint costs' stay within certain prescribed thresholds. The problem setting could generally involve more than one constraint cost. Problems of Safe-RL can in general be formulated in the setting of constrained Markov decision processes (C-MDP), see HasanzadeZonuzy et al. [2021], Jayant and Bhatnagar [2022], Wachi and Sui [2020]. Altman [2021] provides a textbook treatment on C-MDP. As an example, one may consider the problem of navigation of an autonomous vehicle such as a drone or a self-driving car where the goal is to reach the destination in as short a time as possible while ensuring there are no collisions with obstacles or accidents on the way. Such problems can be well formulated in the setting of C-MDPs. Constrained MDPs find several applications in various diverse domains. Some of these applications include finding optimal bandwidth allocation in resource constrained communication networks, determining strategic building and maintenance policies, safe navigation for self-driving cars, drones and robots, optimal energy sharing strategies in energy harvesting networks etc.

In situations where the system model is unavailable, but one has access to data in the form of tuples of states, actions, rewards, penalties, as well as next states, one may formulate and present constrained reinforcement learning (C-RL) algorithms for finding the optimal policies. An important algorithm in this category is the constrained actor critic (C-AC) algorithm originally presented by Borkar [2005] for the long-run average cost setting but for lookup table representations. In Bhatnagar [2010], Bhatnagar and Lakshmanan [2012], C-AC algorithms with function approximation have been presented and analysed for the infinite horizon discounted cost and the long-run average cost objectives respectively. In Bhatnagar and Lakshmanan [2012], an application of the presented algorithm is also studied empirically on a constrained multi-stage routing problem.

The key idea in the aforementioned algorithms has been to relax the constraints into the objective by forming a Lagrangian and then perform a gradient ascent step in the policy parameter while simultaneously performing a descent in

the Lagrange parameter. Note that usual actor-critic (Konda and Tsitsiklis [2003]) or natural actor critic algorithms (Bhatnagar et al., [2009]) ordinarily require two timescale recursions. This is because these algorithms try and mimic the policy iteration procedure whereby the actor or policy parameter update proceeds on the slower timescale while the critic or value function parameter is updated on the faster timescale. In constrained actor critic and constrained natural actor critic algorithms, one needs to introduce an additional (the slowest) timescale over which the Lagrange multiplier is updated.

In this paper, we carry out a finite-time (non-asymptotic) convergence analysis of three-timescale constrained actor-critic and constrained natural actor-critic algorithms to find the sample complexity of these algorithms (in the constrained setting). We assume that the system model via the transition probabilities is not known and linear function approximation is used for the critic recursion. A non-asymptotic analysis helps provide an estimate of the number of samples needed for the algorithm to converge as well as helps provide appropriate learning rates for the algorithm. In the case of the C-NAC algorithm, the natural gradient is estimated by linearly transforming the regular gradient by making use of the inverse Fisher information matrix of the policy which is clearly positive definite, see Kakade [2001]. It is generally observed that using natural gradients speeds up the performance of these algorithms. To the best of our knowledge, a non-asymptotic convergence analysis of three-timescale constrained actor-critic and constrained natural actor-critic algorithms using linear function approximation has not been carried out in the past.

We summarise our principal contributions below:
*(a)* We carry out the first finite-time analyses for the Constrained Actor-Critic and the Constrained Natural Actor-Critic algorithms with linear function approximation in the long-run average cost setting.
*(b)* We conduct the aforementioned analyses under the general assumption of Markovian sampling using TD(0) for the critic recursion and obtain a sample complexity of $\tilde{\mathcal{O}}(\epsilon^{-2.5})$ for both algorithms to find an $\epsilon$-optimal stationary point of the performance function.
*(c)* It is important to note here that the sample complexity of both our constrained algorithms matches exactly the one obtained by Wu et al., [2020] which is also $\tilde{\mathcal{O}}(\epsilon^{-2.5})$ even though the latter has been obtained for the case of two-timescale (unconstrained) regular actor-critic algorithms. Further, our setting is more general as we consider random single-stage costs and also constraint costs having distributions that are dependent on the current state, action and next state. This is unlike Wu et al., [2020] where the single-stage reward is assumed fixed for the given current state and action. Our result thus shows that under a random cost structure, with inequality constraints in the formulation, and having three-timescale algorithms as a result (instead

of two-timescale algorithms) has no impact on the sample complexity which we believe is a significant outcome of our study.
*(d)* We show the results of experiments on three different Safety-Gym environments, namely SafetyPointGoal1-v0, SafetyCarGoal1-v0 and SafetyPointPush1-v0, respectively, where we compare the performance of C-AC and C-NAC with Constrained DQN (C-DQN). For the latter, we incorporate the Lagrange based procedure and update the Lagrange multipliers in the same way as the C-AC and C-NAC procedures for a fair comparison. We observe that C-NAC shows the best results on all three settings while C-AC is the second-best performer on two of the three settings. Further, all algorithms satisfy the specified average cost constraint.

**Notation:** For two sequences $\{c_n\}$ and $\{d_n\}$, we can write $c_n = \mathcal{O}(d_n)$ if there exists a constant $P > 0$ such that $\frac{|c_n|}{|d_n|} \leq P, \forall n$. To further hide logarithm factors, we use the notation $\tilde{\mathcal{O}}(\cdot)$. Without any other specification, $\|\cdot\|$ denotes the $\ell_2$-norm of Euclidean vectors. Further, $d_{TV}(M, N)$ denotes the total variation norm between two probability measures $M$ and $N$, and is defined as $d_{TV}(M, N) = 1/2 \int_{\mathcal{X}} |M(dx) - N(dx)|$.

## 2 RELATED WORK

The Actor-Critic Algorithm was first analyzed theoretically for its asymptotic convergence by Konda and Borkar [1999]. This was however for the case of lookup table representations. In the case when function approximation is used, Konda and Tsitsiklis [2003] analyzed the actor critic algorithm for its asymptotic convergence. Kakade [2001] proposed the natural gradient based algorithm. Under various settings, the asymptotic convergence of actor critic algorithms has also been studied in (Kakade [2001], Castro and Meir [2010], Zhang et al., [2020]). In Bhatnagar et al., [2009], natural actor critic algorithms that bootstrap in both the actor and the critic have been proposed and analyzed for their asymptotic convergence.

In recent times, there have been many works focusing primarily on performing finite time analyses of reinforcement learning algorithms. Such analyses are important as they provide sample complexity estimates and non-asymptotic convergence bounds for these algorithms. More recently, such analyses for actor critic algorithms have also been carried out though in the unconstrained (regular MDP) setting. Ding et al. [2020] obtain finite time bounds for a natural policy gradient algorithm for discounted cost MDP with constraints. Wu et al., [2020] show a non-asymptotic analysis of a two time-scale actor-critic algorithm assuming non-i.i.d samples and obtain a sample complexity of $\tilde{\mathcal{O}}(\epsilon^{-2.5})$ for convergence to an $\epsilon$-approximate stationary point of the performance function. Hairi et al., [2021] consider a fully decentralized multi-agent reinforcement learning (MARL)

Table 1: Comparison of Finite-Time Analysis of Various Actor-Critic Algorithms.

| Reference | Algorithm | Sample Complexity | Function Approximation | Timescales | Critic estimation |
|---|---|---|---|---|---|
| Wu et al., [2020] | Actor-Critic | $\widetilde{\mathcal{O}}(\epsilon^{-2.5})$ | ✓ | two-timescale | TD(0) |
| Chen and Zhao [2022] | Actor-Critic | $\widetilde{\mathcal{O}}(\epsilon^{-2})$ | ✓ | single timescale | TD(0) |
| Zeng et al., [2022] | Constrained NAC | $\mathcal{O}(\epsilon^{-6})$ | ✗ | two time-scale | TD(0) |
| Suttle et al,. [2023] | Actor-Critic | $\widetilde{\mathcal{O}}(\tau_{mix}^2\epsilon^{-2})$ | ✓ | two-timescale | Monte-Carlo |
| Our work | Constrained AC | $\widetilde{\mathcal{O}}(\epsilon^{-2.5})$ | ✓ | three-timescale | TD(0) |
| Our work | Constrained NAC | $\widetilde{\mathcal{O}}(\epsilon^{-2.5})$ | ✓ | three-timescale | TD(0) |

setting and show a finite-time convergence analysis for the actor-critic algorithm in the average reward MDP scenario. Chen and Zhao [2022] carried out finite time analysis of single time-scale actor critic algorithm and obtained a sample complexity of $\tilde{\mathcal{O}}(\epsilon^{-2})$ for convergence to an $\epsilon$-approximate stationary point of the performance function. Suttle et al,. [2023] studied the non-asymptotic convergence properties of Multi-level Monte Carlo Actor-Critic (MAC) algorithm. Mondal and Aggarwal [2024] proposed and studied the convergence properties of Accelerated Natural Policy Gradient (ANPG) algorithm. There have also been other recent works that have analysed Natural Actor-Critic Algorithms for their non-asymptotic convergence, see for instance, Cayci et al., [2022], Xu et al., [2020], Khodadadian et al., [2023], Khodadadian et al., [2021], Chen et al., [2022]. Table 1 summarises a comparison of our work with a few related works in terms of sample complexity.

In early work, Borkar [2005] proposed the first actor-critic algorithm for constrained Markov decision processes in the long-run average cost setting and proved its asymptotic convergence in the lookup table setting. Bhatnagar [2010] presented the first actor-critic algorithm with function approximation for the infinite horizon discounted cost problem under multiple inequality constraints and proved its asymptotic convergence. Bhatnagar and Lakshmanan [2012] presented an actor-critic algorithm in the constrained long-run average cost MDP setting with function approximation under policy gradient actor and temporal difference critic and also analysed its asymptotic convergence.

Zeng et al., [2022] recently showed a finite-time analysis for the non-asymptotic convergence of the natural actor-critic algorithm to the global optimum of a CMDP problem in the case of lookup table representations and the infinite horizon discounted cost setting. In this paper, we consider the long-run average cost problem with function approximation that has been analysed for its asymptotic convergence in (Bhatnagar and Lakshmanan [2012]). We use TD(0) for the critic recursion and use projection for the critic. Further, we present the C-NAC algorithm, where we use natural gradients in the actor recursion along with a TD(0) critic. As mentioned earlier, there are no prior finite time analyses for

average cost/reward constrained actor critic algorithms with function approximation, so our work plugs in an important gap that previously existed in this direction.

# 3 PRELIMINARIES

In this section, we present the C-MDP framework and algorithms that we analyze.

## 3.1 CONSTRAINED MARKOV DECISION PROCESSES

We consider a discrete-time Markov Decision Process with finite state and action spaces. We first explain below the notations used.

• $S$ represents the state space and $A$ the action space. Further, we let $A(j) \subset A$ denote the set of feasible actions in state $j \in S$.

• Let $p(s, s', a)$ denote the probability of transition from state $s$ to $s'$ under action $a$.

• We shall consider only randomized policies $\pi$ in this work. Further, policies are assumed parameterized via a parameter $\theta \in \mathcal{R}^d$. Thus, given $\theta$, $\pi_\theta(a|s)$ is the probability of selecting an action $a \in A(s)$ in state $s$.

• The stationary distribution (over states) induced by the policy $\pi_\theta$ is denoted $\mu_{\pi_\theta}$ or simply $\mu_\theta$ (by an abuse of notation) and is assumed unique for any $\theta$.

Let $q(n), h_1(n), ..., h_N(n), n \geq 0$, denote a set of costs obtained upon transitioning from state $s_n$ to state $s_{n+1}$ under action $a_n \in A(s_n)$. At any time instant $n$, the single-stage costs $q(n), h_k(n), k = 1, ..., N$, do not depend on prior states and actions $s_m, a_m, m < n$ given the current state-action pair $(s_n, a_n)$. For any $i \in S$, $a \in A(i)$, let $d(i, a), h_k(i, a)$ be defined as $d(i, a) = E[q(n)|s_n = i, a_n = a]$, $h_k(i, a) = E[h_k(n)|s_n = i, a_n = a], k = 1, ..., N$, respectively. Note the abuse of notation here. We assume that the single-stage costs are real-valued, non-negative and mutually independent. Further, we assume

that all the single-stage costs $q(n), h_1(n), \ldots, h_N(n)$ are absolutely bounded by a constant $U_c > 0$.

## 3.2 THE OBJECTIVE AND LAGRANGE RELAXATION

Our aim here is to minimize $J(\pi)$ where

$$
\begin{aligned}
J(\pi) &= \lim_{n \to \infty} \frac{1}{n} \mathbb{E}\Big[ \sum_{m=0}^{n-1} q(m) | \pi \Big] \\
&= \sum_{s \in S} \mu_\pi(s) \sum_{a \in A(s)} \pi(s, a) d(s, a),
\end{aligned} \tag{1}
$$

subject to the constraints

$$
\begin{aligned}
G_k(\pi) &= \lim_{n \to \infty} \frac{1}{n} \mathbb{E}\Big[ \sum_{m=0}^{n-1} h_k(m) | \pi \Big] \\
&= \sum_{s \in S} \mu_\pi(s) \sum_{a \in A(s)} \pi(s, a) h_k(s, a) \le \alpha_k,
\end{aligned} \tag{2}
$$

$k = 1, \ldots, N$, where $\alpha_1, \ldots, \alpha_N$ are certain prescribed (positive) constant thresholds. We consider here that the Markov process $\{s_n\}$ under any given policy is ergodic. Hence, the limits in (1)-(2) are well-defined.

Consider a vector $\gamma = (\gamma_1, \ldots, \gamma_N)^T$ representing a set of Lagrange multipliers with $\gamma_1, \ldots, \gamma_N \in R^+ \cup \{0\}$. We define the Lagrangian $L(\pi, \gamma)$ according to

$$
\begin{aligned}
&L(\pi, \gamma) \\
&= J(\pi) + \sum_{k=1}^{N} \gamma_k (G_k(\pi) - \alpha_k) \\
&= \sum_{s \in S} \mu_\pi(s) \sum_{a \in A(s)} \pi(s, a) \Big( d(s, a) + \sum_{k=1}^{N} \gamma_k (h_k(s, a) - \alpha_k) \Big).
\end{aligned}
$$

We now have the unconstrained MDP problem with single-stage cost being $q(t) + \sum_{k=1}^{N} \gamma_k (h_k(t) - \alpha_k)$ at instant $t$. The differential action value function in the relaxed control setting is defined as follows:

$$
\begin{aligned}
&M^{\pi, \gamma}(s, a) \\
&= \sum_{t=1}^{\infty} \mathbb{E}\Big[ q(t) + \sum_{i=1}^{N} \gamma_i (h_i(t) - \alpha_i) \\
&\quad - \Big( J(\boldsymbol{\theta}) + \sum_{i=1}^{N} \gamma_i (G_i(\boldsymbol{\theta}) - \alpha_i) \Big) | s_0 = s, a_0 = a, \pi \Big] \\
&= \sum_{t=1}^{\infty} \mathbb{E}\Big[ q(t) + \sum_{i=1}^{N} \gamma_i h_i(t) \\
&\quad - \Big( J(\boldsymbol{\theta}) + \sum_{i=1}^{N} \gamma_i G_i(\boldsymbol{\theta}) \Big) | s_0 = s, a_0 = a, \pi \Big].
\end{aligned}
$$

As mentioned by Bhatnagar and Lakshmanan [2012], in the constraint scenario, the policy gradient of the Lagrangian would correspond to

$$
\nabla_{\boldsymbol{\theta}} L(\boldsymbol{\theta}, \gamma) = \sum_{s \in S} \mu_\pi(s) \sum_{a \in A(s)} \nabla \pi(a|s) A^{\pi, \gamma}(s, a), \tag{3}
$$

where $A^{\pi, \gamma}(s, a) = M^{\pi, \gamma}(s, a) - V^{\pi, \gamma}(s)$ is the advantage function for the relaxed setting. Here $V^{\pi, \gamma}(s)$ is the differential cost for a given policy $\pi$ and a set of Lagrange parameters $\gamma$. We use linear function approximation for $M^{\pi, \gamma}(s, a)$. Thus, the same is approximated as follows:

$$
\hat{M}_w^{\pi, \gamma}(s, a) \approx w^{\pi, \gamma^T} \Psi_{sa},
$$

where $w^{\pi, \gamma} \in \mathbb{R}^d$ are suitable parameters and $\Psi_{sa} \in \mathbb{R}^d$ are compatible features for the tuples $(s, a)$. Thus, $\Psi_{sa} = \nabla \log \pi(a|s), \forall s \in S, a \in A(s)$.

We also use linear function approximation for the differential value function $V^{\pi, \gamma}(s)$ as follows: We let

$$
\hat{V}_v^{\pi, \gamma}(s) \approx v^{\pi, \gamma^T} f_s,
$$

where $f_s$ is a $d_1$-dimensional feature vector $f_s = (f_s(1), f_s(2), \ldots, f_s(d_1))^T$ associated with state $s$ and $v^{\pi, \gamma} = (v^{\pi, \gamma}(1), v^{\pi, \gamma}(2), \ldots, v^{\pi, \gamma}(d_1))^T$ is the corresponding weight vector.

## 3.3 THE CONSTRAINED ACTOR-CRITIC (C-AC) AND CONSTRAINED NATURAL ACTOR-CRITIC (C-NAC) ALGORITHMS

---

**Algorithm 1** The Three-Timescale Actor-Critic Algorithm for Constrained MDP

---

1: **Input** $\theta_0, v_0, L_0, U_k(0)$ for $1 \le k \le N$, $\gamma_k(0)$ for $1 \le k \le N$, step-size $a(n)$ for critic and average cost estimate, $b(n)$ for actor and $c(n)$ for Lagrange parameter.
2: Draw $s_0$ from some initial distribution
3: **for** $n > 0$ and $k = 1, 2, \ldots, N$ **do**
4:     Sample $a_n \sim \pi_{\theta_n}(\cdot|s_n)$, $s_{n+1} \sim p(s_n, \cdot, a_n)$
5:     Observe the costs $q(n), h_1(n), h_2(n), \ldots, h_N(n)$
6:     $L_{n+1} = L_n + a(n)(q(n) + \sum_{k=1}^{N} \gamma_k(n)(h_k(n) - \alpha_k) - L_n)$
7:     $\delta_n = q(n) + \sum_{k=1}^{N} \gamma_k(n)(h_k(n) - \alpha_k) - L_n + v_n^T(f_{s_{n+1}} - f_{s_n})$
8:     $v_{n+1} = \Gamma(v_n + a(n)\delta_n f_{s_n})$
9:     $\theta_{n+1} = \theta_n + b(n)\delta_n \Psi_{s_n a_n}$
10:     $U_k(n+1) = U_k(n) + a(n)(h_k(n) - U_k(n))$
11:     $\gamma_k(n+1) = \hat{\Gamma}(\gamma_k(n) + c(n)(U_k(n) - \alpha_k))$
12: **end for**

---

We present here the two algorithms that we analyze in our work for their non-asymptotic convergence – the constrained

actor-critic algorithm (Algorithm 1) and the constrained natural actor-critic algorithm (Algorithm 2), respectively. At time instant $t$, we have $\theta_t$ as the actor parameter, $v_t$ as the critic parameter, $L_t$ as the average cost estimate, $U_k(t)$ as the average constraint cost estimate for $k = 1, 2, \ldots, N$, $\gamma(t) = (\gamma_1(t), \gamma_2(t), \ldots, \gamma_N(t))^T$ as the vector of Lagrange multiplier estimates and $G(t)$ as the estimate of the Fisher information matrix respectively.

Let $\Gamma : \mathbb{R}^{d_1} \to C$ project any point in $\mathbb{R}^{d_1}$ to the closest point within the set $C$ which is assumed compact and convex. For any point $h$ contained in the set $C$, $\|h\| \leq U_v$ where $U_v > 0$ is a constant. Further, $\hat{\Gamma} : \mathbb{R} \to [0, M]$ indicates the operation $\hat{\Gamma}(y) = \max(0, \min(y, M))$ for any $y \in \mathbb{R}$ where $M < \infty$ represents a large positive constant. This projection operator guarantees the Lagrange multiplier to stay both non-negative and bounded.

For the natural actor-critic algorithm, we take $G(0) = pI$, where $I$ is a $d \times d$-identity matrix and $p > 0$ is a constant. It can be concluded that $G(n), n \geq 1$ are positive definite and symmetric matrices as from the update rule, we can see that these result from addition of $(1 - a(n))G(n-1)$ and $a(n)\Psi_{s_n a_n}\Psi_{s_n a_n}^T$. Hence, $G(n)^{-1}, n \geq 1$ are positive definite and symmetric matrices as well. Let the smallest eigenvalue of $G(i)^{-1}$ be $\lambda_i > 0$, where $i \geq 1$. Let $\lambda$ be the minimum of all such eigenvalues, i.e., $\lambda = \min_i \lambda_i > 0$.

---

**Algorithm 2** The Three-Timescale Natural Actor-Critic Algorithm for Constrained MDP

---

1: **Input** $\theta_0$, $v_0$, $L_0$, $U_k(0)$ for $1 \leq k \leq N$, $\gamma_k(0)$ for $1 \leq k \leq N$, $G(0)$, step-size $a(n)$ for critic and average cost estimate, $b(n)$ for actor and $c(n)$ for Lagrange parameter.
2: Draw $s_0$ from some initial distribution
3: **for** $n > 0$ and $k = 1, 2, \ldots, N$ **do**
4:     Sample $a_n \sim \pi_{\theta_n}(\cdot|s_n)$, $s_{n+1} \sim p(s_n, \cdot, a_n)$
5:     Observe the costs $q(n), h_1(n), h_2(n), \ldots, h_N(n)$
6:     $L_{n+1} = L_n + a(n)(q(n) + \sum_{k=1}^{N} \gamma_k(n)(h_k(n) - \alpha_k) - L_n)$
7:     $\delta_n = q(n) + \sum_{k=1}^{N} \gamma_k(n)(h_k(n) - \alpha_k) - L_n + v_n^T(f_{s_{n+1}} - f_{s_n})$
8:     $v_{n+1} = \Gamma(v_n + a(n)\delta_n f_{s_n})$
9:     $\theta_{n+1} = \theta_n + b(n)\delta_n G(n)^{-1}\Psi_{s_n a_n}$
10:     $U_k(n+1) = U_k(n) + a(n)(h_k(n) - U_k(n))$
11:     $\gamma_k(n+1) = \hat{\Gamma}(\gamma_k(n) + c(n)(U_k(n) - \alpha_k))$
12:     $G(n+1) = (1 - a(n))G(n) + a(n)\Psi_{s_n a_n}\Psi_{s_n a_n}^T$
13: **end for**

---

# 4 FINITE-TIME CONVERGENCE RESULTS

We provide in this section the main theoretical results for non-asymptotic convergence as well as provide the convergence rate and sample complexity for the two algorithms.

Due to lack of space, we provide the detailed proofs of these results in the appendix. We emphasize here that asymptotic convergence analysis of these algorithms has not been analysed here since for the case of Constrained Actor-Critic, it has been analysed in Bhatnagar and Lakshmanan [2012]. Further, the same for Constrained Natural Actor-Critic, it will carry through in a similar manner using the results of Bhatnagar et al., [2009].

## 4.1 ASSUMPTIONS AND BASIC RESULTS

We consider TD(0) with function approximation for the critic recursion that estimates the state-value function. Let $v^*(\theta, \gamma)$ be the convergence point of the critic under the behavior policy $\pi_{\theta}$ (for given actor and Lagrange parameters $\theta$ and $\gamma$ respectively), and define $\mathbf{A}$ and $\mathbf{b}$ as follows:

$$\mathbf{A} := \mathbb{E}_{s_n, a_n, s_{n+1}}\left[f_{s_n}\left(f_{s_{n+1}} - f_{s_n}\right)^\top\right],$$
$$\mathbf{b} := \mathbb{E}_{s_n, a_n, s_{n+1}}\left[(C(s_n, a_n, \gamma) - L(\boldsymbol{\theta}, \gamma))f_{s_n}\right],$$

where $s_n \sim \mu_{\boldsymbol{\theta}}(\cdot), a_n \sim \pi_{\boldsymbol{\theta}}(\cdot|s), s_{n+1} \sim p(s_n, \cdot, a_n)$ and $C(s_n, a_n, \gamma) = d(s_n, a_n) + \sum_{k=1}^{N} \gamma_k(h_k(s_n, a_n) - \alpha_k)$ corresponds to the single-stage cost for the relaxed problem. Analogous to the unconstrained setting, it can be seen that (see Bhatnagar and Lakshmanan [2012])

$$\mathbf{A}v^*(\theta, \gamma) + \mathbf{b} = \mathbf{0}.$$

**Assumption 1** *The norm of each state feature is bounded by 1, i.e., $\|f_i\| \leq 1$.*

The following assumption is required for the existence and uniqueness of $v^*(\theta, \gamma)$ .

**Assumption 2** *The matrix $\mathbf{A}$ (defined above) is negative definite with maximum eigenvalue as $-\lambda_e < 0$ for all values of $\boldsymbol{\theta}$.*

The approximation error for the feature mapping can vary depending on its complexity. We define the approximation error that arises due to linear function approximation as follows.

$$\epsilon_{\text{app}}(\boldsymbol{\theta}, \boldsymbol{\gamma}) := \sqrt{\mathbb{E}_{s \sim \mu_{\boldsymbol{\theta}}}\left(f_s^\top v^*(\boldsymbol{\theta}, \boldsymbol{\gamma}) - V^{\pi_{\boldsymbol{\theta}}, \boldsymbol{\gamma}}(s)\right)^2}.$$

**Assumption 3**

$$\forall\boldsymbol{\theta}, \forall\boldsymbol{\gamma}, \ \epsilon_{app}(\boldsymbol{\theta}, \boldsymbol{\gamma}) \leq \epsilon_{app},$$

*where $\epsilon_{app} \geq 0$ is some constant.*

Assumption 3 is useful in finding upper bounds of some of the error terms.

**Assumption 4 (Uniform ergodicity)** *For a given $\boldsymbol{\theta}$, we consider the policy $\pi_{\boldsymbol{\theta}}(\cdot|s)$ and the transition probability measure $p(s,\cdot,a)$ that induce a stationary distribution $\mu_{\boldsymbol{\theta}}(\cdot)$. There exists $b > 0$ and $k \in (0,1)$ for the Markov chain where $a_t \sim \pi_{\boldsymbol{\theta}}(\cdot|s_t), s_{t+1} \sim p(s_t,\cdot,a_t)$ such that*

$$d_{TV}\big(p^\tau(x,y,\cdot),\mu_{\boldsymbol{\theta}}(y)\big) \leq bk^\tau, \forall \tau \geq 0, \forall x,y \in \mathcal{S}.$$

Assumption 4 is needed to tackle the issue of Markov sampling in TD learning. It has been used in analyses of TD learning, for instance, in Bhandari et al., [2018]. Refer to Meyn and Tweedie [2009] for various results related to uniform ergodicity as well as other notions of ergodicity of Markov chains.

**Assumption 5** *There exist constants $L, D, M_m$ such that $\forall \boldsymbol{\theta}_1, \boldsymbol{\theta}_2, \boldsymbol{\theta} \in \mathbb{R}^d$, we have*

(a) $\big\|\nabla \log \pi_{\boldsymbol{\theta}}(a|i)\big\| \leq D, \forall i, \forall a,$

(b) $\big\|\nabla \log \pi_{\boldsymbol{\theta}_1}(a|i) - \nabla \log \pi_{\boldsymbol{\theta}_2}(a|i)\big\| \leq M_m\|\boldsymbol{\theta}_1 - \boldsymbol{\theta}_2\|,$ $\forall i, \forall a,$

(c) *There exist scalars $\check{K}, \hat{K} > 0$ such that for any $x \neq 0$ and all $s_n, a_n$,*

$$\check{K}\|x\|^2 \leq x^T \Psi_{s_n a_n} \Psi_{s_n a_n}^T x \leq \hat{K}\|x\|^2.$$

**Remark 1** *As a consequence of Assumption 5(a), it follows that $\big|\pi_{\boldsymbol{\theta}_1}(a|i) - \pi_{\boldsymbol{\theta}_2}(a|i)\big| \leq L\|\boldsymbol{\theta}_1 - \boldsymbol{\theta}_2\|, \forall i, \forall a.$ In other words, the policy for any given $(i,a)$ tuple is Lipschitz continuous in $\theta$.*

Assumption 5 provides smoothness of the parameterized policies and can be seen to be verified by many policies. This assumption is useful for finding upper bounds for some of the error terms while proving the convergence of actor and critic recursions.

**Proposition 1** *The updates $G(t)$ satisfy $\sup_t \|G(t)\| < \infty$ and $\sup_t \|G(t)^{-1}\| < \infty$, respectively.*

*Proof:* Since $a(n) \to 0$ as $n \to \infty, \exists N_0 \geq 1$ such that for all $n \geq N_0$, $G(n+1)$ is a convex combination of $G(n)$ and $\Psi_{s_n a_n} \Psi_{s_n a_n}^T$, with $G_0 = pI, p > 0$, see step 12 of Algorithm 2. Without loss of generality, assume that $a(n) \leq 1, \forall n$. Thus, observe that for $n = 0$,

$$(1 - a(0))p\|x\|^2 + a(0)\check{K}\|x\|^2 \leq x^T G(1)x^T$$
$$\leq (1 - a(0))p\|x\|^2 + a(0)\hat{K}\|x\|^2.$$

Letting $\check{M} = \min(p, \check{K})$ and $\hat{M} = \max(p, \hat{K})$, it can be verified from induction that

$$\check{M}\|x\|^2 \leq x^T G(n)x \leq \hat{M}\|x\|^2,$$

uniformly over $n \geq 0$. The claim now follows from arguments on page 35 of Bertsekas [1999].

**Proposition 2** *There exists a constant $L_1 > 0$ such that $\forall \boldsymbol{\gamma} \in \mathbb{R}^N$ with $\boldsymbol{\gamma} = (\boldsymbol{\gamma}_1, \ldots, \boldsymbol{\gamma}_N)^T$ and $0 \leq \boldsymbol{\gamma}_j \leq M$ for $j = 1, 2, \ldots, N$,*

$$\|v^*(\boldsymbol{\theta}_1, \boldsymbol{\gamma}) - v^*(\boldsymbol{\theta}_2, \boldsymbol{\gamma})\| \leq L_1\|\boldsymbol{\theta}_1 - \boldsymbol{\theta}_2\|, \forall \boldsymbol{\theta}_1, \boldsymbol{\theta}_2 \in \mathbb{R}^d.$$

*Proof Sketch*

Let

$$\mathbf{A}_{\boldsymbol{\theta}} := \mathbb{E}_{s_n, a_n, s_{n+1}}\big[f_{s_n}\big(f_{s_{n+1}} - f_{s_n}\big)^\top\big],$$
$$\mathbf{b}_{\boldsymbol{\theta}, \boldsymbol{\gamma}} := \mathbb{E}_{s_n, a_n, s_{n+1}}[(C(s_n, a_n, \gamma) - L(\boldsymbol{\theta}, \gamma))f_{s_n}],$$

where $s_n \sim \mu_{\boldsymbol{\theta}}(\cdot), a_n \sim \pi_{\boldsymbol{\theta}}(\cdot|s), s_{n+1} \sim p(s_n, \cdot, a_n)$.

We have,

$$\mathbf{A}_{\theta}v^*(\theta, \gamma) + \mathbf{b}_{\theta, \gamma} = \mathbf{0}.$$

Thus,

$$\|v^*(\theta_1, \boldsymbol{\gamma}) - v^*(\theta_2, \boldsymbol{\gamma})\|$$
$$= \|\mathbf{A}_{\theta_1}^{-1}\mathbf{b}_{\theta_1, \gamma} - \mathbf{A}_{\theta_2}^{-1}\mathbf{b}_{\theta_2, \gamma}\|$$
$$\leq \|\mathbf{A}_{\theta_1}^{-1}\mathbf{b}_{\theta_1, \gamma} - \mathbf{A}_{\theta_2}^{-1}\mathbf{b}_{\theta_1, \gamma}\| + \|\mathbf{A}_{\theta_2}^{-1}\mathbf{b}_{\theta_1, \gamma} - \mathbf{A}_{\theta_2}^{-1}\mathbf{b}_{\theta_2, \gamma}\|$$
$$\leq \|\mathbf{A}_{\theta_1}^{-1} - \mathbf{A}_{\theta_2}^{-1}\|\|\mathbf{b}_{\theta_1, \gamma}\| + \|\mathbf{A}_{\theta_2}^{-1}\|\|\mathbf{b}_{\theta_1, \gamma} - \mathbf{b}_{\theta_2, \gamma}\|.$$

It can be shown that

$$\|\mathbf{b}_{\theta_1, \gamma}\| \leq 2U_r,$$
$$\|\mathbf{A}_{\theta_2}^{-1}\| \leq \lambda_e^{-1},$$
$$\mathbf{A}_{\theta_1}^{-1} - \mathbf{A}_{\theta_2}^{-1} = \mathbf{A}_{\theta_1}^{-1}(\mathbf{A}_{\theta_2} - \mathbf{A}_{\theta_1})\mathbf{A}_{\theta_2}^{-1}.$$

We have from section B.2 of Wu et al., [2020] the following:

$$\|\mathbf{A}_{\theta_1} - \mathbf{A}_{\theta_2}\| \leq 4|A|L\left(1 + \lceil \log_k b^{-1}\rceil + \frac{1}{1-k}\right)\|\boldsymbol{\theta}_1 - \boldsymbol{\theta}_2\|,$$

$$\|\mathbf{b}_{\theta_1, \gamma} - \mathbf{b}_{\theta_2, \gamma}\| \leq 6|A|U_rL\left(1 + \lceil \log_k b^{-1}\rceil + \frac{1}{1-k}\right)\|\boldsymbol{\theta}_1 - \boldsymbol{\theta}_2\|.$$

After combining all the terms, we have

$$\big\|v^*(\boldsymbol{\theta}_1, \boldsymbol{\gamma}) - v^*(\boldsymbol{\theta}_2, \boldsymbol{\gamma})\big\| \leq L_1\|\boldsymbol{\theta}_1 - \boldsymbol{\theta}_2\|, \forall \boldsymbol{\theta}_1, \boldsymbol{\theta}_2 \in \mathbb{R}^d,$$

where $L_1 = (8\lambda_e^{-2} + 6\lambda_e^{-1})U_r|A|L\left(1 + \lceil \log_k b^{-1}\rceil + \frac{1}{1-k}\right)$.

**Proposition 3** *Let $\boldsymbol{\gamma}^1 = (\gamma_1^1, \ldots, \gamma_N^1)^T$ and $\boldsymbol{\gamma}^2 = (\gamma_1^2, \ldots, \gamma_N^2)^T$ be any two vectors in $\mathbb{R}^N$ with $0 \leq \gamma_j^i \leq M$ for $i = 1, 2$ and $j = 1, 2, \ldots, N$. There exists a constant $L_2 > 0$ such that*

$$\big\|v^*(\boldsymbol{\theta}, \boldsymbol{\gamma}^1) - v^*(\boldsymbol{\theta}, \boldsymbol{\gamma}^2)\big\| \leq L_2|\gamma_p^1 - \gamma_p^2|, \forall \boldsymbol{\theta} \in \mathbb{R}^d,$$

*where $|\gamma_p^1 - \gamma_p^2| = \max_{i=1,2,\ldots,N} |\gamma_i^1 - \gamma_i^2|$.*

*Proof Sketch*

We have,

$$\left\|v^*(\theta,\boldsymbol{\gamma}^1) - v^*(\theta,\boldsymbol{\gamma}^2)\right\| = \|\mathbf{A}_\theta^{-1}\mathbf{b}_{\theta,\gamma^1} - \mathbf{A}_\theta^{-1}\mathbf{b}_{\theta,\gamma^2}\|$$
$$\leq \|\mathbf{A}_\theta^{-1}\| \underbrace{\|\mathbf{b}_{\theta,\gamma^1} - \mathbf{b}_{\theta,\gamma^2}\|}_{I_1}.$$

Now for the term $I_1$, note that

$$\|\mathbf{b}_{\theta,\gamma^1} - \mathbf{b}_{\theta,\gamma^2}\|$$
$$= \|E_{\{s_n\sim\mu_\theta(\cdot),a_n\sim\pi_\theta(\cdot|s),s_{n+1}\sim p(s_n,\cdot,a_n)\}}[(C(s_n,a_n,\boldsymbol{\gamma}^1)$$
$$- C(s_n,a_n,\boldsymbol{\gamma}^2) + L(\boldsymbol{\theta},\boldsymbol{\gamma}^2) - L(\boldsymbol{\theta},\boldsymbol{\gamma}^1))f_{s_n}]\|$$
$$\leq 2N(U_c + U_\alpha)|\boldsymbol{\gamma}_p^1 - \boldsymbol{\gamma}_p^2|,$$

where, $|\boldsymbol{\gamma}_p^1 - \boldsymbol{\gamma}_p^2| = \max\limits_{i=1,2,\ldots,N}|\gamma_i^1 - \gamma_i^2|$. Hence,

$$\left\|v^*(\boldsymbol{\theta},\boldsymbol{\gamma}^1) - v^*(\boldsymbol{\theta},\boldsymbol{\gamma}^2)\right\| \leq L_2|\boldsymbol{\gamma}_p^1 - \boldsymbol{\gamma}_p^2|, \forall\boldsymbol{\gamma}_1,\boldsymbol{\gamma}_2 \in \mathbb{R}^N,$$

where, $L_2 = (2N(U_c + U_\alpha))/\lambda_e$.

Let $\tau_t$ denote the mixing time of our ergodic Markov chain. So we have

$$\tau_t := \min\left\{m \geq 0 | bk^{m-1} \leq \min\{a(t),b(t),c(t)\}\right\}, \quad (4)$$

where $b, k$ are defined as in Assumption 4.

We now present the result of non-asymptotic analysis of constrained actor-critic methods. We consider $a(t) = c_a(1+t)^{-\omega}$, $b(t) = c_b(1+t)^{-\sigma}$ and $c(t) = c_c(1+t)^{-\beta}$, where $0 < \omega < \sigma < \beta \leq 1$, with $c_a$, $c_b$ and $c_c$ being positive constants.

## 4.2 FINITE-TIME CONVERGENCE RESULTS FOR ALGORITHM 1

We provide here the non-asymptotic convergence results for both the actor and the critic recursions in Algorithm 1. We also present the convergence rate and sample complexity of the algorithm.

### 4.2.1 Convergence of the actor recursion for Algorithm 1

We have the following result after carrying out the non-asymptotic analysis of the actor.

**Theorem 1** *At the $t$-th iteration we have,*

$$\min_{0\leq m\leq t} \mathbb{E}\left\|\nabla_{\boldsymbol{\theta}}L(\boldsymbol{\theta}_m,\boldsymbol{\gamma}(m))\right\|^2 = \mathcal{O}(\epsilon_{app}) + \mathcal{O}(t^{\sigma-\beta})$$

$$+\mathcal{O}\left(\frac{\log^2 t}{t^\sigma}\right) + \mathcal{O}\left(\frac{\sum_{k=\tau_t}^t E\|A_k\|^2}{1+t-\tau_t}\right)$$

$$+\mathcal{O}\left(\frac{\sum_{k=\tau_t}^t E\|B_k\|^2}{1+t-\tau_t}\right),$$

*where*

$$A_k = L_k - L(\boldsymbol{\theta}_k,\boldsymbol{\gamma}(k)), \quad (5)$$
$$B_k = v_k - v^*(\theta_k,\gamma(k)). \quad (6)$$

### 4.2.2 Convergence of the critic recursion for Algorithm 1

For the critic recursion, we obtain the following result for the average estimation error.

**Theorem 2** *We have*

$$\frac{1}{1+t-\tau_t}\sum_{k=\tau_t}^t \mathbb{E}\|v_k - v^*(\theta_k,\gamma(k))\|^2$$
$$= \mathcal{O}\left(\frac{1}{t^{1-\omega}}\right) + \mathcal{O}\left(\frac{\log t}{t^\omega}\right) + \mathcal{O}\left(\frac{1}{t^{2(\sigma-\omega)}}\right), \quad (7)$$

$$\frac{1}{1+t-\tau_t}\sum_{k=\tau_t}^t \mathbb{E}\left(L_k - L(\boldsymbol{\theta}_k,\boldsymbol{\gamma}(k))\right)^2$$
$$= \mathcal{O}\left(\frac{1}{t^{1-\omega}}\right) + \mathcal{O}\left(\frac{\log t}{t^\omega}\right) + \mathcal{O}\left(\frac{1}{t^{2(\sigma-\omega)}}\right). \quad (8)$$

### 4.2.3 Convergence rate and sample complexity for Algorithm 1

We finally provide the convergence rate of the algorithm and characterize the sample complexity of the same in Corollary 1.

**Corollary 1** *We have,*

$$\min_{0\leq k\leq t} \mathbb{E}\|\nabla_{\boldsymbol{\theta}}L(\boldsymbol{\theta}_k,\boldsymbol{\gamma}(k))\|^2 = \mathcal{O}(\epsilon_{app}) + \mathcal{O}\left(\frac{1}{t^{\beta-\sigma}}\right)$$

$$+\mathcal{O}\left(\frac{\log^2 t}{t^\omega}\right) + \mathcal{O}\left(\frac{1}{t^{2(\sigma-\omega)}}\right).$$

*If we set $\omega = 0.4, \sigma = 0.6, \beta = 1$, Algorithm 1 needs $T = \tilde{\mathcal{O}}(\epsilon^{-2.5})$ steps to obtain the following:*

$$\min_{0\leq k\leq T} \mathbb{E}\left\|\nabla_{\boldsymbol{\theta}}L(\boldsymbol{\theta}_k,\boldsymbol{\gamma}(k))\right\|^2 \leq \mathcal{O}(\epsilon_{app}) + \epsilon.$$

**Remark 2** *In Corollary 1, the results of Theorems 1 and 2 are combined which results in the convergence rate of Algorithm 1 as $\tilde{\mathcal{O}}(t^{-0.4})$. The sample complexity of the constrained actor-critic algorithm is $\tilde{\mathcal{O}}(\epsilon^{-2.5})$ as we have exactly one sample per iteration.*

## 4.3 FINITE-TIME CONVERGENCE RESULTS FOR ALGORITHM 2

As with Algorithm 1, we provide here the non-asymptotic convergence results for both the actor and the critic recursions in Algorithm 2. Further, we present the convergence rate and sample complexity of the algorithm.

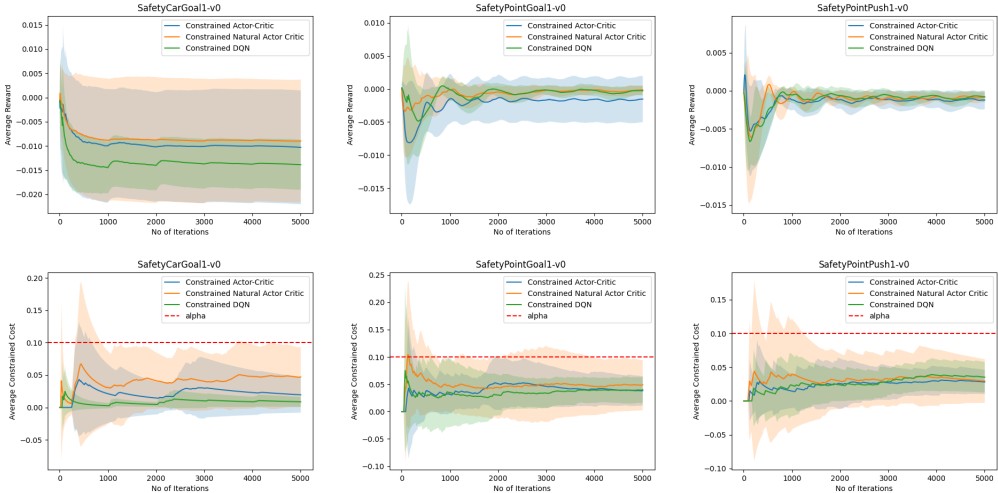

Figure 1: Comparison of C-AC , C-NAC and C-DQN: Plots in the top row are for the average reward performance while those in the bottom row are for the constraint costs for the three environments. These are plotted as functions of the number of iterations.

#### 4.3.1 Convergence of the Actor for Algorithm 2

We have the following result after carrying out non-asymptotic analysis of the actor.

**Theorem 3** *At the tth iteration, we have,*

$$
\min_{0 \leq m \leq t} \mathbb{E}\big\|\nabla_{\boldsymbol{\theta}} L(\boldsymbol{\theta}_m, \boldsymbol{\gamma}(m))\big\|^2
$$
$$
= \mathcal{O}(\epsilon_{app}) + \mathcal{O}(t^{\sigma - \beta})
$$
$$
+ \mathcal{O}\left(\frac{\log^2 t}{t^\omega}\right) + \mathcal{O}\left(\frac{\sum_{k=\tau_t}^{t} E\|A_k\|^2}{1 + t - \tau_t}\right)
$$
$$
+ \mathcal{O}\left(\frac{\sum_{k=\tau_t}^{t} E\|B_k\|^2}{1 + t - \tau_t}\right),
$$

*where*

$$
A_k = L_k - L(\boldsymbol{\theta}_k, \boldsymbol{\gamma}(k)), \tag{9}
$$
$$
B_k = v_k - v^*(\boldsymbol{\theta}_k, \boldsymbol{\gamma}(k)). \tag{10}
$$

#### 4.3.2 Convergence of the Critic for Algorithm 2

We now provide a result analysing the average estimation error for the critic.

**Theorem 4** *We have the following:*

$$
\frac{1}{1 + t - \tau_t} \sum_{k=\tau_t}^{t} \mathbb{E}\|v_k - v^*(\boldsymbol{\theta}_k, \boldsymbol{\gamma}(k))\|^2
$$
$$
= \mathcal{O}\left(\frac{1}{t^{1-\omega}}\right) + \mathcal{O}\left(\frac{\log t}{t^\omega}\right) + \mathcal{O}\left(\frac{1}{t^{2(\sigma-\omega)}}\right), \tag{11}
$$
$$
\frac{1}{1 + t - \tau_t} \sum_{k=\tau_t}^{t} \mathbb{E}\big(L_k - L(\boldsymbol{\theta}_k, \boldsymbol{\gamma}(k))\big)^2
$$
$$
= \mathcal{O}\left(\frac{1}{t^{1-\omega}}\right) + \mathcal{O}\left(\frac{\log t}{t^\omega}\right) + \mathcal{O}\left(\frac{1}{t^{2(\sigma-\omega)}}\right). \tag{12}
$$

#### 4.3.3 Convergence rate and sample complexity for Algorithm 2

We finally provide the convergence rate of the algorithm and characterize the sample complexity of the same in Corollary 2.

**Corollary 2** *We have*

$$
\min_{0 \leq k \leq t} \mathbb{E}\|\nabla_{\boldsymbol{\theta}} L(\boldsymbol{\theta}_k, \boldsymbol{\gamma}(k))\|^2 = \mathcal{O}(\epsilon_{app}) + \mathcal{O}\left(\frac{1}{t^{\beta-\sigma}}\right)
$$
$$
+ \mathcal{O}\left(\frac{\log^2 t}{t^\omega}\right) + \mathcal{O}\left(\frac{1}{t^{2(\sigma-\omega)}}\right).
$$

*If we set $\omega = 0.4, \sigma = 0.6, \beta = 1$, Algorithm 2 needs $T = \tilde{\mathcal{O}}(\epsilon^{-2.5})$ steps to obtain the following,*

$$
\min_{0 \leq k \leq T} \mathbb{E}\big\|\nabla_{\boldsymbol{\theta}} L(\boldsymbol{\theta}_k, \gamma(k))\big\|^2 \leq \mathcal{O}(\epsilon_{app}) + \epsilon,
$$

Table 2: Comparision of C-AC , C-NAC and C-DQN in terms of average reward ± standard error upon convergence.

| Algorithm | SafetyPointGoal1-v0 | SafetyCarGoal1-v0 | SafetyPointPush1-v0 |
|-----------|---------------------|-------------------|---------------------|
| C-AC | $-0.0015 \pm 0.0035$ | $-0.01 \pm 0.0117$ | $-0.0012 \pm 0.0011$ |
| C-NAC | $-0.00012 \pm 0.0006$ | $-0.009 \pm 0.0127$ | $-0.0008 \pm 0.0005$ |
| C-DQN | $-0.0003 \pm 0.0007$ | $-0.014 \pm 0.005$ | $-0.0008 \pm 0.0007$ |

**Remark 3** *Analogous to Remark 2, in Corollary 2, the results of Theorems 3 and 4 are combined which gives the convergence rate of (natural actor critic) Algorithm 2 as $\tilde{\mathcal{O}}(t^{-0.4})$ and a sample complexity of $\tilde{\mathcal{O}}(\epsilon^{-2.5})$ as we have one per-iteration sample in this algorithm as well. It is important to also mention that for the results of both algorithms, $\mathcal{O}(\cdot)$ hides the terms that do not depend on the iteration number.*

## 4.4 PROOF SKETCH FOR THEOREMS 1 AND 3

We provide here an overview of the manner in which the proofs of Theorems 1 and 3 proceed. This also helps us to describe the connection between the various results mentioned above. The detailed arguments are nonetheless provided in the appendix. The proofs of Theorems 1 and 3 rely crucially on Lemma 1 below.

**Lemma 1** *For all $\boldsymbol{\gamma} \in R^N$ with $0 \le \gamma_i \le M$ where $i \in \{1, 2, \ldots, N\}$, there exists a constant $M_L$ greater than 0 such that for all $\theta_1, \theta_2 \in R^d$,*

$$\|\nabla_\theta L(\theta_1, \boldsymbol{\gamma}) - \nabla_\theta L(\theta_2, \boldsymbol{\gamma})\| \le M_L \|\theta_1 - \theta_2\|.$$

As a result of this lemma, we have the following inequality (see Wu et al., [2020], Lemma C.1) $\forall t > 0$, that we use in the proof of Theorem 1.

$$L(\theta_{t+1}, \boldsymbol{\gamma}(t)) \ge L(\theta_t, \boldsymbol{\gamma}(t))$$

$$+ b(t)\langle \nabla_\theta L(\theta_t, \boldsymbol{\gamma}(t)), \delta_t \nabla \log \pi_{\theta_t}(a_t|s_t)\rangle$$

$$- \frac{M_L}{2} b(t)^2 \|\delta_t \nabla \log \pi_{\theta_t}(a_t|s_t)\|^2. \quad (13)$$

The key idea in the proof (see Appendix for details) is to split the middle term in the RHS of (13) into a few terms, one of which is $b(t)\|\nabla L(\boldsymbol{\theta}_t, \boldsymbol{\gamma}(t))\|^2$. We obtain an upper bound for $\|\nabla L(\boldsymbol{\theta}_t, \boldsymbol{\gamma}(t))\|^2$. After summing the expectation of terms on both sides, we analyse each term in the bound to get the desired result for Theorem 1. Note also that the result for Theorem 1 depends on the convergence of the critic parameter and the average cost estimator. So we find a bound on the averaged estimation errors by the critic and average cost estimator in Theorem 2. We then obtain an inequality similar to (13) for Theorem 3 (see Appendix) and carry out an analysis along similar lines for Theorems 3 and 4.

## 4.5 PROOF SKETCH FOR THEOREMS 2 AND 4

We provide here an overview of the manner in which the proofs of Theorems 2 and 4 proceed.

*Proof Sketch for Average Reward estimate Error*
For proving the convergence of average reward estimate in theorems 2 and 4, we start off with expanding $y_{t+1}^2 = (L_{t+1} - L_{t+1}^*)^2$ which gives us the following inequality.

$$y_{t+1}^2 \le (1 - 2a(t))y_t^2 + 2a(t)y_t(C_t - L_t^*)$$

$$+ 2y_t(L_t^* - L_{t+1}^*) + 2(L_t^* - L_{t+1}^*)^2 + 2a(t)^2(C_t - L_t)^2,$$

where $C_t = q(t) + \sum_{k=1}^N \gamma_k(t)(h_k(t) - \alpha_k)$. After rearranging and summing the expectation of both sides from $\tau_t$ to $t$ we have,

$$\sum_{k=\tau_t}^t E[y_k^2] \le \underbrace{\sum_{k=\tau_t}^t \frac{1}{2a(k)} E(y_k^2 - y_{k+1}^2)}_{I_1}$$

$$+ \underbrace{\sum_{k=\tau_t}^t E[\hat{\Xi}(L_k, \theta_k, \gamma(k), q(k), h(k))]}_{I_2}$$

$$+ \underbrace{\sum_{k=\tau_t}^t \frac{1}{a(k)} E[y_k(L_k^* - L_{k+1}^*)]}_{I_3}$$

$$+ \underbrace{\sum_{k=\tau_t}^t \frac{1}{a(k)} E[(L_k^* - L_{k+1}^*)^2]}_{I_4}$$

$$+ \underbrace{\sum_{k=\tau_t}^t a(k) E[(C_k - L_k)^2]}_{I_5}.$$

Upon bounding $I_1 - I_5$ and simplifying, we get the desired result. For the definition of the notations involved, please refer Section B.2.

*Proof Sketch for Critic Convergence*
For proving the convergence of critic in Theorems 2 and 4, we start off with the equation $\|m_{t+1}\|^2 = \|\Gamma(v_t + a(t)\delta_t f_{s_t}) - v^*(t+1)\|^2$ and after expanding the RHS, we

get the following inequality:

$$\|m_{t+1}\|^2 \leq \|m_t\|^2 + 2a(t)\langle m_t, \overline{g}(v_t, \theta_t, \gamma(t))\rangle$$
$$+ 2a(t)\Lambda(O_t, v_t, \theta_t, \gamma(t), q(t), h(t))$$
$$+ 2a(t)\langle m_t, \Delta g(O_t, L_t, \theta_t, \gamma(t))\rangle$$
$$+ 2\langle m_t, v^*(t) - v^*(t+1)\rangle$$
$$+ 8a(t)^2(U_r + U_v)^2 + 2\|v^*(t) - v^*(t+1)\|^2.$$

After rearranging and summing the expectation on both sides from $\tau_t$ to $t$ we obtain (see Appendix),

$$2\lambda_e \sum_{k=\tau_t}^{t} E\|m_k\|^2$$
$$\leq \underbrace{\sum_{k=\tau_t}^{t} \frac{1}{a(k)}\left(E\|m_k\|^2 - E\|m_{k+1}\|^2\right)}_{I_1}$$
$$+ \underbrace{2\sum_{k=\tau_t}^{t} E\Lambda(O_k, v_k, \theta_k, \gamma(k), q(k), h(k))}_{I_2}$$
$$+ \underbrace{2\sum_{k=\tau_t}^{t} \sqrt{E\|m_k\|^2}\sqrt{E[y_k^2]}}_{I_3}$$
$$+ \underbrace{B_q \sum_{k=\tau_t}^{t} \frac{b(k)}{a(k)})\sqrt{E\|m_k\|^2}}_{I_4} + \underbrace{C_q \sum_{k=\tau_t}^{t} a(k)}_{I_5}.$$

After analysing terms $I_1 - I_5$, we get the desired result. For definition of the notations involved, please refer section B.3.

**Remark 4** *As mentioned previously, the asymptotic stability and almost sure convergence of the three-timescale C-AC algorithm has been shown in Bhatnagar and Lakshmanan [2012]. A similar analysis combining the results of Bhatnagar et al., [2009] would also provide similar stability and asymptotic convergence results for the three-timescale C-NAC algorithm. Note that while the non-asymptotic convergence results that we have shown are to the stationary points of the Lagrangian, it is shown in Bhatnagar and Lakshmanan [2012] that for C-AC, stationary-point convergence indeed results in a locally optimal policy, that gives the set of local minima of the objective in the constraint set, see Proposition 4.3 and Remarks 4.3-4.5 there. This is because stationary points that are not local minima are unstable attractors of the underlying ODE. The same is also true of the C-NAC algorithm. Some works such as Zeng et al., [2022] provide bounds on the optimality gap but they primarily consider the look-up table setting and not function approximation. Defining optimality gap precisely in our setting is hard due to the presence of multiple local minima in*

*the constraint set and so obtaining such bounds is not easy unless one has a setting of convex objectives and constraints (unlike us). From results in stochastic approximation theory (Kushner and Clark [1978]), if the stochastic recursion enters a compact neighbourhood of a local minimum infinitely often, it will converge to it w.p.1. The compact neighbourhood of which of the minima the recursion enters in will depend on the initial condition and noise.*

## 5 EXPERIMENTAL RESULTS

In this section, we present the results of our experiments on three different OpenAI Safety-Gym environments: (a) SafetyPointGoal1-v0, (b) SafetyCarGoal1-v0 and (c) SafetyPointPush1-v0, respectively. The performance comparisons on these environments can be seen in Figure 1 and Tables 2 and 3, respectively. The two tables summarize the performance obtained upon convergence of the algorithms. Note that Table 3 has been placed in the appendix for lack of space. We also explain, in the appendix, the Constrained DQN algorithm that we implemented in addition.

All the plots of our experiments are obtained after averaging over 10 different initial seeds. The performance of the algorithms is compared by plotting the average reward along with standard errors. The dotted flat red-line in each of the plots in the lower row of plots in Figure 1 corresponds to the constraint cost threshold. All algorithms are seen to asymptotically satisfy the constraint threshold while optimizing on the average reward performance.

It can be observed that our C-NAC algorithm performs better than the other two algorithms on all three settings and C-AC shows the second best results on two of the three environments. Moreover, the cost threshold is met by all the three algorithms[1].

## 6 CONCLUSIONS

We presented the first (non-asymptotic) finite time convergence analysis of three-timescale constrained actor-critic and constrained natural actor-critic algorithms using linear function approximation and obtained a sample complexity of $\tilde{\mathcal{O}}(\epsilon^{-2.5})$ for both algorithms. Our sample complexity result is significant as for both our algorithms it matches the sample complexity of regular (unconstrained) actor-critic (two-timescale) algorithms analysed in Wu et al., [2020]. We also showed the results of experiments on three different Safety-Gym environments, where we observed that the C-NAC algorithm is better than both the C-AC and the C-DQN algorithms in the average reward performance and the C-AC algorithm is the second best performer on two of these three settings. Further, the average cost constraint is met by all the three algorithms on each of the settings.

---

[1] The code for all of our experiments is available at https://github.com/prashu1306/Constrained-Actor-Critic

## Acknowledgements

The authors were supported by the Walmart Centre for Tech Excellence, Indian Institute of Science. In addition, S. Bhatnagar was supported by the J. C. Bose National Fellowship of SERB, Project No. DFTM/02/3125/M/04/AIR-04 from DRDO under DIA-RCOE and the Robert Bosch Centre for Cyber Physical Systems, Indian Institute of Science.

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

# Appendix

**Prashansa Panda**[1]                    **Shalabh Bhatnagar**[1]

[1]Department of Computer Science and Automation, Indian Institute of Science, Bangalore, India

## A  PRELIMINARIES

For our problem, we have considered random single-stage costs and also constraint costs having distributions that are dependent on the current state, action and next state. For a given tuple $O_t = (s_t, a_t, s_{t+1})$, we consider

$$q(t) \sim \bar{p}(.|s_t, a_t, s_{t+1}), \; h_i(t) \sim p_i(.|s_t, a_t, s_{t+1}),$$

for $i = 1, 2, 3, .., N$. For a given $\gamma = (\gamma_1, \gamma_2, ..., \gamma_N)^T$ with $0 \leq \gamma_i \leq M$ and for $i = 1, 2, .., N$, we define $c(t) = q(t) + \sum_{k=1}^{N} \gamma_k(h_k(t) - \alpha_k)$ such that

$$c(t) \sim \hat{p}(.|s_t, a_t, s_{t+1}).$$

Since the single stage costs are mutually independent for any state-action-next state tuple, we have

$$\hat{p}(c(t)|s_t, a_t, s_{t+1}) = \bar{p}(q(t)|s_t, a_t, s_{t+1}) \prod_{i=1}^{N} p_i(h_i(t)|s_t, a_t, s_{t+1}).$$

Let $\|\nabla L(\theta, \gamma)\| < U_L, \forall \theta, \gamma$ where $U_L > 0$ is a positive constant. Also, $\sup_t \|G(t)^{-1}\| \leq U_G$ where $U_G$ is a positive constant. Recall that the single stage costs $q(n), h_1(n), \ldots, h_N(n)$ are non-negative and absolutely bounded by a constant $U_c > 0$. We define $U_r = U_c + NM(U_c + U_\alpha)$ where $U_\alpha = \max_k |\alpha_k|$. We assume $0 \leq L_0 \leq U_r$. This implies $|L_t| \leq U_r, \forall t > 0$. We also assume $0 \leq U_k(0) \leq U_c, \forall k \in 1, 2, .., N$, which implies $|U_k(t)| \leq U_c, \forall t > 0$. Moreover, we assume that the projection set $C$ to which the recursion $v_t$ is projected is a compact and convex set.

Given time indices $t$ and $\tau$ such that $t \geq \tau > 0$, we consider the following auxiliary Markov chain starting from $s_{t-\tau}$ to deal with Markov noise in the iterates.

$$s_{t-\tau} \xrightarrow{\theta_{t-\tau}} a_{t-\tau} \xrightarrow{\mathcal{P}} s_{t-\tau+1} \xrightarrow{\theta_{t-\tau}} \widetilde{a}_{t-\tau+1} \xrightarrow{\mathcal{P}} \widetilde{s}_{t-\tau+2} \xrightarrow{\theta_{t-\tau}} \widetilde{a}_{t-\tau+2} \cdots \xrightarrow{\mathcal{P}} \widetilde{s}_t \xrightarrow{\theta_{t-\tau}} \widetilde{a}_t \xrightarrow{\mathcal{P}} \widetilde{s}_{t+1}. \tag{14}$$

The original Markov chain has the following transitions:

$$s_{t-\tau} \xrightarrow{\theta_{t-\tau}} a_{t-\tau} \xrightarrow{\mathcal{P}} s_{t-\tau+1} \xrightarrow{\theta_{t-\tau+1}} a_{t-\tau+1} \xrightarrow{\mathcal{P}} s_{t-\tau+2} \xrightarrow{\theta_{t-\tau+2}} a_{t-\tau+2} \cdots \xrightarrow{\mathcal{P}} s_t \xrightarrow{\theta_t} a_t \xrightarrow{\mathcal{P}} s_{t+1}. \tag{15}$$

In (14), for any time instant $k > t - \tau$, $\widetilde{a}_k$ denotes the action taken in the auxiliary Markov chain. Similarly, for any time instant $l > t - \tau + 1$, $\widetilde{s}_l$ denotes the state in the auxiliary Markov chain. In the following, without any other specification, $E[.]$ will denote the expectation w.r.t the joint distribution of all the random variables involved. Finally, we mention that for our analysis we interchangeably use $\pi(a|s)$ and $\pi(s, a)$ to mean the action chosen in state $s$ according to the policy $\pi$. Thus, both these notations are one and the same.

# B   PROOF OF THEOREMS

## B.1   PROOF OF THEOREM 1

We first define several notations to clarify the dependence in the various quantities involved. Let

$$O_t := (s_t, a_t, s_{t+1}),$$

$$L^* := L(\theta, \gamma) = E_{s \sim \mu_\theta, a \sim \pi_\theta}[d(s,a) + \sum_{k=1}^{N} \gamma(k)(h_k(s,a) - \alpha_k)],$$

$$v(k)^* := v^*(\theta_k, \gamma(k)),$$

$$\Delta H(O, L, v, \theta, \gamma) := (L(\theta, \gamma) - L + (f_{s'}{}^T - f_s^T)(v - v^*(\theta, \gamma)))\nabla \log \pi_\theta(a|s), \tag{16}$$

$$H(O, \theta, \gamma, q, h) := (q - L(\theta, \gamma) + \sum_{k=1}^{N} \gamma(k)(h_k - \alpha_k) + (f_{s'}{}^T - f_s^T)v^*(\theta, \gamma))\nabla \log \pi_\theta(a|s),$$

$$\Delta H'(O, \theta, \gamma) := (f_{s'}{}^T v^*(\theta, \gamma) - V^{\theta, \gamma}(s') - (f_s^T v^*(\theta, \gamma) - V^{\theta, \gamma}(s)))\nabla \log \pi_\theta(a|s),$$

$$\check{\Gamma}(O, \theta, \gamma, q, h) := \langle \nabla L(\theta, \gamma), H(O, \theta, \gamma, q, h) - E_{O', q, h}[H(O', \theta, \gamma, q, h)]\rangle.$$

In the above, $O = (s, a, s')$, $h = (h_1, h_2, h_3, ..., h_N)$. $O' = (s, a, s')$ denotes the independent sample $s \sim \mu_\theta$, $a \sim \pi_\theta$, $s' \sim p(s, ., a)$. Hence $E_{O', q, h}[\cdot]$ denotes expectation w.r.t. the joint distribution of $s \sim \mu_\theta$, $a \sim \pi_\theta$, $s' \sim p(s, ., a)$, $q \sim \bar{p}(.|s, a, s')$, $h_i \sim p_i(.|s, a, s')$, $i = 1, \ldots, N$. We now have,

$$E_{O', q, h}[(H(O', \theta, \gamma, q, h) - \Delta H'(O', \theta, \gamma)]$$

$$= E_{O', q, h}[q + \sum_{k=1}^{N} \gamma(k)(h_k - \alpha_k) - L(\theta, \gamma) + V^{\theta, \gamma}(s') - V^{\theta, \gamma}(s))\nabla \log \pi_\theta(a|s)]$$

$$= \nabla L(\theta, \gamma),$$

Next, observe that

$$E_{O'}\|\Delta H'(O, \theta, \gamma)\|^2 = E_{O'}\|(f_{s'}{}^T v^*(\theta, \gamma) - V^{\theta, \gamma}(s') - (f_s^T v^*(\theta, \gamma) - V^{\theta, \gamma}(s)))\nabla \log \pi_\theta(a|s)\|^2$$

$$\leq E_{O'}[D^2(f_{s'}{}^T v^*(\theta, \gamma) - V^{\theta, \gamma}(s') - (f_s^T v^*(\theta, \gamma) - V^{\theta, \gamma}(s)))^2]$$

$$\leq E_{O'}[2D^2((f_{s'}{}^T v^*(\theta, \gamma) - V^{\theta, \gamma}(s'))^2 + (f_s^T v^*(\theta, \gamma) - V^{\theta, \gamma}(s))^2)]$$

$$\leq 4D^2 \epsilon_{app}^2.$$

Before we proceed further, we first state and prove Lemmas 1–4 below that will be used in the proof of Theorem 1. Moreover, the proof of Lemma 4 shall rely on Lemmas 4.1 – 4.4 that we also state and prove in the following. Finally, collecting all these results together, we shall obtain the claim for Theorem 1.

**Lemma 1** *For all* $\gamma \in R^N$, *with* $0 \leq \gamma_i \leq M$, *where* $i \in \{1, 2, ..., N\}$, *there exists a constant* $M_L$ *greater than 0 such that for all* $\theta_1, \theta_2 \in R^d$,

$$\|\nabla L(\theta_1, \gamma) - \nabla L(\theta_2, \gamma)\| \leq M_L \|\theta_1 - \theta_2\|,$$

*which implies*

$$L(\theta_2, \gamma) > L(\theta_1, \gamma) + \langle \nabla L(\theta_1, \gamma), \theta_2 - \theta_1 \rangle - \frac{M_L}{2}\|\theta_1 - \theta_2\|^2.$$

**Proof**   Note that

$$L(\theta, \gamma) = J(\theta) + \sum_{k=1}^{N} \gamma(k)(G_k(\theta) - \alpha_k).$$

$$\Rightarrow \nabla_\theta L(\theta, \gamma) = \nabla_\theta J(\theta) + \sum_{k=1}^{N} \gamma(k)\nabla_\theta G_k(\theta).$$

From Lemma C.1 of Wu et al., [2020], we have $\forall \theta_1, \theta_2 \in R^d$, there exist positive constants $L_J, L_{G_1}, L_{G_2}, ..., L_{G_N}$ such that

$$\|\nabla_\theta J(\theta_1) - \nabla_\theta J(\theta_2)\| \leq L_J \|\theta_1 - \theta_2\|,$$
$$\|\nabla_\theta G_i(\theta_1) - \nabla_\theta G_i(\theta_2)\| \leq L_{G_i} \|\theta_1 - \theta_2\|, \forall i \in 1, 2, .., N.$$

Hence,

$$
\begin{aligned}
\|\nabla_\theta L(\theta_1, \gamma) - \nabla_\theta L(\theta_2, \gamma)\| &= \|\nabla_\theta J(\theta_1) - \nabla_\theta J(\theta_2) + \sum_{k=1}^{N} \gamma_k (\nabla_\theta G_k(\theta_1) - \nabla_\theta G_k(\theta_2))\| \\
&\leq \|\nabla_\theta J(\theta_1) - \nabla_\theta J(\theta_2)\| + \sum_{k=1}^{N} \gamma_k \|\nabla_\theta G_k(\theta_1) - \nabla_\theta G_k(\theta_2)\| \\
&\leq L_J \|\theta_1 - \theta_2\| + \sum_{k=1}^{N} \gamma_k L_{G_k} \|\theta_1 - \theta_2\| \\
&= (L_J + \sum_{k=1}^{N} \gamma_k L_{G_k}) \|\theta_1 - \theta_2\| \\
&\leq M_L \|\theta_1 - \theta_2\|.
\end{aligned}
$$

So we have,

$$\|\nabla_\theta L(\theta_1, \gamma) - \nabla_\theta L(\theta_2, \gamma)\| \leq M_L \|\theta_1 - \theta_2\|, \forall \theta_1, \theta_2 \in R^d,$$

where $M_L = L_J + M \sum_{k=1}^{N} L_{G_k}$. The claim follows. ∎

**Lemma 2** *For all $\gamma^1, \gamma^2 \in R^N$ with $0 \leq \gamma_i^j \leq M$, where $i \in \{1, 2, ..., N\}$, $j = 1, 2$, there exists a constant $C$ greater than 0 such that for all $\theta \in R^d$,*

$$\|\nabla L(\theta, \gamma^1) - \nabla L(\theta, \gamma^2)\| \leq C |\gamma_m^1 - \gamma_m^2|,$$

*where $|\gamma_m^1 - \gamma_m^2| = \max\limits_{i=1,2,..,N} |\gamma_i^1 - \gamma_i^2|$.*

**Proof** We have,

$$
\begin{aligned}
\|\nabla L(\theta, \gamma^1) - \nabla L(\theta, \gamma^2)\| &= \left\| \sum_{s \in S} \mu_\theta(s) \sum_{a \in A(s)} \pi_\theta(a|s) (M^{\pi_\theta, \gamma^1}(s, a) - M^{\pi_\theta, \gamma^2}(s, a)) \nabla \log \pi_\theta(a|s) \right\| \\
&\leq \sum_{s \in S} \mu_\theta(s) \sum_{a \in A(s)} \pi_\theta(a|s) \left\| (M^{\pi_\theta, \gamma^1}(s, a) - M^{\pi_\theta, \gamma^2}(s, a)) \nabla \log \pi_\theta(a|s) \right\| \\
&\leq D \sum_{s \in S} \mu_\theta(s) \sum_{a \in A(s)} \pi_\theta(a|s) \left| M^{\pi_\theta, \gamma^1}(s, a) - M^{\pi_\theta, \gamma^2}(s, a) \right|,
\end{aligned}
$$

where the last inequality follows from Assumption 4. Now,

$$
\begin{aligned}
\left| M^{\pi_\theta, \gamma^1}(s, a) - M^{\pi_\theta, \gamma^2}(s, a)) \right| &= \left| \sum_{t=1}^{\infty} E\left[ \sum_{i=1}^{N} (\gamma_i^1 - \gamma_i^2)(h_i(t) - G_i(\theta)) | s_0 = s, a_0 = a, \pi_\theta \right] \right| \\
&= \left| \sum_{i=1}^{N} (\gamma_i^1 - \gamma_i^2) \sum_{t=1}^{\infty} E\left[ (h_i(t) - G_i(\theta)) | s_0 = s, a_0 = a, \pi_\theta \right] \right| \\
&\leq \sum_{i=1}^{N} |(\gamma_i^1 - \gamma_i^2)| \left| \sum_{t=1}^{\infty} E\left[ (h_i(t) - G_i(\theta)) | s_0 = s, a_0 = a, \pi_\theta \right] \right| \\
&\leq N \bar{Q} |\gamma_m^1 - \gamma_m^2|,
\end{aligned}
$$

where,

$$\bar{Q} = \sup_{i,\boldsymbol{\theta},s,a} |Q_i^{\boldsymbol{\theta}}(s,a)|,$$

$$|\boldsymbol{\gamma}_m^1 - \boldsymbol{\gamma}_m^2| = \max_{i=1,2,..,N} |\boldsymbol{\gamma}_i^1 - \boldsymbol{\gamma}_i^2|,$$

and where $Q_i^{\boldsymbol{\theta}}(s,a)$ is the expected differential cost for the state action pair $(s,a)$ with single-stage cost as $h_i(t)$ at time instant $t$ when actions are picked according to policy $\pi_{\boldsymbol{\theta}}$. Hence,

$$\|\nabla L(\boldsymbol{\theta}, \boldsymbol{\gamma}^1) - \nabla L(\boldsymbol{\theta}, \boldsymbol{\gamma}^2)\| \leq ND\bar{Q}|\boldsymbol{\gamma}_m^1 - \boldsymbol{\gamma}_m^2|.$$

The claim follows by letting $C = ND\bar{Q}$. ∎

**Lemma 3** *For any $t > 0$,*

$$\|\Delta H(O_t, L_t, v_t, \theta_t, \gamma(t))\|^2 \leq D^2(2(L_t - L(\theta_t, \gamma(t)))^2 + 8\|v_t - v(t)^*\|^2).$$

**Proof** Applying the definition of $\Delta H(\cdot)$ immediately yields the result. ∎

**Lemma 4** *For any $t \geq 0$,*

$$E[\check{\Gamma}(O_t, \theta_t, \gamma(t), q(t), h(t))] \geq -(D_1(\tau+1) \sum_{k=t-\tau+1}^{t} E\|\theta_k - \theta_{k-1}\| + D_2 bk^{\tau-1} + T_1 \sum_{i=t-\tau+1}^{t} E|\gamma_m(i) - \gamma_m(i-1)|),$$

*where $D_1, D_2$ and $T_1$ are positive constants and $t \geq \tau \geq 0$.*

**Proof** We have

$$E[\check{\Gamma}(O_t, \theta_t, \gamma(t), q(t), h(t))]$$
$$= E_{s_t \sim p, a_t \sim \pi_{\theta_t}, s_{t+1} \sim p}[E[\langle \nabla L(\theta_t, \gamma(t)), H(O_t, \theta_t, \gamma(t), q(t), h(t)) - E_{O',q,h}[H(O', \theta_t, \gamma(t), q, h)]\rangle|s_t, a_t, s_{t+1}]]$$
$$= E[\langle \nabla L(\theta_t, \gamma(t)), \bar{H}(O_t, \theta_t, \gamma(t)) - E_{O',q,h}[H(O', \theta_t, \gamma(t), q, h)]\rangle]$$
$$= E[\langle \nabla L(\theta_t, \gamma(t)), \bar{H}(O_t, \theta_t, \gamma(t)) - E_{O'}[E_{q,h}[H(O', \theta_t, \gamma(t), q, h)|s, a, s']]\rangle]$$
$$= E[\langle \nabla L(\theta_t, \gamma(t)), \bar{H}(O_t, \theta_t, \gamma(t)) - E_{O'}[\bar{H}(O', \theta_t, \gamma(t))]\rangle]$$
$$= E[Q(O_t, \theta_t, \gamma(t))],$$

where,

$$\bar{H}(O, \theta, \gamma) = (c(s, a, s', \gamma) - L(\theta, \gamma) + (f_{s'}{}^T - f_s{}^T)v^*(\theta, \gamma))\nabla \log \pi_\theta(a|s)$$

$$c(s, a, s', \gamma) = \sum_q (q \cdot \bar{p}(q|s, a, s')) + \sum_{k=1}^{k=N} \gamma_k(\sum_h (h \cdot p_k(h|s, a, s')) - \alpha_k).$$

The second equality is satisfied because $\theta_t$ and $\gamma(t)$ do not depend on $q(t)$ and $h(t)$. The remaining proof of Lemma 4 in turn requires Lemmas 4.1 – 4.4 below that we now state and prove.

**lemma 4.1** *For any $t \geq 0$,*

$$|Q(O_t, \theta_t, \gamma(t)) - Q(O_t, \theta_t, \gamma(t-\tau))| \leq T_1|\gamma_m(t) - \gamma_m(t-\tau)|,$$

*where $T_1 > 0$ is a constant.*

*Proof* Denoting $O = (s, a, s^{'})$, we have for any $\theta, \gamma_1, \gamma_2$, that

$$Q(O, \theta, \gamma^1) - Q(O, \theta, \gamma^2)$$
$$= \langle \nabla L(\theta, \gamma^1), \bar{H}(O, \theta, \gamma^1) - E_{O'}[\bar{H}(O^{'}, \theta, \gamma^1)] \rangle - \langle \nabla L(\theta, \gamma^2), \bar{H}(O, \theta, \gamma^2) - E_{O'}[\bar{H}(O^{'}, \theta, \gamma^2)] \rangle$$
$$= \langle \nabla L(\theta, \gamma^1), \bar{H}(O, \theta, \gamma^1) - E_{O'}[\bar{H}(O^{'}, \theta, \gamma^1)] \rangle - \langle \nabla L(\theta, \gamma^1), \bar{H}(O, \theta, \gamma^2) - E_{O'}[\bar{H}(O^{'}, \theta, \gamma^2)] \rangle$$
$$+ \langle \nabla L(\theta, \gamma^1), \bar{H}(O, \theta, \gamma^2) - E_{O'}[\bar{H}(O^{'}, \theta, \gamma^2)] \rangle - \langle \nabla L(\theta, \gamma^2), \bar{H}(O, \theta, \gamma^2) - E_{O'}[\bar{H}(O^{'}, \theta, \gamma^2)] \rangle$$
$$= \underbrace{\langle \nabla L(\theta, \gamma^1), \bar{H}(O, \theta, \gamma^1) - \bar{H}(O, \theta, \gamma^2) - E_{O'}[\bar{H}(O^{'}, \theta, \gamma^1)] + E_{O'}[\bar{H}(O^{'}, \theta, \gamma^2)] \rangle}_{I_1}$$
$$+ \underbrace{\langle \nabla L(\theta, \gamma^1) - \nabla L(\theta, \gamma^2), \bar{H}(O, \theta, \gamma^2) - E_{O'}[\bar{H}(O^{'}, \theta, \gamma^2)] \rangle}_{I_2}.$$

Now,

$$\| \bar{H}(O, \theta, \gamma^1) - \bar{H}(O, \theta, \gamma^2) \|$$
$$= \| (c(s, a, s^{'}, \gamma^1) - c(s, a, s^{'}, \gamma^2) - L(\theta, \gamma^1) + L(\theta, \gamma^2) + (f_{s'}{}^T - f_s{}^T)(v^*(\theta, \gamma^1) - v^*(\theta, \gamma^2))) \nabla \log \pi_\theta(a|s) \|$$
$$\le D(|c(s, a, s^{'}, \gamma^1) - c(s, a, s^{'}, \gamma^2)| + |L(\theta, \gamma^1) - L(\theta, \gamma^2)| + 2 \| v^*(\theta, \gamma^1) - v^*(\theta, \gamma^2) \|)$$
$$\le D(2N(U_c + U_\alpha)|\gamma_m^1 - \gamma_m^2| + 2L_2|\gamma_m^1 - \gamma_m^2|),$$

where $|\gamma_m^1 - \gamma_m^2| = \max\limits_{i=1,2,3....,N} |\gamma_i^1 - \gamma_i^2|$. In the above, $U_c$ and $U_\alpha$ are as before, see Section A. Further, from Assumption 1, $|f_s| \le 1$ and hence $|f_{s'} - f_s| \le 2$. Thus, for term $I_1$, note that

$$I_1 \le 4D(N(U_c + U_\alpha) + L_2) \| \nabla L(\theta, \gamma^1) \| |\gamma_m^1 - \gamma_m^2|.$$

Further, for term $I_2$, we have

$$I_2 \le \| \nabla L(\theta, \gamma^1) - \nabla L(\theta, \gamma^2) \| \| \bar{H}(O, \theta, \gamma^2) - E_{O'}[\bar{H}(O^{'}, \theta, \gamma^2)] \|$$
$$\le 4D(U_r + U_v) \| \nabla L(\theta, \gamma^1) - \nabla L(\theta, \gamma^2) \|$$
$$\le 4DC(U_r + U_v)|\gamma_m^1 - \gamma_m^2|.$$

The last inequality above is because of Lemma 2. After collecting the two parts, we now have,

$$|Q(O, \theta, \gamma^1) - Q(O, \theta, \gamma^2)| \le T_1 |\gamma_m^1 - \gamma_m^2|,$$

where $T_1 = 4D(N(U_c + U_\alpha) + L_2)U_L + 4DC(U_r + U_v)$.

**lemma 4.2** *For any $t \ge 0, \theta_1, \theta_2 \in R^d, \gamma = (\gamma_1, \gamma_2, ..., \gamma_N)^T$ with $0 \le \gamma_i \le M$ where $i \in \{1, 2, .., N\}$*

$$|Q(O, \theta_1, \gamma) - Q(O, \theta_2, \gamma)| \le T_2 \| \theta_1 - \theta_2 \|,$$

*for some $T_2 > 0$.*

*Proof* Recall that $O_t = (s_t, a_t, s_{t+1})$, hence for any $\theta_1, \theta_2, \gamma$,

$$Q(O, \theta_1, \gamma) - Q(O, \theta_2, \gamma)$$
$$= \langle \nabla L(\theta_1, \gamma), \bar{H}(O, \theta_1, \gamma) - E_{O'}[\bar{H}(O^{'}, \theta_1, \gamma)] \rangle - \langle \nabla L(\theta_2, \gamma), \bar{H}(O, \theta_2, \gamma) - E_{O'}[\bar{H}(O^{'}, \theta_2, \gamma)] \rangle$$
$$= \langle \nabla L(\theta_1, \gamma), \bar{H}(O, \theta_1, \gamma) - E_{O'}[\bar{H}(O^{'}, \theta_1, \gamma)] \rangle - \langle \nabla L(\theta_1, \gamma), \bar{H}(O, \theta_2, \gamma) - E_{O'}[\bar{H}(O^{'}, \theta_2, \gamma)] \rangle$$
$$+ \langle \nabla L(\theta_1, \gamma), \bar{H}(O, \theta_2, \gamma) - E_{O'}[\bar{H}(O^{'}, \theta_2, \gamma)] \rangle - \langle \nabla L(\theta_2, \gamma), \bar{H}(O, \theta_2, \gamma) - E_{O'}[\bar{H}(O^{'}, \theta_2, \gamma)] \rangle$$
$$= \underbrace{\langle \nabla L(\theta_1, \gamma), \bar{H}(O, \theta_1, \gamma) - \bar{H}(O, \theta_2, \gamma) - E_{O'}[\bar{H}(O^{'}, \theta_1, \gamma)] + E_{O'}[\bar{H}(O^{'}, \theta_2, \gamma)] \rangle}_{I_1}$$
$$+ \underbrace{\langle \nabla L(\theta_1, \gamma) - \nabla L(\theta_2, \gamma), \bar{H}(O, \theta_2, \gamma) - E_{O'}[\bar{H}(O^{'}, \theta_2, \gamma)] \rangle}_{I_2}.$$

Now,

$$\|\bar{H}(O, \theta_1, \gamma) - \bar{H}(O, \theta_2, \gamma)\|$$

$$= \|(c(s, a, s', \gamma) - L(\theta_1, \gamma) + (f_{s'}^T - f_s^T)v^*(\theta_1, \gamma))\nabla \log \pi_{\theta_1}(a|s)$$
$$- (c(s, a, s', \gamma) - L(\theta_2, \gamma) + (f_{s'}^T - f_s^T)v^*(\theta_2, \gamma))\nabla \log \pi_{\theta_2}(a|s)\|$$
$$\leq \|(c(s, a, s', \gamma) - L(\theta_1, \gamma) + (f_{s'}^T - f_s^T)v^*(\theta_1, \gamma))(\nabla \log \pi_{\theta_1}(a|s) - \nabla \log \pi_{\theta_2}(a|s))\|$$
$$+ \|(L(\theta_2, \gamma) - L(\theta_1, \gamma) + (f_{s'}^T - f_s^T)(v^*(\theta_1, \gamma) - v^*(\theta_1, \gamma)))\nabla \log \pi_{\theta_2}(a|s)\|$$
$$\leq 2(U_r + U_v)M_m\|\theta_1 - \theta_2\| + D(|L(\theta_2, \gamma) - L(\theta_1, \gamma)| + 2L_1\|\theta_1 - \theta_2\|)$$

Clearly,

$$|L(\theta_1, \gamma) - L(\theta_2, \gamma)|$$

$$= |\sum_{s \in S} \mu_{\theta_1}(s) \sum_{a \in A(s)} \pi_{\theta_1}(s, a)(d(s, a) + \sum_{k=1}^N \gamma(k)(h_k(s, a) - \alpha_k)) - \sum_{s \in S} \mu_{\theta_2}(s) \sum_{a \in A(s)} \pi_{\theta_2}(s, a)(d(s, a) + \sum_{k=1}^N \gamma(k)(h_k(s, a) - \alpha_k))|$$

$$\leq 2U_r d_{TV}(\mu_{\theta_1} \otimes \pi_{\theta_1}, \mu_{\theta_2} \otimes \pi_{\theta_2})$$

$$\leq 2U_r|A|L\left(1 + \lceil \log_k b^{-1} \rceil + 1/(1-k)\right)\|\theta_1 - \theta_2\|$$

$$= C_L\|\theta_1 - \theta_2\|.$$

The second inequality is from lemma B.1 of Wu et al., [2020]. Hence,

$$\|\bar{H}(O, \theta_1, \gamma) - \bar{H}(O, \theta_2, \gamma)\| \leq 2(U_r + U_v)M_m\|\theta_1 - \theta_2\| + DC_L\|\theta_1 - \theta_2\| + 2L_1D\|\theta_1 - \theta_2\|$$
$$= A_1\|\theta_1 - \theta_2\|$$

Now,

$$\|E_{O'}[\bar{H}(O', \theta_1, \gamma)] - E_{O'}[\bar{H}(O', \theta_2, \gamma)]\|$$

$$= \|E_{\theta_1}[\bar{H}(O', \theta_1, \gamma)] - E_{\theta_2}[\bar{H}(O', \theta_2, \gamma)]\|$$
$$\leq \|E_{\theta_1}[\bar{H}(O', \theta_1, \gamma)] - E_{\theta_1}[\bar{H}(O', \theta_2, \gamma)]\| + \|E_{\theta_1}[\bar{H}(O', \theta_2, \gamma)] - E_{\theta_2}[\bar{H}(O', \theta_2, \gamma)]\|$$
$$\leq E_{\theta_1}\|\bar{H}(O', \theta_1, \gamma) - \bar{H}(O', \theta_2, \gamma)\| + 4D(U_r + U_v)d_{TV}(\mu_{\theta_1} \otimes \pi_{\theta_1}, \mu_{\theta_2} \otimes \pi_{\theta_2})$$
$$\leq \left[2(U_r + U_v)M_m + DC_L + 2L_1D + 4D(U_r + U_v)|A|U_rL\left(1 + \lceil \log_k b^{-1} \rceil + \frac{1}{1-k}\right)\right]\|\theta_1 - \theta_2\|$$
$$= A_2\|\theta_1 - \theta_2\|$$

Thus, we have,

$$I_1 \leq U_L(A_1 + A_2)\|\theta_1 - \theta_2\|$$

For term $I_2$, we have,

$$I_2 \leq \|\nabla L(\theta_1, \gamma) - \nabla L(\theta_2, \gamma)\|\|\bar{H}(O, \theta_2, \gamma) - E_{O'}[\bar{H}(O', \theta_2, \gamma)]\|$$
$$\leq 4D(U_r + U_v)M_L\|\theta_1 - \theta_2\|.$$

The last inequality follows from Lemma 1. Thus,

$$Q(O, \theta_1, \gamma) - Q(O, \theta_2, \gamma) \leq T_2\|\theta_1 - \theta_2\|,$$

where

$$T_2 = (A_1 + A_2)U_L + 4D(U_r + U_v)M_L$$
$$A_1 = 2(U_r + U_v)M_m + DC_L + 2L_1D$$
$$A_2 = 2(U_r + U_v)M_m + DC_L + 2L_1D + 4D(U_r + U_v)|A|U_rL\left(1 + \lceil \log_k b^{-1} \rceil + \frac{1}{1-k}\right)$$

Here $E_\theta$ denotes that $O' = (s, a, s')$ has been sampled as $s \sim \mu_\theta, a \sim \pi_\theta, s' \sim p(s, ., a)$.

**lemma 4.3** *For any $t \geq 0$, we have*

$$|E[(Q(O_t, \theta_{t-\tau}, \gamma(t-\tau)) - Q(\tilde{O}_t, \theta_{t-\tau}, \gamma(t-\tau)))|\theta_{t-\tau}, \gamma(t-\tau), s_{t-\tau+1}]| \leq 2D(U_r + U_v)U_L|A|L \sum_{i=t-\tau}^{t} E\|\theta_i - \theta_{t-\tau}\|.$$

*Proof* From the definition of $Q(O, \theta, \gamma)$, we have that

$$E[(Q(O_t, \theta_{t-\tau}, \gamma(t-\tau)) - Q(\tilde{O}_t, \theta_{t-\tau}, \gamma(t-\tau))|\theta_{t-\tau}, \gamma(t-\tau), s_{t-\tau+1}]$$
$$= E[\langle \nabla L(\theta_{t-\tau}, \gamma(t-\tau)), \bar{H}(O_t, \theta_{t-\tau}, \gamma(t-\tau) - \bar{H}(\tilde{O}_t, \theta_{t-\tau}, \gamma(t-\tau)\rangle|\theta_{t-\tau}, \gamma(t-\tau), s_{t-\tau+1}]$$
$$= E[(\langle \nabla L(\theta_{t-\tau}, \gamma(t-\tau)), \bar{H}(O_t, \theta_{t-\tau}, \gamma(t-\tau)\rangle - \langle \nabla L(\theta_{t-\tau}, \gamma(t-\tau)), \bar{H}(\tilde{O}_t, \theta_{t-\tau}, \gamma(t-\tau))\rangle)|\theta_{t-\tau}, \gamma(t-\tau), s_{t-\tau+1}]$$
$$\leq 4D(U_r + U_v)U_L d_{TV}(P(O_t = .|s_{t-\tau+1}, \theta_{t-\tau}), (P(\tilde{O}_t = .|s_{t-\tau+1}, \theta_{t-\tau})).$$

Now,

$$d_{TV}(P(O_t = .|s_{t-\tau+1}, \theta_{t-\tau}), P(\tilde{O}_t = .|s_{t-\tau+1}, \theta_{t-\tau})) \leq \frac{1}{2}|A|L \sum_{i=t-\tau}^{t} E\|\theta_i - \theta_{t-\tau}\|.$$

The inequality above is similar to the one shown in the proof of Lemma D.2 of Wu et al., [2020]. Hence,

$$E[Q(O_t, \theta_{t-\tau}, \gamma(t-\tau)) - Q(\tilde{O}_t, \theta_{t-\tau}, \gamma(t-\tau))] \leq 2D(U_r + U_v)U_L|A|L \sum_{i=t-\tau}^{t} E\|\theta_i - \theta_{t-\tau}\|.$$

**lemma 4.4** *For any $t \geq 0$, we have*

$$|E[Q(\tilde{O}_t, \theta_{t-\tau}, \gamma(t-\tau)) - Q(O_t', \theta_{t-\tau}, \gamma(t-\tau))|\theta_{t-\tau}, \gamma(t-\tau), s_{t-\tau+1}]| \leq 4D(U_r + U_v)U_L bk^{\tau-1}.$$

*Proof* Note that

$$E[Q(\tilde{O}_t, \theta_{t-\tau}, \gamma(t-\tau)) - Q(O_t', \theta_{t-\tau}, \gamma(t-\tau))|\theta_{t-\tau}, \gamma(t-\tau), s_{t-\tau+1}]$$
$$= E[\langle \nabla L(\theta_{t-\tau}, \gamma(t-\tau)), \bar{H}(\tilde{O}_t, \theta_{t-\tau}, \gamma(t-\tau)) - \bar{H}(O_t', \theta_{t-\tau}, \gamma(t-\tau))\rangle|\theta_{t-\tau}, \gamma(t-\tau), s_{t-\tau+1}]$$
$$= E[(\langle \nabla L(\theta_{t-\tau}, \gamma(t-\tau)), \bar{H}(\tilde{O}_t, \theta_{t-\tau}, \gamma(t-\tau))\rangle - \langle \nabla L(\theta_{t-\tau}, \gamma(t-\tau)), \bar{H}(O_t', \theta_{t-\tau}, \gamma(t-\tau))\rangle)|\theta_{t-\tau}, \gamma(t-\tau), s_{t-\tau+1}]$$
$$\leq 4D(U_r + U_v)U_L d_{TV}(P(\tilde{O}_t = .|s_{t-\tau+1}, \theta_{t-\tau}), \mu_{\theta_{t-\tau}} \otimes \pi_{\theta_{t-\tau}} \otimes P)$$
$$\leq 4D(U_r + U_v)U_L bk^{\tau-1}$$

The last inequality comes from the proof of Lemma D.3 of Wu et al., [2020].

We now continue with the remaining proof of Lemma 4.

*Proof of Lemma 4 (Contd.)*
We decompose $E[Q(O_t, \theta_t, \gamma(t))]$ as follows:

$$E[Q(O_t, \theta_t, \gamma(t))] = E[Q(O_t, \theta_t, \gamma(t)) - Q(O_t, \theta_t, \gamma(t-\tau))] + E[Q(O_t, \theta_t, \gamma(t-\tau)) - Q(O_t, \theta_{t-\tau}, \gamma(t-\tau))]$$
$$+ E[Q(O_t, \theta_{t-\tau}, \gamma(t-\tau)) - Q(\tilde{O}_t, \theta_{t-\tau}, \gamma(t-\tau))] + E[Q(\tilde{O}_t, \theta_{t-\tau}, \gamma(t-\tau)) - Q(O_t', \theta_{t-\tau}, \gamma(t-\tau))]$$
$$+ E[Q(O_t', \theta_{t-\tau}, \gamma(t-\tau))]$$

where $\tilde{O}_t = (\tilde{s}_t, \tilde{a}_t, \tilde{s}_{t+1})$ is from the auxiliary Markov chain and $O_t' = (s_t, a_t, s_{t+1})$ is from the stationary distribution with $s_t \sim \mu_{\theta_{t-\tau}}, a_t \sim \pi_{\theta_{t-\tau}}, s_{t+1} \sim p(s_t, ., a_t)$ which actually satisfies $E[Q(O_t', \theta_{t-\tau}, \gamma(t-\tau))] = 0$. By collecting the

corresponding bounds from Lemmas 4.1–4.4, we have

$$E[Q(O_t, \theta_t, \gamma(t))] \geq -T_1 E|\gamma_m(t) - \gamma_m(t-\tau)| - T_2 E\|\theta_t - \theta_{t-\tau}\| - 2D(U_r + U_v)U_L|A|L \sum_{i=t-\tau}^{t} E\|\theta_i - \theta_{t-\tau}\|$$

$$- 4D(U_r + U_v)U_L bk^{\tau-1}$$

$$\geq -T_1 \sum_{i=t-\tau+1}^{t} E|\gamma_m(i) - \gamma_m(i-1)| - T_2 \sum_{i=t-\tau+1}^{t} E\|\theta_i - \theta_{i-1}\|$$

$$- 2D(U_r + U_v)U_L|A|L \sum_{i=t-\tau+1}^{t} \sum_{j=t-\tau+1}^{i} E\|\theta_j - \theta_{j-1}\| - 4D(U_r + U_v)U_L bk^{\tau-1}$$

$$\geq -T_1 \sum_{i=t-\tau+1}^{t} E|\gamma_m(i) - \gamma_m(i-1)| - T_2 \sum_{i=t-\tau+1}^{t} E\|\theta_i - \theta_{i-1}\|$$

$$- 2D(U_r + U_v)U_L|A|L\tau \sum_{j=t-\tau+1}^{t} E\|\theta_j - \theta_{j-1}\| - 4D(U_r + U_v)U_L bk^{\tau-1}$$

$$\geq -\left(D_1(\tau+1) \sum_{k=t-\tau+1}^{t} E\|\theta_k - \theta_{k-1}\| + D_2 bk^{\tau-1} + T_1 \sum_{i=t-\tau+1}^{t} E|\gamma_m(i) - \gamma_m(i-1)|\right),$$

where $D_1 := \max\{T_2, 2D(U_r + U_v)U_L|A|L\}$ and $D_2 := 4D(U_r + U_v)U_L$. This completes the proof. $\blacksquare$

*Proof of Theorem 1*

Under the update rule of Algorithm 1 for the actor recursion, we have:

$$\theta_{t+1} = \theta_t + b(t)\delta_t \Psi_{s_t a_t}.$$

So, using lemma 1, we have

$$L(\theta_{t+1}, \gamma(t)) \geq L(\theta_t, \gamma(t)) + b(t)\langle \nabla L(\theta_t, \gamma(t)), \delta_t \nabla \log \pi_{\theta_t}(a_t|s_t)\rangle - M_L b(t)^2 \|\delta_t \nabla \log \pi_{\theta_t}(a_t|s_t)\|^2.$$

Now

$$\delta_t \nabla \log \pi_{\theta_t}(a_t|s_t)$$

$$= (q(t) + \sum_{k=1}^{N} \gamma_k(t)(h_k(t) - \alpha_k) - L_t + v_t^T(f(s_{t+1}) - f(s_t))\nabla \log \pi_{\theta_t}(a_t|s_t)$$

$$= (q(t) + \sum_{k=1}^{k=N} \gamma_k(t)(h_k(t) - \alpha_k) - L(\theta_t, \gamma(t)) + L(\theta_t, \gamma(t)) - L_t + (f(s_{t+1})^T - f(s_t)^T)(v_t - v_t^*)$$

$$+ (f(s_{t+1})^T - f(s_t)^T)v_t^*)\nabla \log \pi_{\theta_t}(a_t|s_t)$$

$$= \Delta H(O_t, L_t, v_t, \theta_t, \gamma(t)) + H(O_t, \theta_t, \gamma(t), q(t), h(t)),$$

where $h(t) = (h_1(t), h_2(t), h_3(t), ....., h_N(t))$. Hence,

$$L(\theta_{t+1}, \gamma(t)) \geq L(\theta_t, \gamma(t)) + b(t)\langle \nabla L(\theta_t, \gamma(t)), \Delta H(O_t, L_t, v_t, \theta_t, \gamma(t)) + H(O_t, \theta_t, \gamma(t), q(t), h(t))\rangle$$

$$- M_L b(t)^2 \|\delta_t \nabla \log \pi_{\theta_t}(a_t|s_t)\|^2$$

$$\geq L(\theta_t, \gamma(t)) + b(t)\langle \nabla L(\theta_t, \gamma(t)), \Delta H(O_t, L_t, v_t, \theta_t, \gamma(t))\rangle$$

$$+ b(t)\langle \nabla L(\theta_t, \gamma(t)), H(O_t, \theta_t, \gamma(t), q(t), h(t)) - E_{O', q(t), h(t)}[H(O', \theta_t, \gamma(t), q(t), h(t))]\rangle$$

$$+ b(t)\langle \nabla L(\theta_t, \gamma(t)), E_{O', q(t), h(t)}[H(O', \theta_t, \gamma(t), q(t), h(t))]\rangle - M_L b(t)^2 \|\delta_t \nabla \log \pi_{\theta_t}(a_t|s_t)\|^2$$

$$\geq L(\theta_t, \gamma(t)) + b(t)\langle \nabla L(\theta_t, \gamma(t)), \Delta H(O_t, L_t, v_t, \theta_t, \gamma(t))\rangle$$

$$+ b(t)\breve{\Gamma}(O_t, \theta_t, \gamma(t), q(t), h(t)) + b(t)\|\nabla L(\theta_t, \gamma(t))\|^2 + b(t)\langle \nabla L(\theta_t, \gamma(t)), E_{O'}[\Delta H'(O', \theta_t, \gamma(t))]\rangle$$

$$- M_L b(t)^2 \|\delta_t \nabla \log \pi_{\theta_t}(a_t|s_t)\|^2.$$

In the first inequality, we discard the $1/2$ in front of the square norm term. After rearranging the terms, we obtain

$$\|\nabla L(\theta_t, \gamma(t))\|^2 \leq \frac{1}{b(t)}(L(\theta_{t+1}, \gamma(t)) - L(\theta_t, \gamma(t))) - \langle \nabla L(\theta_t, \gamma(t)), \Delta H(O_t, L_t, v_t, \theta_t, \gamma(t)) \rangle$$
$$- \check{\Gamma}(O_t, \theta_t, \gamma(t), q(t), h(t)) - \langle \nabla L(\theta_t, \gamma(t)), E_{O'}[\Delta H'(O', \theta_t, \gamma(t)] \rangle$$
$$+ M_L b(t) \|\delta_t \nabla \log \pi_{\theta_t}(a_t | s_t)\|^2.$$

After taking expectations we have,

$$E[\|\nabla L(\theta_t, \gamma(t))\|^2] \leq \frac{1}{b(t)} E[(L(\theta_{t+1}, \gamma(t)) - L(\theta_t, \gamma(t)))] - E[\langle \nabla L(\theta_t, \gamma(t)), \Delta H(O_t, L_t, v_t, \theta_t, \gamma(t)) \rangle]$$
$$- E[\check{\Gamma}(O_t, \theta_t, \gamma(t), q(t), h(t))] - E[\langle \nabla L(\theta_t, \gamma(t)), E_{O'}[\Delta H'(O', \theta_t, \gamma(t))] \rangle]$$
$$+ M_L b(t) E[\|\delta_t \nabla \log \pi_{\theta_t}(a_t | s_t)\|^2].$$
$$(17)$$

Now, observe that

$$E[\langle \nabla L(\theta_t, \gamma(t)), \Delta H(O_t, L_t, v_t, \theta_t, \gamma(t)) \rangle] \geq -D\sqrt{E\|\nabla L(\theta_t, \gamma(t))\|^2}\sqrt{2E\|A_t\|^2 + 8E\|B_t\|^2},$$

where

$$A_t = L_t - L(\theta_t, \gamma(t)),$$
$$B_t = v_t - v^*(\theta_t, \gamma(t)),$$

and the inequality follows from the Cauchy inequality and Lemma 3.

Next, we have that

$$E[\check{\Gamma}(O_t, \theta_t, \gamma(t), q(t), h(t))] \geq -\left(D_1(\tau+1)\sum_{k=t-\tau+1}^{t} E\|\theta_k - \theta_{k-1}\| + D_2 bk^{\tau-1} + T_1 \sum_{i=t-\tau+1}^{t} E|\gamma_m(i) - \gamma_m(i-1)|\right)$$
$$\geq -\left(2D_1 D(\tau+1)(U_r + U_v)\sum_{k=t-\tau+1}^{t-1} b(k) + D_2 bk^{\tau-1}\right.$$
$$+ T_1(U_\alpha + U_c)\sum_{k=t-\tau+1}^{t-1} c(k)\right)$$

Also, note that

$$\langle \nabla L(\theta_t, \gamma(t)), E_{O'}[\Delta H'(O', \theta_t, \gamma(t))] \rangle \geq -U_L\sqrt{\|E_{O'}[\Delta H'(O', \theta_t, \gamma(t))]\|^2}$$
$$\geq -U_L\sqrt{E_{O'}\|\Delta H'(O', \theta_t, \gamma(t))\|^2}$$
$$\geq -2DU_L\epsilon_{app}$$

Now we return to the inequality in (17) and plug the above terms back in it to obtain

$$E[\|\nabla L(\theta_t, \gamma(t))\|^2] \leq \frac{1}{b(t)} E[(L(\theta_{t+1}, \gamma(t)) - L(\theta_t, \gamma(t)))] + D\sqrt{E\|\nabla L(\theta_t, \gamma(t))\|^2}\sqrt{2E\|A_t\|^2 + 8E\|B_t\|^2}$$
$$+ 2D_1 D(\tau+1)(U_r + U_v)\sum_{k=t-\tau}^{t-1} b(k) + D_2 bk^{\tau-1}$$
$$+ T_1(U_\alpha + U_c)\sum_{k=t-\tau}^{t-1} c(k) + 2DU_L\epsilon_{app}$$
$$+ M_L b(t) E[\|\delta_t \nabla \log \pi_{\theta_t}(a_t | s_t)\|^2].$$

By setting $\tau = \tau_t$, we have,

$$E[\|\nabla L(\theta_t, \gamma(t))\|^2] \leq \frac{1}{b(t)} E[(L(\theta_{t+1}, \gamma(t)) - L(\theta_t, \gamma(t)))] + D\sqrt{E\|\nabla L(\theta_t, \gamma(t))\|^2}\sqrt{2E\|A_t\|^2 + 8E\|B_t\|^2}$$

$$+ 2D_1 D(\tau_t + 1)^2(U_r + U_v)b(t - \tau_t) + D_2 b(t) + 4M_L D^2(U_r + U_v)^2 b(t)$$

$$+ T_1(U_\alpha + U_c)(\tau_t + 1)c(t - \tau_t) + 2DU_L \epsilon_{app}$$

$$\leq \frac{1}{b(t)} E[(L(\theta_{t+1}, \gamma(t)) - L(\theta_t, \gamma(t)))] + D\sqrt{E\|\nabla L(\theta_t, \gamma(t))\|^2}\sqrt{2E\|A_t\|^2 + 8E\|B_t\|^2}$$

$$+ M_1(\tau_t + 1)^2 b(t - \tau_t) + M_2 b(t) + M_3(\tau_t + 1)c(t - \tau_t) + 2DU_L\epsilon_{\text{app}}.$$

Summing the expectation from $\tau_t$ to $t$ we have,

$$\sum_{k=\tau_t}^{t} E[\|\nabla L(\theta_k, \gamma(k))\|^2] \leq \underbrace{\sum_{k=\tau_t}^{t} \frac{1}{b(k)} E[(L(\theta_{k+1}, \gamma(k)) - L(\theta_k, \gamma(k)))]}_{I_1}$$

$$+ \sum_{k=\tau_t}^{t} D\sqrt{E\|\nabla L(\theta_k, \gamma(k))\|^2}\sqrt{2E\|A_k\|^2 + 8E\|B_k\|^2}$$

$$+ \underbrace{\sum_{k=\tau_t}^{t} (M_1(\tau_t + 1)^2 b(k - \tau_t) + M_2 b(k) + M_3(\tau_t + 1)c(k - \tau_t))}_{I_2}$$

$$+ 2DU_L\epsilon_{\text{app}}(t - \tau_t + 1)$$

For the term $I_1$ above,

$$\sum_{k=\tau_t}^{t} \frac{1}{b(k)} E[(L(\theta_{k+1}, \gamma(k)) - L(\theta_k, \gamma(k)))]$$

$$= \sum_{k=\tau_t}^{t} \frac{1}{b(k)} E[(L(\theta_{k+1}, \gamma(k)) - L(\theta_{k+1}, \gamma(k+1))) + (L(\theta_{k+1}, \gamma(k+1)) - L(\theta_k, \gamma(k)))]$$

$$\leq \sum_{k=\tau_t}^{t} \frac{1}{b(k)}(U_c + U_\alpha) E[\sum_{m=1}^{N} |\gamma_m(k) - \gamma_m(k+1)|]$$

$$+ \sum_{k=\tau_t}^{t}\left(\frac{1}{b(k-1)} - \frac{1}{b(k)}\right)E[L(\theta_k, \gamma(k))] - \frac{1}{b(\tau_t - 1)}E[L(\theta_{\tau_t}, \gamma(\tau_t))] + \frac{1}{b(t)}E[L(\theta_{t+1}.\gamma(t+1))]$$

$$\leq N(U_c + U_\alpha)^2 \sum_{k=\tau_t}^{t} \frac{c(k)}{b(k)} + \sum_{k=\tau_t}^{t}\left(\frac{1}{b(k-1)} - \frac{1}{b(k)}\right)U_r + \frac{1}{b(\tau_t - 1)}U_r + \frac{1}{b(t)}U_r$$

$$= N(U_c + U_\alpha)^2 \sum_{k=\tau_t}^{t} \frac{c(k)}{b(k)} + U_r\left[\sum_{k=\tau_t}^{t}\left(\frac{1}{b(k-1)} - \frac{1}{b(k)}\right) + \frac{1}{b(\tau_t - 1)} + \frac{1}{b(t)}\right]$$

$$= N(U_c + U_\alpha)^2 \frac{c_c}{c_b} \sum_{k=\tau_t}^{t} (1+k)^{\sigma-\beta} + 2U_r b(t)^{-1}$$

$$\leq \frac{N(U_c + U_\alpha)^2 c_c}{c_b(1 + \beta - \sigma)}(t - \tau_t + 1)^{1-\beta+\sigma} + 2\frac{U_r}{c_b}(1+t)^{\sigma}$$

$$= B_1(t - \tau_t + 1)^{1-\beta+\sigma} + B_2(1+t)^{\sigma}.$$

The first inequality above holds because

$$L(\theta_{k+1}, \gamma(k)) - L(\theta_{k+1}, \gamma(k+1)) = \sum_{s \in S} \mu_{\theta_{k+1}}(s) \sum_{a \in A(s)} \pi_{\theta_{k+1}}(a|s)(\sum_{m=1}^{N}(\gamma_m(k) - \gamma_m(k+1))(h_k(s,a) - \alpha_k))$$

$$\leq (U_c + U_\alpha) \sum_{m=1}^{N}|\gamma_m(k) - \gamma_m(k+1)|.$$

Now, for the term $I_2$,

$$\sum_{k=\tau_t}^{t}(M_1(\tau_t + 1)^2 b(k - \tau_t) + M_2 b(k) + M_3(\tau_t + 1)c(k - \tau_t))$$

$$\leq (M_1(\tau_t + 1)^2 + M_2) \sum_{k=0}^{t-\tau_t} b(k) + M_3(\tau_t + 1)\sum_{k=0}^{t-\tau_t} c(k)$$

$$= (M_1(\tau_t + 1)^2 + M_2)c_b \sum_{k=0}^{t-\tau_t}(1 + k)^{-\sigma} + M_3(\tau_t + 1)c_c \sum_{k=0}^{t-\tau_t}(1 + k)^{-\beta}$$

$$\leq \frac{(M_1(\tau_t + 1)^2 + M_2)c_b}{1 - \sigma}(t - \tau_t + 1)^{1-\sigma} + \frac{M_3(\tau_t + 1)c_c}{1 - \beta}(t - \tau_t + 1)^{1-\beta}$$

$$\leq \left(\frac{(M_1(\tau_t + 1)^2 + M_2)c_b}{1 - \sigma} + \frac{M_3(\tau_t + 1)^2 c_c}{1 - \beta}\right)(t - \tau_t + 1)^{1-\sigma}$$

$$= B_3(\tau_t + 1)^2(t - \tau_t + 1)^{1-\sigma}.$$

The second inequality holds because

$$\sum_{k=0}^{t-\tau_t}(1 + k)^{-\sigma} \leq \int_0^{t-\tau_t+1} x^{-\sigma} dx = \frac{(t - \tau_t + 1)^{1-\sigma}}{(1 - \sigma)}.$$

After combining all the terms, we have

$$\sum_{k=\tau_t}^{t} E[\|\nabla L(\theta_k, \gamma(k))\|^2] \leq B_1(t - \tau_t + 1)^{1-\beta+\sigma} + B_2(1 + t)^\sigma + B_3(\tau_t + 1)^2(t - \tau_t + 1)^{1-\sigma}$$

$$+ D\sqrt{\sum_{k=\tau_t}^{t} E\|\nabla L(\theta_k, \gamma(k))\|^2} \sqrt{2\sum_{k=\tau_t}^{t} E\|A_k\|^2 + 8\sum_{k=\tau_t}^{t} E\|B_k\|^2}$$

$$+ 2DU_L\epsilon_{\text{app}}(t - \tau_t + 1).$$

Dividing $(1 + t - \tau_t)$ on both sides and assuming $t \geq 2\tau_t - 1$, we can express the result as

$$1/(1 + t - \tau_t)\sum_{k=\tau_t}^{t} E[\|\nabla L(\theta_k, \gamma(k))\|^2] \leq B_1(t - \tau_t + 1)^{\sigma-\beta} + 2B_2(1 + t)^{\sigma-1} + B_3(\tau_t + 1)^2(t - \tau_t + 1)^{-\sigma}$$

$$+ D\sqrt{\frac{1}{1 + t - \tau_t}\sum_{k=\tau_t}^{t} E\|\nabla L(\theta_k, \gamma(k))\|^2}\sqrt{Z(t)} + 2DU_L\epsilon_{app},$$

where,

$$Z(t) = (2\sum_{k=\tau_t}^{t} E\|A_k\|^2 + 8\sum_{k=\tau_t}^{t} E\|B_k\|^2)/(1 + t - \tau_t).$$

Let

$$F(t) = 1/(1 + t - \tau_t) \sum_{k=\tau_t}^{t} E[\|\nabla L(\theta_k, \gamma(k))\|^2].$$

So, we have

$$F(t) \le \mathcal{O}(t^{\sigma-\beta}) + \mathcal{O}((\log t)^2 t^{-\sigma}) + \mathcal{O}(\epsilon_{app}) + 2D\sqrt{F(t)}\sqrt{Z(t)},$$

because $\tau_t = \mathcal{O}(\log t)$, which gives

$$(\sqrt{F(t)} - D\sqrt{Z(t)})^2 \le \mathcal{O}(t^{\sigma-\beta}) + \mathcal{O}((\log t)^2 t^{-\sigma}) + \mathcal{O}(\epsilon_{app}) + D^2 Z(t).$$

Let

$$A(t) = \mathcal{O}(t^{\sigma-\beta}) + \mathcal{O}((\log t)^2 t^{-\sigma}) + \mathcal{O}(\epsilon_{app}).$$

Thus, we have

$$(\sqrt{F(t)} - D\sqrt{Z(t)})^2 \le A(t) + D^2 Z(t)$$
$$\Rightarrow \sqrt{F(t)} - D\sqrt{Z(t)} \le \sqrt{A(t)} + D\sqrt{Z(t)}$$
$$\Rightarrow \sqrt{F(t)} \le \sqrt{A(t)} + 2D\sqrt{Z(t)}$$
$$\Rightarrow F(t) \le 2A(t) + 8D^2 Z(t).$$

The first and third implications hold because for a function $M(t) \le Q(t) + R(t)$(with each positive), we have,

$$\sqrt{M(t)} \le \sqrt{Q(t)} + \sqrt{R(t)}$$
$$M(t)^2 \le 2Q(t)^2 + 2R(t)^2$$

So finally we have the following:

$$\min_{0 \le k \le t} E[\|\nabla L(\theta_k, \gamma(k))\|^2] \le 1/(1 + t - \tau_t) \sum_{k=\tau_t}^{t} E[\|\nabla L(\theta_k, \gamma(k))\|^2]$$
$$= \mathcal{O}(t^{\sigma-\beta})) + \mathcal{O}((\log t)^2 t^{-\sigma}) + \mathcal{O}(\epsilon_{app}) + \mathcal{O}(Z(t)).$$

## B.2  PROOF OF THEOREM 2: ESTIMATING THE AVERAGE REWARD FOR CONSTRAINED ACTOR CRITIC

We define several notations to clarify the probabilistic dependency below.

$$O_t := (s_t, a_t, s_{t+1}),$$
$$O := (s, a, s'),$$
$$L_t^* := L(\theta_t, \gamma(t)),$$
$$y_t := L_t - L_t^*,$$
$$\hat{\Xi}(L_t, \theta_t, \gamma(t), q(t), h(t)) := y_t \left( q(t) + \sum_{k=1}^{N} \gamma_k(t)(h_k(t) - \alpha_k) - L_t^* \right).$$

Before we proceed further, we first state and prove Lemmas 5 and 6 below that will be used in the proof of Theorem 2. Moreover, the proof of Lemma 6 shall rely on Lemmas 6.1–6.5 that we also state and prove in the following. Finally, collecting all these results together, we shall obtain the claim for Theorem 2.

**Lemma 5** *For any $\theta_1, \theta_2, \gamma^1 = (\gamma_1^1, \gamma_2^1, ....., \gamma_N^1)^T, \gamma^2 = (\gamma_1^2, \gamma_2^2, ....., \gamma_N^2)^T$ with $0 \leq \gamma_j^i \leq M$, we have*

$$|L(\theta_1, \gamma_1) - L(\theta_2, \gamma_2)| \leq C_1 \|\theta_1 - \theta_2\| + C_2 |\gamma_1^p - \gamma_2^p|,$$

*where $C_1 = N(U_c + U_\alpha), C_2 = 2U_r|A|L(1 + \lceil \log_k b^{-1} \rceil + \frac{1}{1-k})$ and $|\gamma_p^1 - \gamma_p^2| = \max\limits_{i=1,2,...,N} |\gamma_i^1 - \gamma_i^2|$.*

**Proof** Note that

$$|L(\theta_1, \gamma^1) - L(\theta_2, \gamma^2)| \leq |L(\theta_1, \gamma^1) - L(\theta_1, \gamma^2)| + |L(\theta_1, \gamma^2) - L(\theta_2, \gamma^2)|$$

$$\leq \sum_{s \in S} \mu_{\theta_1}(s) \sum_{a \in A(s)} \pi_{\theta_1}(s, a) \sum_{k=1}^{N} |\gamma(k)^1 - \gamma(k)^2| |(g_k(s, a) - \alpha_k)|$$

$$+ |\sum_{s \in S} \mu_{\theta_1}(s) \sum_{a \in A(s)} \pi_{\theta_1}(s, a)(d(s, a) + \sum_{k=1}^{N} \gamma(k)^2(h_k(s, a) - \alpha_k))$$

$$- \sum_{s \in S} \mu_{\theta_2}(s) \sum_{a \in A(s)} \pi_{\theta_2}(s, a)(d(s, a) + \sum_{k=1}^{N} \gamma(k)^2(h_k(s, a) - \alpha_k))|$$

$$\leq N(U_c + U_\alpha)|\gamma_p^1 - \gamma_p^2| + 2U_r|A|L(1 + \lceil \log_k b^{-1} \rceil + \frac{1}{1-k})\|\theta_1 - \theta_2\|$$

$$\leq C_1 |\gamma_p^1 - \gamma_p^2| + C_2 \|\theta_1 - \theta_2\|,$$

where $C_1 = N(U_c + U_\alpha), C_2 = 2U_r|A|L(1 + \lceil \log_k b^{-1} \rceil + \frac{1}{1-k})$ and $|\gamma_p^1 - \gamma_p^2| = \max\limits_{i=1,2,...,N} |\gamma_i^1 - \gamma_i^2|$.

The third inequality is because of Lemma B.1 of Wu et al., [2020]. ∎

**Lemma 6** *Given the definition of $\hat{\Xi}(L_t, \theta_t, \gamma(t), q(t), h(t))$, for any $t > 0$, we have*

$$E[\hat{\Xi}(L_t, \theta_t, \gamma(t), q(t), h(t))] \leq 6U_r N(U_c + U_\alpha) E|\gamma_p(t) - \gamma_p(t - \tau)| + 8U_r \overline{C} E\|\theta_t - \theta_{t-\tau}\| + 2U_r E|L_t - L_{t-\tau}|$$

$$+ 2U_r^2 |A|L \sum_{i=t-\tau}^{t} E\|\theta_i - \theta_{t-\tau}\| + 4U_r^2 bk^{\tau-1},$$

*where*

$$|\gamma_p(t) - \gamma_p(t - \tau)| = \max\limits_{i=1,2,...,N} |\gamma_i(t) - \gamma_i(t - \tau)|,$$

$$\overline{C} = U_r|A|L(1 + \lceil \log_k b^{-1} \rceil + \frac{1}{1-k}),$$

$$t \geq \tau \geq 0.$$

**Proof** We have

$$E[\hat{\Xi}(L_t, \theta_t, \gamma(t), q(t), h(t))] = E_{s_t \sim p, a_t \sim \pi_{\theta_t}, s_{t+1} \sim p}[E[y_t(q(t) + \sum_{k=1}^{k=N} \gamma_k(t)(h_k(t) - \alpha_k) - L_t^*)|s_t, a_t, s_{t+1}]]$$

$$= E[y_t(c(s_t, a_t, s_{t+1}, \gamma(t)) - L_t^*)]$$

$$= E[\Xi(O_t, L_t, \theta_t, \gamma(t))]$$

where

$$c(s, a, s^{'}, \gamma) = \sum_{q}(q \cdot \bar{p}(q|s, a, s^{'})) + \sum_{k=1}^{k=N} \gamma_k(\sum_{h}(h \cdot p_k(h|s, a, s^{'})) - \alpha_k).$$

The proof will be built on supporting lemmas 6.1–6.5 that we first state and prove below.

**lemma 6.1** *For any* $\theta, L, \gamma^1 = (\gamma_1^1, \gamma_2^1, ...., \gamma_N^1)^T, \gamma^2 = (\gamma_1^2, \gamma_2^2, ...., \gamma_N^2)^T, O = (s, a, s^{'})$ *with* $0 \leq \gamma_j^i \leq M$ *for* $i \in \{1, 2\}$ *and* $j \in \{1, 2, ..., N\}$, *we have*

$$|\Xi(O, L, \theta, \gamma^1) - \Xi(O, L, \theta, \gamma^2)| \leq 6U_r N(U_c + U_\alpha)|\gamma_p^1 - \gamma_p^2|,$$

*where*

$$|\gamma_p^1 - \gamma_p^2| = \max_{i=1,2,...,N} |\gamma_i^1 - \gamma_i^2|.$$

*Proof* We have,

$$|\Xi(O, L, \theta, \gamma^1) - \Xi(O, L, \theta, \gamma^2)| = |(L - L(\theta, \gamma^1))(c(s, a, s^{'}, \gamma^1) - L(\theta, \gamma^1)) - (L - L(\theta, \gamma^2))(c(s, a, s^{'}, \gamma^2) - L(\theta, \gamma^2))|$$
$$\leq |(L - L(\theta, \gamma^1))(c(s, a, s^{'}, \gamma^1) - c(s, a, s^{'}, \gamma^2) + L(\theta, \gamma^2) - L(\theta, \gamma^1))|$$
$$+ |(L(\theta, \gamma^2) - L(\theta, \gamma^1))(c(s, a, s^{'}, \gamma^2) - L(\theta, \gamma^2))|$$
$$\leq 2U_r(|c(s, a, s^{'}, \gamma^1) - c(s, a, s^{'}, \gamma^2)| + 2|L(\theta, \gamma^2) - L(\theta, \gamma^1)|)$$
$$\leq 6U_r N(U_c + U_\alpha)|\gamma_p^1 - \gamma_p^2|,$$

where

$$|\gamma_p^1 - \gamma_p^2| = \max_{i=1,2,...,N} |\gamma_i^1 - \gamma_i^2|.$$

**lemma 6.2** *For any* $L, \theta_1, \theta_2, O = (s, a, s^{'}), \gamma = (\gamma_1, \gamma_2, ..., \gamma_N)^T$ *with* $0 \leq \gamma_i \leq M$ *for* $i \in 1, 2, .., N$, *we have*

$$|\Xi(O, L, \theta_1, \gamma) - \Xi(O, L, \theta_2, \gamma)| \leq 8U_r\overline{C}\|\theta_1 - \theta_2\|,$$

*where*

$$\overline{C} = U_r|A|L(1 + \lceil \log_k b^{-1} \rceil + \frac{1}{1 - k}).$$

*Proof* By definition of $\Xi(O, L, \theta, \gamma)$, we have

$$|\Xi(O, L, \theta_1, \gamma) - \Xi(O, L, \theta_2, \gamma)| = |(L - L(\theta_1, \gamma))(C(s, a, s^{'}, \gamma) - L(\theta_1, \gamma)) - (L - L(\theta_2, \gamma))(C(s, a, s^{'}, \gamma) - L(\theta_2, \gamma))|$$
$$\leq |(L - L(\theta_1, \gamma))(C(s, a, s^{'}, \gamma) - L(\theta_1, \gamma)) - (L - L(\theta_1, \gamma))(C(s, a, s^{'}, \gamma) - L(\theta_2, \gamma))|$$
$$+ |(L - L(\theta_1, \gamma))(C(s, a, s^{'}, \gamma) - L(\theta_2, \gamma)) - (L - L(\theta_2, \gamma))(C(s, a, s^{'}, \gamma) - L(\theta_2, \gamma))|$$
$$\leq 4U_r|L(\theta_1, \gamma) - L(\theta_2, \gamma)|$$
$$\leq 8U_r\overline{C}\|\theta_1 - \theta_2\|,$$

where $\overline{C} = U_r|A|L(1 + \lceil \log_k b^{-1} \rceil + \frac{1}{1-k})$.

**lemma 6.3** *For any* $L_1, L_2, \theta, O = (s, a, s^{'}), \gamma = (\gamma_1, \gamma_2, ..., \gamma_N)^T$ *with* $0 \leq \gamma_i \leq M$ *for* $i \in 1, 2, .., N$, *we have*

$$|\Xi(O, L_1, \theta, \gamma) - \Xi(O, L_2, \theta, \gamma)| \leq 2U_r|L_1 - L_2|.$$

*Proof* By definition,

$$|\Xi(O, L_1, \theta, \gamma) - \Xi(O, L_2, \theta, \gamma)| = |(L_1 - L(\theta, \gamma))(C(s, a, s^{'}, \gamma) - L(\theta, \gamma)) - (L_2 - L(\theta, \gamma))(C(s, a, s^{'}, \gamma) - L(\theta, \gamma))|$$
$$\leq 2U_r|L_1 - L_2|.$$

The claim follows.

**lemma 6.4** *Consider the tuples $O_t = (s_t, a_t, s_{t+1})$ and $\tilde{O}_t = (\tilde{s}_t, \tilde{a}_t, \tilde{s}_{t+1})$ of the original and auxiliary Markov chains respectively. Then the following holds:*

$$|E[(\Xi(O_t, L_{t-\tau}, \theta_{t-\tau}, \gamma(t-\tau)) - \Xi(\tilde{O}_t, L_{t-\tau}, \theta_{t-\tau}, \gamma(t-\tau)))|L_{t-\tau}, \theta_{t-\tau}, \gamma(t-\tau), s_{t-\tau+1}]|$$

$$\leq 2U_r^2 |A|L \sum_{i=t-\tau}^{t} E\|\theta_i - \theta_{t-\tau}\|$$

*Proof* By the Cauchy-Schwartz inequality and the definition of the total variation norm, we have

$$E[(\Xi(O_t, L_{t-\tau}, \theta_{t-\tau}, \gamma(t-\tau)) - \Xi(\tilde{O}_t, L_{t-\tau}, \theta_{t-\tau}, \gamma(t-\tau)))|L_{t-\tau}, \theta_{t-\tau}, \gamma(t-\tau), s_{t-\tau+1}]$$
$$= (L_{t-\tau} - L_{t-\tau}^*)E[(C(s_t, a_t, s_{t+1}\gamma(t-\tau)) - C(\tilde{s}_t, \tilde{a}_t, \tilde{s}_{t+1}, \gamma(t-\tau)))|L_{t-\tau}, \theta_{t-\tau}, \gamma(t-\tau), s_{t-\tau+1}].$$

Now,

$$E[(C(s_t, a_t, s_{t+1}\gamma(t-\tau)) - C(\tilde{s}_t, \tilde{a}_t, \tilde{s}_{t+1}, \gamma(t-\tau)))|L_{t-\tau}, \theta_{t-\tau}, \gamma(t-\tau), s_{t-\tau+1}]$$
$$\leq 2U_r d_{TV}(P(O_t = \cdot|s_{t-\tau+1}, \theta_{t-\tau}), (P(\tilde{O}_t = \cdot|s_{t-\tau+1}, \theta_{t-\tau})).$$

The following bound on the total variation norm has been shown in the proof of lemma D.2 of Wu et al., [2020]:

$$d_{TV}(P(O_t = \cdot|s_{t-\tau+1}, \theta_{t-\tau}), (P(\tilde{O}_t = \cdot|s_{t-\tau+1}, \theta_{t-\tau})) \leq \frac{1}{2}|A|L \sum_{i=t-\tau}^{t} E\|\theta_i - \theta_{t-\tau}\|.$$

Plugging this bound above we have,

$$|E[(\Xi(O_t, L_{t-\tau}, \theta_{t-\tau}, \gamma(t-\tau)) - \Xi(\tilde{O}_t, L_{t-\tau}, \theta_{t-\tau}, \gamma(t-\tau)))|L_{t-\tau}, \theta_{t-\tau}, \gamma(t-\tau), s_{t-\tau+1}]| \leq 2U_r^2|A|L \sum_{i=t-\tau}^{t} E\|\theta_i - \theta_{t-\tau}\|.$$

The claim follows.

**lemma 6.5** *Conditioned on $s_{t-\tau+1}, \theta_{t-\tau}, L_{t-\tau}, \gamma(t-\tau)$, we have*

$$E[\Xi(\tilde{O}_t, L_{t-\tau}, \theta_{t-\tau}, \gamma(t-\tau))|L_{t-\tau}, \theta_{t-\tau}, \gamma(t-\tau), s_{t-\tau+1}] \leq 4U_r^2 bk^{\tau-1}$$

*Proof* The proof follows in a similar manner as Lemma D.7 of Wu et al., [2020].

After collecting the corresponding results from lemmas 6.1–6.5, we have

$$E[\hat{\Xi}(L_t, \theta_t, \gamma(t), q(t), h(t))] \leq 6U_r N(U_c + U_\alpha)E|\gamma_p(t) - \gamma_p(t-\tau)| + 8U_r\overline{C}E\|\theta_t - \theta_{t-\tau}\| + 2U_r E|L_t - L_{t-\tau}|$$

$$+ 2U_r^2|A|L \sum_{i=t-\tau}^{t} E\|\theta_i - \theta_{t-\tau}\| + 4U_r^2 bk^{\tau-1}.$$

∎

*Proof of Theorem 2: Estimating the Average Reward for Constrained Actor critic*

We have the following update rule in the algorithm that we now analyze:

$$L_{t+1} = L_t + a(t)(q(t) + \sum_{k=1}^{N} \gamma_k(t)(h_k(t) - \alpha_k)) - L_t).$$

Unrolling the above, we obtain

$$y_{t+1}^2 = (L_{t+1} - L_{t+1}^*)^2$$

$$= \left( L_t + a(t) \left( q(t) + \sum_{k=1}^{k=N} \gamma_k(t)(h_k(t) - \alpha_k) - L_t \right) - L_{t+1}^* \right)^2$$

$$= \left( y_t + L_t^* - L_{t+1}^* + a(t) \left( q(t) + \sum_{k=1}^{k=N} \gamma_k(t)(h_k(t) - \alpha_k) - L_t \right) \right)^2$$

$$= y_t^2 + 2a(t)y_t(C_t - L_t) + 2y_t(L_t^* - L_{t+1}^*) + (L_t^* - L_{t+1}^* + a(t)(C_t - L_t))^2$$

$$\leq y_t^2 + 2a(t)y_t(C_t - L_t) + 2y_t(L_t^* - L_{t+1}^*) + 2(L_t^* - L_{t+1}^*)^2 + 2a(t)^2(C_t - L_t)^2$$

$$= y_t^2 - 2a(t)y_t^2 + 2a(t)y_t^2 + 2a(t)y_t(C_t - L_t) + 2y_t(L_t^* - L_{t+1}^*) + 2(L_t^* - L_{t+1}^*)^2 + 2a(t)^2(C_t - L_t)^2$$

$$= y_t^2 - 2a(t)y_t^2 + 2a(t)y_t(C_t - L_t + y_t) + 2y_t(L_t^* - L_{t+1}^*) + 2(L_t^* - L_{t+1}^*)^2 + 2a(t)^2(C_t - L_t)^2$$

$$= (1 - 2a(t))y_t^2 + 2a(t)y_t(C_t - L_t^*) + 2y_t(L_t^* - L_{t+1}^*) + 2(L_t^* - L_{t+1}^*)^2 + 2a(t)^2(C_t - L_t)^2,$$

where $C_t = q(t) + \sum_{k=1}^{N} \gamma_k(t)(h_k(t) - \alpha_k)$. The first inequality is due to $(x+y)^2 \leq 2x^2 + 2y^2$. Rearranging and summing from $\tau_t$ to $t$, we have

$$\sum_{k=\tau_t}^t E[y_k^2] \leq \underbrace{\sum_{k=\tau_t}^t \frac{1}{2a(k)} E(y_k^2 - y_{k+1}^2)}_{I_1} + \underbrace{\sum_{k=\tau_t}^t E[\hat{\Xi}(L_k, \theta_k, \gamma(k), q(k), h(k))]}_{I_2}$$

$$+ \underbrace{\sum_{k=\tau_t}^t \frac{1}{a(k)} E[y_k(L_k^* - L_{k+1}^*)]}_{I_3} + \underbrace{\sum_{k=\tau_t}^t \frac{1}{a(k)} E[(L_k^* - L_{k+1}^*)^2]}_{I_4}$$

$$+ \underbrace{\sum_{k=\tau_t}^t a(k) E[(C_k - L_k)^2]}_{I_5}.$$

We now consider $I_1, \ldots, I_5$ term by term. For $I_1$, we have

$$I_1 = \sum_{k=\tau_t}^t \frac{1}{2a(k)} E(y_k^2 - y_{k+1}^2)$$

$$= \sum_{k=\tau_t}^t \left( \frac{1}{2a(k)} - \frac{1}{2a(k-1)} \right) E[y_k^2] + \frac{1}{2a(\tau_t - 1)} E[y_{\tau_t}^2] - \frac{1}{2a(t)} E[y_{t+1}^2]$$

$$\leq \frac{2U_r^2}{a(t)}.$$

For $I_2$, from Lemma 6, we have

$$E[\hat{\Xi}(L_t, \theta_t, \gamma(t), q(t), h(t))] \leq 6U_r N(U_c + U_\alpha) E|\gamma_p(t) - \gamma_p(t-\tau)| + 8U_r \overline{C} E\|\theta_t - \theta_{t-\tau}\| + 2U_r E|L_t - L_{t-\tau}|$$

$$+ 2U_r^2 |A| L \sum_{i=t-\tau}^t E\|\theta_i - \theta_{t-\tau}\| + 4U_r^2 b k^{\tau-1}$$

$$\leq 16U_r \overline{C} D(U_r + U_v)\tau b(t-\tau) + 6U_r N(U_c + U_\alpha)\tau(U_c + U_\alpha)c(t-\tau) + 4U_r^2 \tau a(t-\tau)$$

$$+ 4DU_r^2(U_r + U_v)|A| L\tau(\tau+1)b(t-\tau) + 4U_r^2 b k^{\tau-1}$$

$$\leq B_1 \tau^2 b(t-\tau) + B_2 \tau a(t-\tau) + B_3 b k^{\tau-1} + B_4 \tau c(t-\tau).$$

By the choice of $\tau_t$, we then have

$$I_2 = \sum_{k=\tau_t}^{t} E[\hat{\Xi}(L_t, \theta_t, \gamma(t), q(t), h(t))]$$

$$\leq (B_1 \tau_t^2 + B_3) \sum_{k=\tau_t}^{t} b(k - \tau_t) + B_2 \tau_t \sum_{k=\tau_t}^{t} a(t - \tau_t) + B_4 \tau_t \sum_{k=\tau_t}^{t} c(t - \tau_t).$$

For $I_3$, we have

$$I_3 \leq \left( \sum_{k=\tau_t}^{t} E[y_k^2] \right)^{1/2} \left( \sum_{k=\tau_t}^{t} E \left[ \frac{(L_k^* - L_{k+1}^*)^2}{a(k)^2} \right] \right)^{1/2}$$

$$\leq \left( \sum_{k=\tau_t}^{t} E[y_k^2] \right)^{1/2} \left( \sum_{k=\tau_t}^{t} E \left[ \frac{(C_1 \|\theta_k - \theta_{k+1}\| + C_2 \|\gamma(k)^p - \gamma(k+1)^p\|)^2}{a(k)^2} \right] \right)^{1/2}$$

$$\leq \left( \sum_{k=\tau_t}^{t} E[y_k^2] \right)^{1/2} \left( \sum_{k=\tau_t}^{t} E \left[ \frac{(2C_1 D(U_r + U_v)b(k) + C_2(U_c + U_\alpha)c(k))^2}{a(k)^2} \right] \right)^{1/2}$$

$$\leq \left( \sum_{k=\tau_t}^{t} E[y_k^2] \right)^{1/2} \left( \overline{K}^2 \sum_{k=\tau_t}^{t} \frac{b(k)^2}{a(k)^2} \right)^{1/2},$$

where $\overline{K} = ((2C_1 D(U_r + U_v) + (U_c + U_\alpha))$.

For $I_4$, we have

$$I_4 = \sum_{k=\tau_t}^{t} \frac{1}{a(k)} E[(L_k^* - L_{k+1}^*)^2]$$

$$= \sum_{k=\tau_t}^{t} \frac{1}{a(k)} E[(L(\theta_k, \gamma(k)) - L(\theta_{k+1}, \gamma(k+1)))^2]$$

$$\leq \sum_{k=\tau_t}^{t} \frac{\overline{K}^2 b(k)^2}{a(k)}$$

$$= \mathcal{O}\left( \sum_{k=\tau_t}^{t} \frac{b(k)^2}{a(k)} \right).$$

For $I_5$, we have

$$I_5 = \sum_{k=\tau_t}^{t} a(k) E[(C_k - L_k)^2]$$

$$\leq \sum_{k=\tau_t}^{t} 4U_r^2 a(k)$$

$$= \mathcal{O}\left( \sum_{k=\tau_t}^{t} a(k) \right).$$

Next, after combining $I_1, \ldots, I_5$, using the uniform ergodicity requirement (Assumption 3), the definition of $\tau_t$ and the relation between step-size and mixing time in Equation (4) of the main paper, we obtain the following:

$$\sum_{k=\tau_t}^{t} E[y_k^2] \le \frac{2U_r^2}{c_a}(1+t)^\omega + (B_1\tau_t^2 + B_3)c_b \sum_{k=\tau_t}^{t}(1+k-\tau_t)^{-\sigma} + B_2 c_a \tau_t \sum_{k=\tau_t}^{t}(1+k-\tau_t)^{-\omega}$$

$$+ B_4 \tau_t c_c \sum_{k=\tau_t}^{t}(1+k-\tau_t)^{-\beta} + \overline{K}\frac{c_b}{c_a}(\sum_{k=\tau_t}^{t} E[y_k^2])^{1/2}(\sum_{k=\tau_t}^{t}(1+k)^{-2(\sigma-\omega)})^{1/2}$$

$$+ \overline{K}\frac{c_b^2}{c_a} \sum_{k=\tau_t}^{t}(1+k)^{\omega-2\sigma} + 4U_r^2 c_a \sum_{k=\tau_t}^{t}(1+k)^{-\omega}$$

$$\le \frac{2U_r^2}{c_a}(1+t)^\omega + ((B_1\tau_t^2 + B_3)c_b + B_2 c_a \tau_t + \overline{K}c_b^2 + 4U_r^2 c_a + B_4 \tau_t c_c)\sum_{k=\tau_t}^{t}(1+k-\tau_t)^{-\omega}$$

$$+ \overline{K}\frac{c_b}{c_a}(\sum_{k=\tau_t}^{t} E[y_k^2])^{1/2}(\sum_{k=\tau_t}^{t}(1+k)^{-2(\sigma-\omega)})^{1/2}$$

$$\le \frac{2U_r^2}{c_a}(1+t)^\omega + ((B_1\tau_t^2 + B_3)c_b + B_2 c_a \tau_t + \overline{K}c_b^2 + 4U_r^2 c_a + B_4 \tau_t c_c)\sum_{k=0}^{t-\tau_t}(1+k)^{-\omega}$$

$$+ \overline{K}\frac{c_b}{c_a}(\sum_{k=\tau_t}^{t} E[y_k^2])^{1/2}(\sum_{k=\tau_t}^{t}(1+k)^{-2(\sigma-\omega)})^{1/2}$$

$$\le \frac{2U_r^2}{c_a}(1+t)^\omega + ((B_1\tau_t^2 + B_3)c_b + B_2 c_a \tau_t + \overline{K}c_b^2 + 4U_r^2 c_a + B_4 \tau_t c_c)\frac{(t-\tau_t+1)^{1-\omega}}{1-\omega}$$

$$+ \overline{K}\frac{c_b}{c_a}(\sum_{k=\tau_t}^{t} E[y_k^2])^{1/2}(\frac{(1+t-\tau_t)^{1-2(\sigma-\omega)}}{1-2(\sigma-\omega)})^{1/2}$$

Note also that we have used above the precise form of the step-sizes as mentioned towards the end of Section 4.1 (main paper). After applying the squaring technique (as in proof of Theorem B.1), we have:

$$\sum_{k=\tau_t}^{t} E[y_k^2] \le \frac{4U_r^2}{c_a}(1+t)^\omega + 2((B_1\tau_t^2 + B_3)c_b + B_2 c_a \tau_t + \overline{K}c_b^2 + 4U_r^2 c_a + B_4 \tau_t c_c)\frac{(t-\tau_t+1)^{1-\omega}}{1-\omega}$$

$$+ 8\overline{K}^2 \frac{c_b^2}{c_a^2}\frac{(1+t-\tau_t)^{1-2(\sigma-\omega)}}{1-2(\sigma-\omega)}$$

$$= \mathcal{O}(t^\omega) + \mathcal{O}(\log^2 t \cdot t^{1-\omega}) + \mathcal{O}(t^{1-2(\sigma-\omega)}).$$

Dividing by $(1+t-\tau_t)$ and assuming $t \ge 2\tau_t - 1$, we have

$$\sum_{k=\tau_t}^{t} E[y_k^2]/(1+t-\tau_t) = \mathcal{O}(t^{\omega-1}) + \mathcal{O}(\log^2 t \cdot t^{-\omega}) + \mathcal{O}(t^{-2(\sigma-\omega)}).$$

## B.3 PROOF OF THEOREM 2: ESTIMATING THE CONVERGENCE POINT OF CRITIC FOR CONSTRAINED ACTOR CRITIC

We first describe the notations used here.

$$O_t := (s_t, a_t, s_{t+1}),$$
$$O := (s, a, s'),$$
$$v^*(t) := v^*(\theta_t, \gamma(t)),$$
$$L_t^* := L(\theta_t, \gamma(t)),$$
$$m_t := v_t - v^*(t),$$
$$y_t := L_t - L_t^*,$$
$$g(O, v, \theta, \gamma, q, h) := (q + \sum_{k=1}^{k=N} \gamma(k)(h_k - \alpha_k) - L(\theta, \gamma) + v^T(f'_s - f_s))f_s, \tag{18}$$
$$c(s, a, s', \gamma) := \sum_q (q \cdot \bar{p}(q|s, a, s')) + \sum_{k=1}^{k=N} \gamma(k)(\sum_h (h \cdot p_k(h|s, a, s')) - \alpha_k),$$
$$\bar{g}(v, \theta, \gamma) := E_{s \sim \mu_\theta, a \sim \pi_\theta, s' \sim p}[(c(s, a, s', \gamma) - L(\theta, \gamma) + (f_{s'} - f_s)^T v)f_s],$$
$$\Lambda(O, v, \theta, \gamma, q, h) := \langle v - v^*(\theta, \gamma), g(O, v, \theta, \gamma, q, h) - \bar{g}(v, \theta, \gamma) \rangle,$$
$$\Delta g(O, L, \theta, \gamma) := (L(\theta, \gamma) - L)f_s.$$

In the above, $h = (h_1, h_2, h_3, ..., h_N)$.

Before we proceed further, we first state and prove Lemma 7 below that will be used in the proof of Theorem 2(estimating the convergence point of critic) .

**Lemma 7** *From the definition of* $\Lambda(O_t, v_t, \theta_t, \gamma(t), q(t), h(t))$, *for any* $0 \leq \tau \leq t$, *we have*

$$E[\Lambda(O_t, v_t, \theta_t, \gamma(t), q(t), h(t))] \leq C_1(\tau + 1)E\|\theta_t - \theta_{t-\tau}\| + C_2 bk^{\tau-1} + C_3 E\|v_t - v(t - \tau)\|$$
$$+ C_4 E|\gamma_m(t) - \gamma_m(t - \tau)|,$$

*where* $C_1, C_2, C_3, C_4$ *are positive constants and* $|\gamma_m(t) - \gamma_m(t - \tau)| = \max_{i=1,2,...,N} |\gamma_i(t) - \gamma_i(t - \tau)|$.

**Proof** We have,

$$E[\Lambda(O_t, v_t, \theta_t, \gamma(t), q(t), h(t))] = E[\langle v_t - v^*(\theta_t, \gamma(t)), g(O_t, v_t, \theta_t, \gamma(t), q(t), h(t)) - \bar{g}(v_t, \theta_t, \gamma(t)) \rangle].$$

Note now that $v_t, \theta_t, \gamma(t)$ do not depend on $q(t)$ and $h(t)$. Hence we can write,

$$E[\langle v_t - v^*(\theta_t, \gamma(t)), g(O_t, v_t, \theta_t, \gamma(t), q(t), h(t)) - \bar{g}(v_t, \theta_t, \gamma(t)) \rangle]$$
$$= E_{s_t \sim p, a_t \sim \pi_{\theta_t}, s_{t+1} \sim p}[E[\langle v_t - v^*(\theta_t, \gamma(t)), g(O_t, v_t, \theta_t, \gamma(t), q(t), h(t)) - \bar{g}(v_t, \theta_t, \gamma(t)) \rangle | s_t, a_t, s_{t+1}]]$$
$$= E[\langle v_t - v^*(\theta_t, \gamma(t)), \check{g}(O_t, v_t, \theta_t, \gamma(t)) - \bar{g}(v_t, \theta_t, \gamma(t)) \rangle],$$

where,

$$\check{g}(O_t, v_t, \theta_t, \gamma(t)) = (c(s_t, a_t, s_{t+1}, \gamma(t)) - L(\theta, \gamma) + v^T(f'_s - f_s))f_s,$$
$$c(s_t, a_t, s_{t+1}, \gamma(t)) = \sum_q (q \cdot \bar{p}(q|s_t, a_t, s_{t+1})) + \sum_{k=1}^{k=N} \gamma_k(t)(\sum_h (h \cdot p_k(h|s_t, a_t, s_{t+1})) - \alpha_k).$$

Let $\langle v_t - v^*(\theta_t, \gamma(t)), \check{g}(O_t, v_t, \theta_t, \gamma(t)) - \bar{g}(v_t, \theta_t, \gamma(t)) \rangle = \bar{\Lambda}(O_t, v_t, \theta_t, \gamma(t))$. Note that we can decompose

$E[\bar{\Lambda}(O_t, v_t, \theta_t, \gamma(t))]$ as follows:

$$E[\bar{\Lambda}(O_t, v_t, \theta_t, \gamma(t))] = \underbrace{E[\bar{\Lambda}(O_t, v_t, \theta_t, \gamma(t)) - \bar{\Lambda}(O_t, v_t, \theta_t, \gamma(t-\tau))]}_{I_1} + \underbrace{E[\bar{\Lambda}(O_t, v_t, \theta_t, \gamma(t-\tau)) - \bar{\Lambda}(O_t, v_t, \theta_{t-\tau}, \gamma(t-\tau))]}_{I_2}$$

$$+ \underbrace{E[\bar{\Lambda}(O_t, v_t, \theta_{t-\tau}, \gamma(t-\tau)) - \bar{\Lambda}(O_t, v_{t-\tau}, \theta_{t-\tau}, \gamma(t-\tau))]}_{I_3}$$

$$+ \underbrace{E[\bar{\Lambda}(O_t, v_{t-\tau}, \theta_{t-\tau}, \gamma(t-\tau)) - \bar{\Lambda}(\tilde{O}_t, v_{t-\tau}, \theta_{t-\tau}, \gamma(t-\tau))]}_{I_4}$$

$$+ \underbrace{E[\bar{\Lambda}(\tilde{O}_t, v_{t-\tau}, \theta_{t-\tau}, \gamma(t-\tau))]}_{I_5}.$$

For term $I_1$,

$$\bar{\Lambda}(O_t, v_t, \theta_t, \gamma(t)) - \bar{\Lambda}(O_t, v_t, \theta_t, \gamma(t-\tau))$$
$$= \langle v_t - v^*(\theta_t, \gamma(t)), \check{g}(O_t, v_t, \theta_t, \gamma(t)) - \bar{g}(v_t, \theta_t, \gamma(t)) \rangle$$
$$\quad - \langle v_t - v^*(\theta_t, \gamma(t-\tau)), \check{g}(O_t, v_t, \theta_t, \gamma(t-\tau)) - \bar{g}(v_t, \theta_t, \gamma(t-\tau)) \rangle$$
$$= \langle v_t - v^*(\theta_t, \gamma(t)), \check{g}(O_t, v_t, \theta_t, \gamma(t)) - \check{g}(O_t, v_t, \theta_t, \gamma(t-\tau)) + \bar{g}(v_t, \theta_t, \gamma(t-\tau)) - \bar{g}(v_t, \theta_t, \gamma(t)) \rangle$$
$$\quad + \langle v^*(\theta_t, \gamma(t-\tau)) - v^*(\theta_t, \gamma(t)), \check{g}(O_t, v_t, \theta_t, \gamma(t-\tau)) - \bar{g}(v_t, \theta_t, \gamma(t-\tau)) \rangle$$
$$\leq 8 U_v N (U_c + U_\alpha)|\gamma_m(t) - \gamma_m(t-\tau)| + 4 L_2 (U_r + U_v)|\gamma_m(t) - \gamma_m(t-\tau)|,$$

where $|\gamma_m(t) - \gamma_m(t-\tau)| = \max\limits_{i=1,2,\ldots,N} |\gamma_i(t) - \gamma_i(t-\tau)|$.

For the remaining terms $I_2 - -I_5$, exactly similar analysis as Lemmas D.8–D.11 of Wu et al., [2020] can be carried out to obtain similar claims. For terms $I_4$ and $I_5$ we bound the expectation conditioned on $\theta_{t-\tau}, \gamma(t-\tau), v_{t-\tau}$ and $s_{t-\tau+1}$. Hence, after combining all the terms we get

$$E[\Lambda(O_t, v_t, \theta_t, \gamma(t), q(t), h(t))] \leq C_1 (\tau + 1) E\|\theta_t - \theta_{t-\tau}\| + C_2 b k^{\tau-1} + C_3 E\|v_t - v(t-\tau)\|$$
$$+ C_4 E|\gamma_m(t) - \gamma_m(t-\tau)|,$$

where $C_1, C_2, C_3, C_4$ are positive constants. ∎

*Proof of Theorem 2: Estimating the convergence point of Critic for Constrained Actor Critic*

We use here the update rule of $v_t$ with projection. We shall assume here that the projection set $C$ is large enough so that $v^*(t+1)$ lies within the set $C$. If this is not the case, then the algorithm will practically converge to a point that is closest in $C$ to $v^*(t+1)$. We avoid such a case by assuming that $v^*(t+1)$ lies within $C$ itself. Recall also that the set $C$ is compact and convex which ensures that the point in $C$ to which the update with increment is projected to is the closest to it and is

also unique. Thus, we obtain using the definition of $m_t$ described at the beginning of this section that

$$
\begin{aligned}
\|m_{t+1}\|^2 &= \|\Gamma(v_t + a(t)\delta_t f_{s_t}) - v^*(t+1)\|^2 \\
&\leq \|v_t + a(t)\delta_t f_{s_t} - v^*(t+1)\|^2 \\
&= \|m_t + a(t)\delta_t f_{s_t} + v^*(t) - v^*(t+1)\|^2 \\
&= \left\|m_t + a(t)\left(q(t) + \sum_{k=1}^{N} \gamma_k(t)(h_k(t) - \alpha_k) - L_t + v_t^T(f_{s_{t+1}} - f_{s_t})\right)f_{s_t} + v^*(t) - v^*(t+1)\right\|^2 \\
&= \|m_t + a(t)(g(O_t, v_t, \theta_t, \gamma(t), q(t), h(t)) + \Delta g(O_t, L_t, \theta_t, \gamma(t))) + v^*(t) - v^*(t+1)\|^2 \\
&= \|m_t\|^2 + 2a(t)\langle m_t, (g(O_t, v_t, \theta_t, \gamma(t), q(t), h(t)))\rangle + 2a(t)\langle m_t, \Delta g(O_t, L_t, \theta_t, \gamma(t))\rangle \\
&\quad + 2\langle m_t, v^*(t) - v^*(t+1)\rangle \\
&\quad + \|a(t)(g(O_t, v_t, \theta_t, \gamma(t), q(t), h(t)) + \Delta g(O_t, L_t, \theta_t, \gamma(t))) + v^*(t) - v^*(t+1)\|^2 \\
&\leq \|m_t\|^2 + 2a(t)\langle m_t, (g(O_t, v_t, \theta_t, \gamma(t), q(t), h(t)))\rangle + 2a(t)\langle m_t, \Delta g(O_t, L_t, \theta_t, \gamma(t))\rangle \\
&\quad + 2\langle m_t, v^*(t) - v^*(t+1)\rangle \\
&\quad + 2a(t)^2\|(g(O_t, v_t, \theta_t, \gamma(t), q(t), h(t)) + \Delta g(O_t, L_t, \theta_t, \gamma(t)))\|^2 + 2\|v^*(t) - v^*(t+1)\|^2 \\
&= \|m_t\|^2 + 2a(t)\langle m_t, \overline{g}(v_t, \theta_t, \gamma(t))\rangle + 2a(t)\Lambda(O_t, v_t, \theta_t, \gamma(t), q(t), h(t)) \\
&\quad + 2a(t)\langle m_t, \Delta g(O_t, L_t, \theta_t, \gamma(t))\rangle + 2\langle m_t, v^*(t) - v^*(t+1)\rangle \\
&\quad + 2a(t)^2\|(g(O_t, v_t, \theta_t, \gamma(t), q(t), h(t)) + \Delta g(O_t, L_t, \theta_t, \gamma(t)))\|^2 + 2\|v^*(t) - v^*(t+1)\|^2 \\
&\leq \|m_t\|^2 + 2a(t)\langle m_t, \overline{g}(v_t, \theta_t, \gamma(t))\rangle + 2a(t)\Lambda(O_t, v_t, \theta_t, \gamma(t), q(t), h(t)) \\
&\quad + 2a(t)\langle m_t, \Delta g(O_t, L_t, \theta_t, \gamma(t))\rangle + 2\langle m_t, v^*(t) - v^*(t+1)\rangle \\
&\quad + 8a(t)^2(U_r + U_v)^2 + 2\|v^*(t) - v^*(t+1)\|^2,
\end{aligned}
$$

where the second inequality is due to $\|x+y\|^2 \leq 2\|x\|^2 + 2\|y\|^2$ and the third one is due to $\|(g(O_t, v_t, \theta_t, \gamma(t), q(t), h(t)) + \Delta g(O_t, L_t, \theta_t, \gamma(t)))\| \leq 2(U_r + U_v)$. Now, as a consequence of Assumption 2, we have

$$
\begin{aligned}
\langle m_t, \overline{g}(v_t, \theta_t, \gamma(t))\rangle &= \langle m_t, \overline{g}(v_t, \theta_t, \gamma(t)) - \overline{g}(v^*(t), \theta_t, \gamma(t))\rangle \\
&= \langle m_t, E[(f_{s'} - f_s)^T(v_t - v^*(t))f_s]\rangle \\
&= m_t^T E[f_s(f_{s'} - f_s)^T]m_t \\
&= m_t^T A m_t \\
&\leq -\lambda_e \|m_t\|^2,
\end{aligned}
$$

where the first equation is because of the equation in Section 4.1 of the main paper. Taking expectations up to $s_{t+1}$, we have

$$
\begin{aligned}
E\|m_{t+1}\|^2 \leq\ & E\|m_t\|^2 + 2a(t)E\langle m_t, \bar{g}(v_t, \theta_t, \gamma(t),)\rangle + 2a(t)E\Lambda(O_t, v_t, \theta_t, \gamma(t), q(t), h(t)) \\
& + 2a(t)E\langle m_t, \Delta g(O_t, L_t, \theta_t, \gamma(t))\rangle + 2E\langle m_t, v^*(t) - v^*(t+1)\rangle \\
& + 8a(t)^2(U_r + U_v)^2 + 2E\|v^*(t) - v^*(t+1)\|^2 \\
\leq\ & (1 - 2\lambda_e a(t))E\|m_t\|^2 \\
& + 2a(t)E\Lambda(O_t, v_t, \theta_t, \gamma(t), q(t), h(t)) + 2a(t)E\langle m_t, \Delta g(O_t, L_t, \theta_t, \gamma(t))\rangle \\
& + 2E\langle m_t, v^*(t) - v^*(t+1)\rangle \\
& + 8a(t)^2(U_r + U_v)^2 + 2E\|v^*(t) - v^*(t+1)\|^2 \\
\leq\ & (1 - 2\lambda_e a(t))E\|m_t\|^2 \\
& + 2a(t)E\Lambda(O_t, v_t, \theta_t, \gamma(t), q(t), h(t)) + 2a(t)E\|m_t\|\|y_t\| \\
& + 2E\|m_t\|(\|v^*(\theta_t, \gamma(t)) - v^*(\theta_t, \gamma(t+1))\| + \|v^*(\theta_t, \gamma(t+1)) - v^*(\theta_{t+1}, \gamma(t+1))\|) \\
& + 8a(t)^2(U_r + U_v)^2 \\
& + 4E[\|v^*(\theta_t, \gamma(t)) - v^*(\theta_t, \gamma(t+1))\|^2 + \|v^*(\theta_t, \gamma(t+1)) - v^*(\theta_{t+1}, \gamma(t+1))\|^2] \\
\leq\ & (1 - 2\lambda_e a(t))E\|m_t\|^2 \\
& + 2a(t)E\Lambda(O_t, v_t, \theta_t, \gamma(t), q(t), h(t)) + 2a(t)E\|m_t\|\|y_t\| \\
& + 2E\|m_t\|(L_2\|\gamma_m(t) - \gamma_m(t+1)\| + L_1\|\theta_t - \theta_{t+1}\|) \\
& + 8a(t)^2(U_r + U_v)^2 + 4E[L_2^2\|\gamma_m(t) - \gamma_m(t+1)\|^2 + L_1^2\|\theta_t - \theta_{t+1}\|^2] \\
\leq\ & (1 - 2\lambda_e a(t))E\|m_t\|^2 \\
& + 2a(t)E\Lambda(O_t, v_t, \theta_t, \gamma(t), q(t), h(t)) + 2a(t)E\|m_t\|\|y_t\| \\
& + 2E\|m_t\|(L_2(U_c + U_\alpha)c(t) + 2DL_1(U_r + U_v)b(t)) \\
& + 8a(t)^2(U_r + U_v)^2 + 4E[L_2^2(U_c + U_\alpha)^2 c(t)^2 + 4L_1^2 D^2(U_r + U_v)^2 b(t)^2] \\
\leq\ & (1 - 2\lambda_e a(t))E\|m_t\|^2 \\
& + 2a(t)E[\Lambda(O_t, v_t, \theta_t, \gamma(t), q(t), h(t))] + 2a(t)E\|m_t\|\|y_t\| \\
& + 2((L_2(U_c + U_\alpha) + 2DL_1(U_r + U_v))b(t))E\|m_t\| \\
& + 8a(t)^2((U_r + U_v)^2 + 4L_2^2(U_c + U_\alpha)^2 + 16L_1^2 D^2(U_r + U_v)^2)).
\end{aligned}
$$

Rearranging now the terms in the inequality results in the following:

$$
\begin{aligned}
2\lambda_e E\|m_t\|^2 \leq\ & \frac{1}{a(t)}\left(E\|m_t\|^2 - E\|m_{t+1}\|^2\right) + 2E\Lambda(O_t, v_t, \theta_t, \gamma(t), q(t), h(t)) \\
& + 2E\|m_t\|\|y_t\| + B_q\frac{b(t)}{a(t)})E\|m_t\| + C_q a(t) \\
\leq\ & \frac{1}{a(t)}\left(E\|m_t\|^2 - E\|m_{t+1}\|^2\right) + 2E\Lambda(O_t, v_t, \theta_t, \gamma(t), q(t), h(t)) \\
& + 2\sqrt{E\|m_t\|^2}\sqrt{Ey_t^2} + B_q\frac{b(t)}{a(t)}\sqrt{E\|m_t\|^2} + C_q a(t),
\end{aligned}
$$

where

$$
\begin{aligned}
B_q &= 2((L_2(U_c + U_\alpha) + 2DL_1(U_r + U_v)), \\
C_q &= 8((U_r + U_v)^2 + 4L_2^2(U_c + U_\alpha)^2 + 16L_1^2 D^2(U_r + U_v)^2)).
\end{aligned}
$$

Now,

$$2\lambda_e \sum_{k=\tau_t}^{t} E\|m_k\|^2 \leq \underbrace{\sum_{k=\tau_t}^{t} \frac{1}{a(k)} \left(E\|m_k\|^2 - E\|m_{k+1}\|^2\right)}_{I_1}$$

$$+ \underbrace{2 \sum_{k=\tau_t}^{t} E\Lambda(O_k, v_k, \theta_k, \gamma(k), q(k), h(k))}_{I_2} + \underbrace{2 \sum_{k=\tau_t}^{t} \sqrt{E\|m_k\|^2} \sqrt{E[y_k^2]}}_{I_3}$$

$$+ \underbrace{B_q \sum_{k=\tau_t}^{t} \frac{b(k)}{a(k)}) \sqrt{E\|m_k\|^2}}_{I_4} + \underbrace{C_q \sum_{k=\tau_t}^{t} a(k)}_{I_5}. \tag{19}$$

Consider now the term $I_1$. We have the following:

$$I_1 := \sum_{k=\tau_t}^{t} \frac{1}{a(k)} \left(E\|m_k\|^2 - E\|m_{k+1}\|^2\right)$$

$$= \sum_{k=\tau_t}^{t} \left(\frac{1}{a(k)} - \frac{1}{a(k-1)}\right) E\|m_k\|^2 + \frac{1}{a(\tau_t - 1)} E\|m_{\tau_t}\|^2 - \frac{1}{a(t)} E\|m_{t+1}\|^2$$

$$\leq \sum_{k=\tau_t}^{t} \left(\frac{1}{a(k)} - \frac{1}{a(k-1)}\right) E\|m_k\|^2 + \frac{1}{a(\tau_t - 1)} E\|m_{\tau_t}\|^2$$

$$\leq 4U_v^2 \left(\sum_{k=\tau_t}^{t} \left(\frac{1}{a(k)} - \frac{1}{a(k-1)}\right) + \frac{1}{a(\tau_t - 1)}\right)$$

$$= 4U_v^2 \frac{1}{a(t)} = 4\frac{U_v^2}{c_a}(1+t)^\omega = \mathcal{O}(t^\omega).$$

For the term $I_2$, note that

$$I_2 = 2 \sum_{k=\tau_t}^{t} E[\Lambda(O_k, v_k, \theta_k, \gamma(k), q(k), h(k))]$$

$$\leq 4DC_1(U_r + U_v)(\tau_t + 1)^2 \sum_{k=\tau_t}^{t} b(k - \tau_t) + 2C_2 \sum_{k=\tau_t}^{t} b(t)$$

$$+ 4C_3(U_r + U_v)\tau_t \sum_{k=0}^{t-\tau_t} a(k) + 2C_4(U_c + U_\alpha)\tau_t \sum_{k=0}^{t-\tau_t} c(k)$$

$$\leq 4DC_1(U_r + U_v)(\tau_t + 1)^2 \sum_{k=0}^{t-\tau_t} b(k) + 2C_2(t - \tau_t + 1)b(t)$$

$$+ (4C_3(U_r + U_v)\tau_t + 2C_4(U_c + U_\alpha)\tau_t) \sum_{k=0}^{t-\tau_t} a(k)$$

$$\leq 4DC_1(U_r + U_v)(\tau_t + 1)^2 c_b \frac{(1+t-\tau_t)^{1-\sigma}}{1-\sigma} + 2C_2(t - \tau_t + 1)c_b(1+t)^{-\sigma}$$

$$+ (4C_3(U_r + U_v)\tau_t + 2C_4(U_c + U_\alpha)\tau_t)c_a \frac{(1+t-\tau_t)^{1-\omega}}{1-\omega}$$

$$\leq \left[\frac{4DC_1(U_r + U_v)(\tau_t + 1)^2 c_b}{1-\sigma} + 2C_2 c_b + \frac{(4C_3(U_r + U_v)\tau_t + 2C_4(U_c + U_\alpha)\tau_t)c_a}{1-\omega}\right](1+t)^{1-\omega}$$

$$= \mathcal{O}((\log t)^2 t^{1-\omega}).$$

Now, we get the following inequalities for the terms $I_3$, $I_4$ and $I_5$, respectively:

$$I_3 := 2 \sum_{k=\tau_t}^{t} \sqrt{E\|m_k\|^2}\sqrt{Ey_k^2} \leq 2\Big( \sum_{k=\tau_t}^{t} Ey_k^2 \Big)^{1/2}\Big( \sum_{k=\tau_t}^{t} E\|m_k\|^2 \Big)^{1/2},$$

$$I_4 := B_q \sum_{k=\tau_t}^{t} \frac{b(k)}{a(k)})\sqrt{E\|m_k\|^2} \leq \Big( \sum_{k=0}^{t-\tau_t} \frac{b(k)^2}{a(k)^2} \Big)^{1/2}\Big( \sum_{k=\tau_t}^{t} E\|m_k\|^2 \Big)^{1/2},$$

$$I_5 := C_q \sum_{k=\tau_t}^{t} a(k) \leq C_q c_a (1+t)^{1-\omega}/(1-\omega).$$

Combining all the terms, we obtain

$$2\lambda_e \sum_{k=\tau_t}^{t} E\|m_k\|^2 \leq \frac{4U_v^2}{c_a}(1+t)^{\omega}$$

$$+ \Big[\frac{4DC_1(U_r+U_v)(\tau_t+1)^2 c_b}{1-\sigma} + 2C_2 c_b + \frac{(4C_3(U_r+U_v)\tau_t + 2C_4(U_c+U_\alpha)\tau_t + C_q)c_a}{1-\omega}\Big](1+t)^{1-\omega}$$

$$+ \Big( \sum_{k=\tau_t}^{t} Ey_k^2 \Big)^{1/2}\Big( \sum_{k=\tau_t}^{t} E\|m_k\|^2 \Big)^{1/2} + \Big( \sum_{k=0}^{t-\tau_t} \frac{b(k)^2}{a(k)^2} \Big)^{1/2}\Big( \sum_{k=\tau_t}^{t} E\|m_k\|^2 \Big)^{1/2}.$$

We assume $t \geq 2\tau_t - 1$. After substituting the value of $y_k$ and applying the squaring technique as in the proof of Theorem 1, we obtain

$$\Big( \sum_{k=\tau_t}^{t} E\|v_k - v^*(k)\|^2 \Big)/(1+t-\tau_t) = \mathcal{O}\Big(\frac{1}{t^{1-\omega}}\Big) + \mathcal{O}\Big(\frac{\log t}{t^{\omega}}\Big) + \mathcal{O}\Big(\frac{1}{t^{2(\sigma-\omega)}}\Big).$$

**Remark 5** *It is important to mention here that the requirement that $v_{t+1}^*$ lies within the projection region $C$ has also been made by Wu et al., [2020] except however that they assume that the set $C$ is a ball of some radius $R_w$. We do not assume any such structure on the set $C$ except that it be compact and convex which suffices for our purpose.*

### B.4   PROOF OF COROLLARY 1

Note that we have the following result from Theorem 1:

$$\min_{0 \leq k \leq t} E[\|\nabla L(\theta_k, \gamma(k))\|^2] = \mathcal{O}(t^{\sigma-\beta})) + \mathcal{O}((\log t)^2 t^{-\sigma}) + \mathcal{O}(\epsilon_{app}) + \mathcal{O}(\varepsilon(t)), \tag{20}$$

where,

$$\varepsilon(t) = (2 \sum_{k=\tau_t}^{t} E\|A_k\|^2 + 8 \sum_{k=\tau_t}^{t} E\|B_k\|^2)/(1+t-\tau_t),$$

$$A_k = L_k - L(\theta_k, \gamma(k)),$$

$$B_k = v_k - v(\theta_k, \gamma(k)).$$

Now, from the results of Theorem 2, we have

$$\varepsilon(t) = \mathcal{O}(t^{\omega-1}) + \mathcal{O}(\log t \cdot t^{-\omega}) + \mathcal{O}(t^{-2(\sigma-\omega)}).$$

Substituting the above in (20), we have

$$\min_{0 \leq k \leq t} E[\|\nabla L(\theta_k, \gamma(k))\|^2] = \mathcal{O}(t^{\sigma-\beta}) + \mathcal{O}(\log^2 t \cdot t^{-\sigma}) + \mathcal{O}(t^{\omega-1}) + \mathcal{O}(\log t \cdot t^{-\omega}) + \mathcal{O}(t^{-2(\sigma-\omega)}) + \mathcal{O}(\epsilon_{app})$$

$$= \mathcal{O}(t^{\sigma-\beta}) + \mathcal{O}(\log^2 t \cdot t^{-\omega}) + \mathcal{O}(t^{\omega-1}) + \mathcal{O}(t^{-2(\sigma-\omega)}) + \mathcal{O}(\epsilon_{app})$$

$$= \mathcal{O}(t^{\sigma-\beta}) + \mathcal{O}(\log^2 t \cdot t^{-\omega}) + \mathcal{O}(t^{-2(\sigma-\omega)}) + \mathcal{O}(\epsilon_{app}).$$

The second equality holds because $\omega < \sigma$ while the third equality is true because $\sigma - \beta > \omega - 1$. Optimising over the choice of $\omega, \sigma, \beta$, we obtain $\omega = 0.4$, $\sigma = 0.6$ and $\beta = 1$. Hence,

$$\min_{0 \le k \le t} E[\|\nabla L(\theta_k, \gamma(k))\|^2] = \mathcal{O}(\log^2 t \cdot t^{-0.4}) + \mathcal{O}(\epsilon_{app}).$$

Therefore, in order to obtain an $\epsilon$-approximate (ignoring the approximation error as with Wu et al., [2020]) stationary point of the performance function $L(\theta, \gamma)$, namely,

$$\min_{0 \le k \le T} E[\|\nabla L(\theta_k, \gamma(k))\|^2] = \mathcal{O}(\log^2 T \cdot T^{-0.4}) + \mathcal{O}(\epsilon_{app}) \le \mathcal{O}(\epsilon_{app}) + \epsilon,$$

we need to set $T = \tilde{\mathcal{O}}(\epsilon^{-2.5})$.

## B.5    PROOF OF THEOREM 3

We use the following notation here.

$$\zeta(O, \theta, \gamma, q, h, G) = \langle \nabla L(\theta, \gamma), G^{-1} H(O, \theta, \gamma, q, h) E_{O', q, h}[G^{-1} H(O', \theta, \gamma, q, h)] \rangle,$$

where $H(\cdot)$ has been defined in the proof of Theorem 1. Further, $O' = (s, a, s')$ denotes the independent sample $s \sim \mu_\theta$, $a \sim \pi_\theta$, $s' \sim p(s, ., a)$. Hence, $E_{O', q, h}[\cdot]$ denotes the expectation w.r.t. the joint distribution of $s \sim \mu_\theta$, $a \sim \pi_\theta$, $s' \sim p(s, ., a)$, $q \sim \bar{p}(.|s, a, s')$, $h_i \sim p_i(.|s, a, s')$, $i = 1, \ldots, N$. The remaining notations are the same as those used in the proof of Theorem 1.

Now we will state and prove Lemma 8 below that will be used in the proof of Theorem 3 . Moreover, the proof of Lemma 8 shall rely on Lemmas 8.1–8.5 that we also state and prove in the following. Finally, collecting all these results together, we shall obtain the claim for Theorem 3.

**Lemma 8** *For any $t \ge 0$,*

$$E[\zeta(O_t, \theta_t, \gamma(t), q(t), h(t), G(t))] \ge -(D_1(\tau + 1) \sum_{k=t-\tau+1}^{t} E[\|\theta_k - \theta_{k-1}\|] + D_2 b k^{\tau-1} + T_1 \sum_{i=t-\tau+1}^{t} E[|\gamma_m(i) - \gamma_m(i-1)|]$$

$$+ T_G \sum_{i=t-\tau+1}^{t} E\|G(i)^{-1} - G(i-1)^{-1}\|),$$

where $D_1, D_2, T_1, T_G$ are positive constants and $t \ge \tau \ge 0$.

**Proof** We have

$$E[\zeta(O_t, \theta_t, \gamma(t), q(t), h(t), G(t))]$$
$$= E_{s_t \sim p, a_t \sim \pi_{\theta_t}, s_{t+1} \sim p}[E[\langle \nabla L(\theta_t, \gamma(t)), G(t)^{-1} H(O_t, \theta_t, \gamma(t), q(t), h(t))$$
$$\qquad - E_{O', q, h}[G(t)^{-1} H(O', \theta_t, \gamma(t), q, h)]\rangle | s_t, a_t, s_{t+1}]]$$
$$= E[\langle \nabla L(\theta_t, \gamma(t)), G(t)^{-1} \bar{H}(O_t, \theta_t, \gamma(t)) - E_{O', q, h}[G(t)^{-1} H(O', \theta_t, \gamma(t), q, h)]\rangle]$$
$$= E[\langle \nabla L(\theta_t, \gamma(t)), G(t)^{-1} \bar{H}(O_t, \theta_t, \gamma(t)) - E_{O'}[E_{q, h}[G(t)^{-1} H(O', \theta_t, \gamma(t), q, h)|s, a, s']]\rangle]$$
$$= E[\langle \nabla L(\theta_t, \gamma(t)), G(t)^{-1} \bar{H}(O_t, \theta_t, \gamma(t)) - E_{O'}[G(t)^{-1} \bar{H}(O', \theta_t, \gamma(t))]\rangle]$$
$$= E[\hat{Q}(O_t, \theta_t, \gamma(t), G(t))],$$

where

$$\bar{H}(O, \theta, \gamma) = (c(s, a, s', \gamma) - L(\theta, \gamma) + (f_{s'}{}^T - f_s{}^T) v^*(\theta, \gamma)) \nabla \log \pi_\theta(a|s),$$

$$c(s, a, s', \gamma) = \sum_q (q \cdot \bar{p}(q|s, a, s')) + \sum_{k=1}^{k=N} \gamma_k (\sum_h (h \cdot p_k(h|s, a, s')) - \alpha_k).$$

The proof makes use of the supporting lemmas 8.1–8.5 below.

**lemma 8.1** *For any $t \geq 0$,*

$$|\hat{Q}(O_t, \theta_t, \gamma(t), G(t)) - \hat{Q}(O_t, \theta_t, \gamma(t), G(t-\tau))| \leq T_G \|G(t)^{-1} - G(t-\tau)^{-1}\|,$$

*for some $T_G > 0$.*

*Proof* The following holds:

$$\hat{Q}(O_t, \theta_t, \gamma(t), G(t)) - \hat{Q}(O_t, \theta_t, \gamma(t), G(t-\tau))$$
$$= \langle \nabla L(\theta_t, \gamma(t)), G(t)^{-1}\bar{H}(O_t, \theta_t, \gamma(t)) - E_{O'}[G(t)^{-1}\bar{H}(O', \theta_t, \gamma(t))]\rangle$$
$$\quad - \langle \nabla L(\theta_t, \gamma(t)), G(t-\tau)^{-1}\bar{H}(O_t, \theta_t, \gamma(t)) - E_{O'}[G(t-\tau)^{-1}\bar{H}(O', \theta_t, \gamma(t))]\rangle$$
$$= \langle \nabla L(\theta_t, \gamma(t)), (G(t)^{-1} - G(t-\tau)^{-1})\bar{H}(O_t, \theta_t, \gamma(t)) - E_{O'}[(G(t)^{-1} - G(t-\tau)^{-1})\bar{H}(O', \theta_t, \gamma(t))]\rangle$$
$$\leq 2D(U_r + U_v)\|\nabla L(\theta_t, \gamma(t))\|\|G(t)^{-1} - G(t-\tau)^{-1}\|$$
$$\leq 2D(U_r + U_v)U_L\|G(t)^{-1} - G(t-\tau)^{-1}\|.$$

The claim follows by letting $T_G = 2D(U_r + U_v)U_L > 0$.

**lemma 8.2** *For any $t \geq 0$,*

$$|\hat{Q}(O_t, \theta_t, \gamma(t), G(t-\tau)) - \hat{Q}(O_t, \theta_t, \gamma(t-\tau), G(t-\tau))| \leq T_1|\gamma_m(t) - \gamma_m(t-\tau)|$$

*for some $T_1 > 0$.*

*Proof* Denoting $O = (s, a, s')$, we have for any $\theta, \gamma_1, \gamma_2$, that

$$\hat{Q}(O, \theta, \gamma^1, G) - \hat{Q}(O, \theta, \gamma^2, G)$$
$$= \langle \nabla L(\theta, \gamma^1), G^{-1}\bar{H}(O, \theta, \gamma^1) - E_{O'}[G^{-1}\bar{H}(O', \theta, \gamma^1)]\rangle - \langle \nabla L(\theta, \gamma^2), G^{-1}\bar{H}(O, \theta, \gamma^2) - E_{O'}[G^{-1}\bar{H}(O', \theta, \gamma^2)]\rangle$$
$$= \langle \nabla L(\theta, \gamma^1), G^{-1}\bar{H}(O, \theta, \gamma^1) - E_{O'}[G^{-1}\bar{H}(O', \theta, \gamma^1)]\rangle - \langle \nabla L(\theta, \gamma^1), G^{-1}\bar{H}(O, \theta, \gamma^2) - E_{O'}[G^{-1}\bar{H}(O', \theta, \gamma^2)]\rangle$$
$$\quad + \langle \nabla L(\theta, \gamma^1), G^{-1}\bar{H}(O, \theta, \gamma^2) - E_{O'}[G^{-1}\bar{H}(O', \theta, \gamma^2)]\rangle - \langle \nabla L(\theta, \gamma^2), G^{-1}\bar{H}(O, \theta, \gamma^2) - E_{O'}[G^{-1}\bar{H}(O', \theta, \gamma^2)]\rangle$$
$$= \underbrace{\langle \nabla L(\theta, \gamma^1), G^{-1}\bar{H}(O, \theta, \gamma^1) - G^{-1}\bar{H}(O, \theta, \gamma^2) - E_{O'}[G^{-1}\bar{H}(O', \theta, \gamma^1)] + E_{O'}[G^{-1}\bar{H}(O', \theta, \gamma^2)]\rangle}_{I_1}$$
$$\quad + \underbrace{\langle \nabla L(\theta, \gamma^1) - \nabla L(\theta, \gamma^2), G^{-1}\bar{H}(O, \theta, \gamma^2) - E_{O'}[G^{-1}\bar{H}(O', \theta, \gamma^2)]\rangle}_{I_2},$$

where $G$ is a non-singular square matrix and $\|G\| < U_G$. We have by lemma 2 that

$$\|\nabla L(\theta, \gamma^1) - \nabla L(\theta, \gamma^2)\| \leq C|\gamma_m^1 - \gamma_m^2|.$$

Now,

$$\|\bar{H}(O, \theta, \gamma^1) - \bar{H}(O, \theta, \gamma^2)\|$$
$$= \|(c(s, a, s', \gamma^1) - c(s, a, s', \gamma^2) - L(\theta, \gamma^1) + L(\theta, \gamma^2) + (f_{s'}^T - f_s^T)(v^*(\theta, \gamma^1) - v^*(\theta, \gamma^2)))\nabla \log \pi_\theta(a|s)\|$$
$$\leq D(|c(s, a, s', \gamma^1) - c(s, a, s', \gamma^2)| + |L(\theta, \gamma^1) - L(\theta, \gamma^2)| + 2\|v^*(\theta, \gamma^1) - v^*(\theta, \gamma^2)\|)$$
$$\leq D(2N(U_c + U_\alpha)|\gamma_m^1 - \gamma_m^2| + 2L_2|\gamma_m^1 - \gamma_m^2|),$$

where $|\gamma_m^1 - \gamma_m^2| = \max\limits_{i=1,2,3...,N}|\gamma_i^1 - \gamma_i^2|$. Hence (for the term $I_1$), we have that

$$I_1 \leq 4D(N(U_c + U_\alpha) + L_2)\|\nabla L(\theta, \gamma^1)\|U_G|\gamma_m^1 - \gamma_m^2|.$$

Now observe that (for the term $I_2$),

$$I_2 \leq \|\nabla L(\theta, \gamma^1) - \nabla L(\theta, \gamma^2)\|\|\bar{H}(O, \theta, \gamma^2) - E_{O'}[\bar{H}(O', \theta, \gamma^2)]\|$$
$$\leq 4D(U_r + U_v)\|\nabla L(\theta, \gamma^1) - \nabla L(\theta, \gamma^2)\|$$
$$\leq 4DC(U_r + U_v)U_G|\gamma_m^1 - \gamma_m^2|$$

Combining the RHS of the two terms, we obtain

$$|\hat{Q}(O,\theta,\gamma^1,G) - \hat{Q}(O,\theta,\gamma^2,G)| \leq T_1|\gamma_m^1 - \gamma_m^2|,$$

where $T_1 = 4D(N(U_c + U_\alpha) + L_2)U_L U_G + 4DC(U_r + U_v)U_G$.

**lemma 8.3** *For any* $t \geq 0, \theta_1, \theta_2, G, \gamma = (\gamma_1, \gamma_2, ..., \gamma_N)^T$ *with* $0 \leq \gamma_i \leq M$ *for* $i \in \{1,2,,,..,N\}$ *and* $G$ *being a non-singular square matrix with* $\|G^{-1}\| \leq U_G$,

$$|\hat{Q}(O,\theta_1,\gamma,G) - \hat{Q}(O,\theta_2,\gamma,G)| \leq T_2\|\theta_1 - \theta_2\|,$$

*for some* $T_2 > 0$.

*Proof* Denote $O = (s,a,s')$. We have for any $\theta_1, \theta_2, \gamma, G$, the following:

$$\hat{Q}(O,\theta_1,\gamma,G) - \hat{Q}(O,\theta_2,\gamma,G)$$
$$= \langle \nabla L(\theta_1,\gamma), G^{-1}\bar{H}(O,\theta_1,\gamma) - E_{O'}[G^{-1}\bar{H}(O',\theta_1,\gamma)]\rangle - \langle \nabla L(\theta_2,\gamma), G^{-1}\bar{H}(O,\theta_2,\gamma) - E_{O'}[G^{-1}\bar{H}(O',\theta_2,\gamma)]\rangle$$
$$= \langle \nabla L(\theta_1,\gamma), G^{-1}\bar{H}(O,\theta_1,\gamma) - E_{O'}[G^{-1}\bar{H}(O',\theta_1,\gamma)]\rangle - \langle \nabla L(\theta_1,\gamma), G^{-1}\bar{H}(O,\theta_2,\gamma) - E_{O'}[G^{-1}\bar{H}(O',\theta_2,\gamma)]\rangle$$
$$\quad + \langle \nabla L(\theta_1,\gamma), G^{-1}\bar{H}(O,\theta_2,\gamma) - E_{O'}[G^{-1}\bar{H}(O',\theta_2,\gamma)]\rangle - \langle \nabla L(\theta_2,\gamma), G^{-1}\bar{H}(O,\theta_2,\gamma) - E_{O'}[G^{-1}\bar{H}(O',\theta_2,\gamma)]\rangle$$
$$= \underbrace{\langle \nabla L(\theta_1,\gamma), G^{-1}\bar{H}(O,\theta_1,\gamma) - G^{-1}\bar{H}(O,\theta_2,\gamma) - E_{O'}[G^{-1}\bar{H}(O',\theta_1,\gamma)] + E_{O'}[G^{-1}\bar{H}(O',\theta_2,\gamma)]\rangle}_{I_1}$$
$$\quad + \underbrace{\langle \nabla L(\theta_1,\gamma) - \nabla L(\theta_2,\gamma), G^{-1}\bar{H}(O,\theta_2,\gamma) - E_{O'}[G^{-1}\bar{H}(O',\theta_2,\gamma)]\rangle}_{I_2}.$$

Now,

$$\|\bar{H}(O,\theta_1,\gamma) - \bar{H}(O,\theta_2,\gamma)\|$$
$$= \|(c(s,a,s',\gamma) - L(\theta_1,\gamma) + (f_{s'}{}^T - f_s{}^T)v^*(\theta_1,\gamma))\nabla \log \pi_{\theta_1}(a|s)$$
$$\quad - (c(s,a,s',\gamma) - L(\theta_2,\gamma) + (f_{s'}{}^T - f_s{}^T)v^*(\theta_2,\gamma))\nabla \log \pi_{\theta_2}(a|s)\|$$
$$\leq \|(c(s,a,s',\gamma) - L(\theta_1,\gamma) + (f_{s'}{}^T - f_s{}^T)v^*(\theta_1,\gamma))(\nabla \log \pi_{\theta_1}(a|s) - \nabla \log \pi_{\theta_2}(a|s))\|$$
$$\quad + \|(L(\theta_2,\gamma) - L(\theta_1,\gamma) + (f_{s'}{}^T - f_s{}^T)(v^*(\theta_1,\gamma) - v^*(\theta_1,\gamma)))\nabla \log \pi_{\theta_2}(a|s)\|$$
$$\leq 2(U_r + U_v)M_m\|\theta_1 - \theta_2\| + D(|L(\theta_2,\gamma) - L(\theta_1,\gamma)| + 2L_1\|\theta_1 - \theta_2\|).$$

Also, clearly

$$|L(\theta_1,\gamma) - L(\theta_2,\gamma)|$$
$$= |\sum_{s \in S}\mu_{\theta_1}(s)\sum_{a \in A(s)}\pi_{\theta_1}(s,a)(d(s,a) + \sum_{k=1}^N\gamma(k)(h_k(s,a) - \alpha_k)) - \sum_{s \in S}\mu_{\theta_2}(s)\sum_{a \in A(s)}\pi_{\theta_2}(s,a)(d(s,a) + \sum_{k=1}^N\gamma(k)(h_k(s,a) - \alpha_k))|$$
$$\leq 2U_r d_{TV}(\mu_{\theta_1} \otimes \pi_{\theta_1}, \mu_{\theta_2} \otimes \pi_{\theta_2})$$
$$\leq 2U_r|A|L\left(1 + \lceil \log_k b^{-1}\rceil + 1/(1-k)\right)\|\theta_1 - \theta_2\|$$
$$= C_L\|\theta_1 - \theta_2\|.$$

Hence,

$$\|\bar{H}(O,\theta_1,\gamma) - \bar{H}(O,\theta_2,\gamma)\| \leq 2(U_r + U_v)M_m\|\theta_1 - \theta_2\| + DC_L\|\theta_1 - \theta_2\| + 2L_1D\|\theta_1 - \theta_2\|.$$

Also, note that

$$\|E_{O'}[\bar{H}(O', \theta_1, \gamma)] - E_{O'}[\bar{H}(O', \theta_2, \gamma)]\|$$
$$= \|E_{\theta_1}[\bar{H}(O', \theta_1, \gamma)] - E_{\theta_2}[\bar{H}(O', \theta_2, \gamma)]\|$$
$$\leq \|E_{\theta_1}[\bar{H}(O', \theta_1, \gamma)] - E_{\theta_1}[\bar{H}(O', \theta_2, \gamma)]\| + \|E_{\theta_1}[\bar{H}(O', \theta_2, \gamma)] - E_{\theta_2}[\bar{H}(O', \theta_2, \gamma)]\|$$
$$\leq E_{\theta_1}\|\bar{H}(O', \theta_1, \gamma) - \bar{H}(O', \theta_2, \gamma)\| + 4D(U_r + U_v)d_{TV}(\mu_{\theta_1} \otimes \pi_{\theta_1}, \mu_{\theta_2} \otimes \pi_{\theta_2})$$
$$\leq \left[2(U_r + U_v)M_m + DC_L + 2L_1D + 4D(U_r + U_v)|A|U_rL\left(1 + \lceil\log_k b^{-1}\rceil + \frac{1}{1-k}\right)\right]\|\theta_1 - \theta_2\|$$
$$= A_2\|\theta_1 - \theta_2\|.$$

Thus, we have

$$I_1 \leq 2U_LU_G(U_r + U_v)M_m\|\theta_1 - \theta_2\| + U_LU_GDC_L\|\theta_1 - \theta_2\| + 2U_LU_GL_1D\|\theta_1 - \theta_2\| + U_GU_LA_2\|\theta_1 - \theta_2\|.$$

For the term $I_2$, we have

$$I_2 \leq \|\nabla L(\theta_1, \gamma) - \nabla L(\theta_2, \gamma)\|\|G^{-1}\bar{H}(O, \theta_2, \gamma) - E_{O'}[G^{-1}\bar{H}(O', \theta_2, \gamma)]\|$$
$$\leq 4D(U_r + U_v)U_GM_L\|\theta_1 - \theta_2\|.$$

The last inequality follows from Lemma 1.

Finally, we have

$$\hat{Q}(O, \theta_1, \gamma, G) - \hat{Q}(O, \theta_2, \gamma, G) \leq T_2\|\theta_1 - \theta_2\|,$$

where,

$$T_2 = 2U_LU_G(U_r + U_v)M_m + U_LU_GDC_L + 2U_LU_GL_1D + U_GU_LA_2 + 4D(U_r + U_v)U_GM_L,$$
$$A_2 = 2(U_r + U_v)M_m + DC_L + 2L_1D + 4D(U_r + U_v)|A|U_rL\left(1 + \lceil\log_k b^{-1}\rceil + \frac{1}{1-k}\right).$$

**lemma 8.4** *For any* $t \geq 0$, *conditioned on* $\theta_{t-\tau}, \gamma(t - \tau)$ *and* $G(t - \tau)$,

$$|E[(\hat{Q}(O_t, \theta_{t-\tau}, \gamma(t - \tau), G(t - \tau)) - \hat{Q}(\tilde{O}_t, \theta_{t-\tau}, \gamma(t - \tau), G(t - \tau)))|\theta_{t-\tau}, \gamma(t - \tau), G(t - \tau)]|$$

$$\leq 2DU_G(U_r + U_v)U_L|A|L\sum_{i=t-\tau}^{t} E\|\theta_i - \theta_{t-\tau}\|$$

*Proof* By the definition of $\hat{Q}(O, \theta, \gamma)$,

$$E[(\hat{Q}(O_t, \theta_{t-\tau}, \gamma(t - \tau), G(t - \tau)) - \hat{Q}(\tilde{O}_t, \theta_{t-\tau}, \gamma(t - \tau), G(t - \tau))|\theta_{t-\tau}, \gamma(t - \tau), G(t - \tau)]$$
$$= E[\langle\nabla L(\theta_{t-\tau}, \gamma(t - \tau)), G(t - \tau)^{-1}\bar{H}(O_t, \theta_{t-\tau}, \gamma(t - \tau) - G(t - \tau)^{-1}\bar{H}(\tilde{O}_t, \theta_{t-\tau}, \gamma(t - \tau)\rangle|\theta_{t-\tau}, \gamma(t - \tau), G(t - \tau)]$$
$$= E[(\langle\nabla L(\theta_{t-\tau}, \gamma(t - \tau)), G(t - \tau)^{-1}\bar{H}(O_t, \theta_{t-\tau}, \gamma(t - \tau)\rangle$$
$$\quad - \langle\nabla L(\theta_{t-\tau}, \gamma(t - \tau)), G(t - \tau)^{-1}\bar{H}(\tilde{O}_t, \theta_{t-\tau}, \gamma(t - \tau))\rangle)|\theta_{t-\tau}, \gamma(t - \tau), G(t - \tau)]$$
$$\leq 4U_GD(U_r + U_v)U_Ld_{TV}(P(O_t = .|s_{t-\tau+1}, \theta_{t-\tau}), (P(\tilde{O}_t = .|s_{t-\tau+1}, \theta_{t-\tau})).$$

Now,

$$d_{TV}(P(O_t = .|s_{t-\tau+1}, \theta_{t-\tau}), (P(\tilde{O}_t = .|s_{t-\tau+1}, \theta_{t-\tau})) \leq \frac{1}{2}|A|L\sum_{i=t-\tau}^{t} E\|\theta_i - \theta_{t-\tau}\|.$$

This inequality follows in a similar manner as the proof of Lemma D.2 of Wu et al., [2020]. Hence,

$$E[\hat{Q}(O_t, \theta_{t-\tau}, \gamma(t-\tau), G(t-\tau)) - \hat{Q}(\tilde{O}_t, \theta_{t-\tau}, \gamma(t-\tau), G(t-\tau))|\theta_{t-\tau}, \gamma(t-\tau), G(t-\tau)]$$

$$\leq 2DU_G(U_r + U_v)U_L|A|L \sum_{i=t-\tau}^{t} E\|\theta_i - \theta_{t-\tau}\|.$$

The claim follows.

**lemma 8.5** *For any $t \geq 0$, conditioned on $\theta_{t-\tau}, \gamma(t-\tau)$ and $G(t-\tau)$,*

$$|E[(\hat{Q}(\tilde{O}_t, \theta_{t-\tau}, \gamma(t-\tau), G(t-\tau)) - \hat{Q}(O'_t, \theta_{t-\tau}, \gamma(t-\tau), G(t-\tau)))|\theta_{t-\tau}, \gamma(t-\tau), G(t-\tau)]|$$

$$\leq 4DU_G(U_r + U_v)U_L bk^{\tau-1}.$$

*Proof*

$$E[(\hat{Q}(\tilde{O}_t, \theta_{t-\tau}, \gamma(t-\tau), G(t-\tau)) - \hat{Q}(O'_t, \theta_{t-\tau}, \gamma(t-\tau), G(t-\tau)))|\theta_{t-\tau}, \gamma(t-\tau), G(t-\tau)]$$

$$= E[\langle \nabla L(\theta_{t-\tau}, \gamma(t-\tau)), G(t-\tau)^{-1}\bar{H}(\tilde{O}_t, \theta_{t-\tau}, \gamma(t-\tau)) - G(t-\tau)^{-1}\bar{H}(O'_t, \theta_{t-\tau}, \gamma(t-\tau))\rangle|\theta_{t-\tau}, \gamma(t-\tau), G(t-\tau)]$$

$$= E[(\langle \nabla L(\theta_{t-\tau}, \gamma(t-\tau)), G(t-\tau)^{-1}\bar{H}(\tilde{O}_t, \theta_{t-\tau}, \gamma(t-\tau))\rangle$$

$$\quad - \langle \nabla L(\theta_{t-\tau}, \gamma(t-\tau)), G(t-\tau)^{-1}\bar{H}(O'_t, \theta_{t-\tau}, \gamma(t-\tau))\rangle)|\theta_{t-\tau}, \gamma(t-\tau), G(t-\tau)]$$

$$\leq 4DU_G(U_r + U_v)U_L d_{TV}(P(\tilde{O}_t = .|s_{t-\tau+1}, \theta_{t-\tau}), \mu_{\theta_{t-\tau}} \otimes \pi_{\theta_{t-\tau}} \otimes P)$$

$$\leq 4DU_G(U_r + U_v)U_L bk^{\tau-1}.$$

The last inequality comes using an inequality on the total variation distance that is shown in the proof of Lemma D.3 of Wu et al., [2020].

*Proof of Lemma 8:*

We decompose $E[\hat{Q}(O_t, \theta_t, \gamma(t), G(t))]$ as:

$$E[\hat{Q}(O_t, \theta_t, \gamma(t), G(t))]$$

$$= E[\hat{Q}(O_t, \theta_t, \gamma(t), G(t)) - \hat{Q}(O_t, \theta_t, \gamma(t), G(t-\tau))] + E[\hat{Q}(O_t, \theta_t, \gamma(t), G(t-\tau)) - \hat{Q}(O_t, \theta_t, \gamma(t-\tau), G(t-\tau))]$$

$$\quad + E[\hat{Q}(O_t, \theta_t, \gamma(t-\tau), G(t-\tau)) - \hat{Q}(O_t, \theta_{t-\tau}, \gamma(t-\tau), G(t-\tau))]$$

$$\quad + E[\hat{Q}(O_t, \theta_{t-\tau}, \gamma(t-\tau), G(t-\tau)) - \hat{Q}(\tilde{O}_t, \theta_{t-\tau}, \gamma(t-\tau), G(t-\tau))]$$

$$\quad + E[\hat{Q}(\tilde{O}_t, \theta_{t-\tau}, \gamma(t-\tau), G(t-\tau)) - \hat{Q}(O'_t, \theta_{t-\tau}, \gamma(t-\tau), G(t-\tau))] + E[\hat{Q}(O'_t, \theta_{t-\tau}, \gamma(t-\tau))],$$

where $\tilde{O}_t$ is from the auxiliary Markov chain and $O'_t = (s_t, a_t, s_{t+1})$ is from the stationary distribution with $s_t \sim \mu_{\theta_{t-\tau}}, a_t \sim \pi_{\theta_{t-\tau}}, s_{t+1} \sim p(s_t, ., a_t)$ and which actually satisfies $E[\hat{Q}(O'_t, \theta_{t-\tau}, \gamma(t-\tau))] = 0$. By collecting the corresponding bounds from Lemmas 8.1–8.5, we have

$$E[\hat{Q}(O_t, \theta_t, \gamma(t), G(t))] \geq -T_1 E|\gamma_m(t) - \gamma_m(t-\tau)| - T_2 E\|\theta_t - \theta_{t-\tau}\| - 2DU_G(U_r + U_v)U_L|A|L\sum_{i=t-\tau}^{t} E\|\theta_i - \theta_{t-\tau}\|$$

$$- 4DU_G(U_r + U_v)U_L bk^{\tau-1} - T_G E\|G(t)^{-1} - G(t-\tau)^{-1}\|$$

$$\geq -T_1 \sum_{i=t-\tau+1}^{t} E|\gamma_m(i) - \gamma_m(i-1)| - T_2 \sum_{i=t-\tau+1}^{t} E\|\theta_i - \theta_{i-1}\|$$

$$- 2D(U_r + U_v)U_L|A|L \sum_{i=t-\tau+1}^{t} \sum_{j=t-\tau+1}^{i} E\|\theta_j - \theta_{j-1}\| - 4D(U_r + U_v)U_L bk^{\tau-1}$$

$$- T_G E\|G(t)^{-1} - G(t-\tau)^{-1}\|$$

$$\geq -T_1 \sum_{i=t-\tau+1}^{t} E|\gamma_m(i) - \gamma_m(i-1)| - T_2 \sum_{i=t-\tau+1}^{t} E\|\theta_i - \theta_{i-1}\|$$

$$- 2D(U_r + U_v)U_L|A|L\tau \sum_{j=t-\tau+1}^{t} E\|\theta_j - \theta_{j-1}\| - 4D(U_r + U_v)U_L bk^{\tau-1}$$

$$- T_G E\|G(t)^{-1} - G(t-\tau)^{-1}\|$$

$$\geq -\Big(D_1(\tau+1) \sum_{k=t-\tau+1}^{t} E\|\theta_k - \theta_{k-1}\| + D_2 bk^{\tau-1} + T_1 \sum_{i=t-\tau+1}^{t} E|\gamma_m(i) - \gamma_m(i-1)|$$

$$+ T_G \sum_{i=t-\tau+1}^{t} E\|G(i)^{-1} - G(i-1)^{-1}\|\Big),$$

where $D_1 := \max\{T_2, 2DU_G(U_r + U_v)U_L|A|L\}$ and $D_2 := 4DU_G(U_r + U_v)U_L$, respectively, which completes the proof. ∎

Under the update rule of Algorithm 2 for the actor, we have:

$$\theta_{t+1} = \theta_t + b(t)G(t)^{-1}\delta_t \Psi_{s_t a_t}.$$

Now,

$$G(t)^{-1}\delta_t \nabla \log \pi_{\theta_t}(a_t|s_t)$$

$$= G(t)^{-1}\Big(q(t) + \sum_{k=1}^{k=N} \gamma_k(t)(h_k(t) - \alpha_k) - L_t + v_t^T(f(s_{t+1}) - f(s_t))\Big)\nabla \log \pi_{\theta_t}(a_t|s_t)$$

$$= G(t)^{-1}\Big(q(t) + \sum_{k=1}^{k=N} \gamma_k(t)(h_k(t) - \alpha_k) - L(\theta_t, \gamma(t)) + L(\theta_t, \gamma(t)) - L_t + (f(s_{t+1})^T - f(s_t)^T)(v_t - v_t^*)$$

$$+ (f(s_{t+1})^T - f(s_t)^T)v_t^*\Big)\nabla \log \pi_{\theta_t}(a_t|s_t)$$

$$= G(t)^{-1}\Big(L(\theta_t, \gamma(t)) - L_t + (f(s_{t+1})^T - f(s_t)^T)(v_t - v_t^*)\Big)\nabla \log \pi_{\theta_t}(a_t|s_t)$$

$$+ G(t)^{-1}\Big(q(t) + \sum_{k=1}^{k=N} \gamma_k(t)(h_k(t) - \alpha_k) - L(\theta_t, \gamma(t)) + (f(s_{t+1})^T - f(s_t)^T)v_t^*\Big)\nabla \log \pi_\theta(a_t|s_t)$$

$$= G(t)^{-1}\big(\Delta H(O_t, L_t, v_t, \theta_t, \gamma(t)) + H(O_t, \theta_t, \gamma(t), q(t), h(t))\big).$$

We have,

$$E_{O',q,h}[G^{-1}(H(O',\theta,\gamma,q,h) - \Delta H'(O',\theta,\gamma))]$$

$$= G^{-1}E_{O',q,h}[q + \sum_{k=1}^{k=N}\gamma(k)(h_k - \alpha_k) - L(\theta,\gamma) + V^{(\theta,\gamma)}(s') - V^{(\theta,\gamma)}(s))\nabla\log\pi_\theta(a|s)]$$

$$= G^{-1}\nabla L(\theta,\gamma),$$

where $E_{O',q,h}[\cdot]$ denotes the expectation w.r.t $s \sim \mu_\theta, a \sim \pi_\theta, s' \sim p(s,.,a), q \sim \bar{p}(.|s,a,s'), h_i \sim p_i(.|s,a,s')$. Hence,

$$L(\theta_{t+1},\gamma(t)) \geq L(\theta_t,\gamma(t)) + b(t)\langle\nabla L(\theta_t,\gamma(t)), G(t)^{-1}\Delta H(O_t,L_t,v_t,\theta_t,\gamma(t)) + G(t)^{-1}H(O_t,\theta_t,\gamma(t),q(t),h(t))\rangle$$

$$- M_L b(t)^2\|G(t)^{-1}\delta_t\nabla\log\pi_{\theta_t}(a_t|s_t)\|^2$$

$$\geq L(\theta_t,\gamma(t)) + b(t)\langle\nabla L(\theta_t,\gamma(t)), G(t)^{-1}\Delta H(O_t,L_t,v_t,\theta_t,\gamma(t))\rangle$$

$$+ b(t)\langle\nabla L(\theta_t,\gamma(t)), G(t)^{-1}H(O_t,\theta_t,\gamma(t),q(t),h(t)) - E_{O',q(t),h(t)}[G(t)^{-1}H(O',\theta_t,\gamma(t),q(t),h(t))]\rangle$$

$$+ b(t)\langle\nabla L(\theta_t,\gamma(t)), E_{O',q(t),h(t)}[G(t)^{-1}H(O',\theta_t,\gamma(t),q(t),h(t))]\rangle - M_L b(t)^2\|G(t)^{-1}\delta_t\nabla\log\pi_{\theta_t}(a_t|s_t)\|^2$$

$$\geq L(\theta_t,\gamma(t)) + b(t)\langle\nabla L(\theta_t,\gamma(t)), G(t)^{-1}\Delta H(O_t,L_t,v_t,\theta_t,\gamma(t))\rangle$$

$$+ b(t)\zeta(O_t,\theta_t,\gamma(t),q(t),h(t),G(t)) + b(t)\langle\nabla L(\theta_t,\gamma(t)), G(t)^{-1}\nabla L(\theta_t,\gamma(t))\rangle$$

$$+ b(t)\langle\nabla L(\theta_t,\gamma(t)), E_{O'}[G(t)^{-1}\Delta H'(O',\theta_t,\gamma(t))]\rangle - M_L b(t)^2\|G(t)^{-1}\delta_t\nabla\log\pi_{\theta_t}(a_t|s_t)\|^2$$

$$\geq L(\theta_t,\gamma(t)) + b(t)\langle\nabla L(\theta_t,\gamma(t)), G(t)^{-1}\Delta H(O_t,L_t,v_t,\theta_t,\gamma(t))\rangle$$

$$+ b(t)\zeta(O_t,\theta_t,\gamma(t),q(t),h(t),G(t)) + b(t)\lambda\|\nabla L(\theta_t,\gamma(t))\|^2$$

$$+ b(t)\langle\nabla L(\theta_t,\gamma(t)), E_{O'}[G(t)^{-1}\Delta H'(O',\theta_t,\gamma(t))]\rangle - M_L b(t)^2\|G(t)^{-1}\delta_t\nabla\log\pi_{\theta_t}(a_t|s_t)\|^2.$$

The last inequality holds as $G(t)^{-1}$ is a positive definite and symmetric matrix with minimum eigenvalue $\geq \lambda$. After rearranging the terms and summing the expectation of the terms from $\tau_t$ to $t$, we have

$$\lambda\sum_{k=\tau_t}^{t}E\|\nabla L(\theta_k,\gamma(k))\|^2 \leq \underbrace{\sum_{k=\tau_t}^{t}\frac{1}{b(k)}(E[L(\theta_{k+1},\gamma(k))] - E[L(\theta_k,\gamma(k))])}_{I_1}$$

$$\underbrace{-\sum_{k=\tau_t}^{t}E\langle\nabla L(\theta_k,\gamma(k)), G(k)^{-1}\Delta H(O_k,L_k,v_k,\theta_k,\gamma(k))\rangle}_{I_2}$$

$$\underbrace{-\sum_{k=\tau_t}^{t}E[\zeta(O_k,\theta_k,\gamma(k),q(k),h(k),G(k))]}_{I_3}$$

$$\underbrace{-\sum_{k=\tau_t}^{t}E\langle\nabla L(\theta_k,\gamma(k)), E_{O'}[G(k)^{-1}\Delta H'(O',\theta_k,\gamma(k))]\rangle}_{I_4}$$

$$\underbrace{+\sum_{k=\tau_t}^{t}b(k)E[M_L\|G(k)^{-1}\delta_k\nabla\log\pi_{\theta_k}(a_k|s_k)\|^2]}_{I_5}.$$

For term $I_1$,

$$\sum_{k=\tau_t}^{t}\frac{1}{b(k)}(E[L(\theta_{k+1},\gamma(k))] - E[L(\theta_k,\gamma(k))]) \leq B_1(t - \tau_t + 1)^{1-\beta+\sigma} + B_2(1+t)^\sigma.$$

This inequality comes from part $I_1$ of Section B.1.

For term $I_2$,

$$-\sum_{k=\tau_t}^{t} E\langle\nabla L(\theta_k,\gamma(k)), G(k)^{-1}\Delta H(O_k, L_k, v_k, \theta_k, \gamma(k))\rangle$$

$$\leq \sqrt{\sum_{k=\tau_t}^{t} E\|\nabla L(\theta_k,\gamma(k))\|^2}\sqrt{\sum_{k=\tau_t}^{t} E\|G(k)^{-1}\Delta H(O_k, L_k, v_k, \theta_k, \gamma(k))\|^2}.$$

Now,

$$E\|G(k)^{-1}\Delta H(O_k, L_k, v_k, \theta_k, \gamma(k))\|^2$$
$$= E\|G(k)^{-1}(L(\theta_k,\gamma(k)) - L_k + (f(s_{k+1})^T - f(s_k)^T)(v(k) - v(\theta_k,\gamma(k))^*))\nabla\log\pi_{\theta_k}(a_k|s_k)\|^2$$
$$\leq U_G^2 D^2(2E\|A_k\|^2 + 8E\|B_k\|^2).$$

Hence,

$$I_2 \leq DU_G\sqrt{\sum_{k=\tau_t}^{t} E\|\nabla L(\theta_k,\gamma(k))\|^2}\sqrt{\sum_{k=\tau_t}^{t}(2E\|A_k\|^2 + 8E\|B_k\|^2)}.$$

Next, for the term $I_3$, we have

$$E[\zeta(O_t,\theta_t,\gamma(t),q(t),h(t),G(t))] \geq -(D_1(\tau+1)\sum_{k=t-\tau+1}^{t} E\|\theta_k - \theta_{k-1}\| + D_2 bk^{\tau-1} + T_1\sum_{i=t-\tau+1}^{t} E|\gamma_m(i) - \gamma_m(i-1)|$$

$$+ T_G\sum_{i=t-\tau+1}^{t} E\|G(i)^{-1} - G(i-1)^{-1}\|).$$

We can write

$$G(i)^{-1} - G(i-1)^{-1} = G(i)^{-1}(G(i-1) - G(i))G(i-1)^{-1}.$$

By letting $\tau \triangleq \tau_t$, we have

$$E[\zeta(O_k,\theta_k,\gamma(k),q(k),h(k),G(k))] \geq -(2D_1 D(\tau_t+1)^2(U_r + U_v)U_G b(k-\tau_t) + D_2 b(t)$$
$$+ T_1(U_c + U_\alpha)(\tau_t+1)c(k-\tau_t) + T_G(\tau_t+1)U_G^2(U_G + D^2)a(k-\tau_t)).$$

After simplifying, we have

$$\sum_{k=\tau_t}^{t} E[\zeta(O_k,\theta_k,\gamma(k),q(k),h(k),G(k))] \geq -A_1(\tau_t+1)^2(1+t)^{1-\omega}$$

for some $A_1 > 0$. Now, for term $I_4$, we have

$$\sum_{k=\tau_t}^{t} E\langle\nabla L(\theta_k,\gamma(k)), E_{O'}[G(k)^{-1}\Delta H'(O',\theta_k,\gamma(k))]\rangle \geq -2DU_L U_G\epsilon_{app}(1+t-\tau_t).$$

Also, for the term $I_5$, we have

$$\sum_{k=\tau_t}^{t} b(k)E[M_L\|G(k)^{-1}\delta_k\nabla\log\pi_{\theta_k}(a_k|s_k)\|^2] \leq C_1(1+t-\tau_t)^{1-\sigma},$$

where $C_1$ is a positive constant. After combining all terms, we have

$$\lambda\sum_{k=\tau_t}^{t} E\|\nabla L(\theta_k,\gamma(k))\|^2 \leq \mathcal{O}((t+1)^{1-\beta+\sigma}) + \mathcal{O}((\tau_t+1)^2(1+t)^{1-\omega})$$
$$+ DU_G\sqrt{\sum_{k=\tau_t}^{t} E\|\nabla L(\theta_k,\gamma(k))\|^2}\sqrt{\sum_{k=\tau_t}^{t}(2E[A_k^2]+8E[B_k^2])}$$
$$+ 2DU_LU_G\epsilon_{app}(1+t-\tau_t).$$

Dividing now both sides by $(1+t-\tau_t)$ and assuming $t \geq 2\tau_t - 1$, we obtain

$$\lambda\sum_{k=\tau_t}^{t} E\|\nabla L(\theta_k,\gamma(k))\|^2/(1+t-\tau_t) \leq \mathcal{O}(1+t)^{\sigma-\beta} + \mathcal{O}((\tau_t+1)^2(1+t)^{-\omega}) + 2DU_LU_G\epsilon_{app}$$
$$+ DU_G\sqrt{\frac{1}{1+t-\tau_t}\sum_{k=\tau_t}^{t} E\|\nabla L(\theta_k,\gamma(k))\|^2}\sqrt{\frac{\sum_{k=\tau_t}^{t}(2E[A_k^2]+8E[B_k^2])}{1+t-\tau_t}}.$$

After applying the earlier square technique, we obtain

$$\sum_{k=\tau_t}^{t} E\|\nabla L(\theta_k,\gamma(k))\|^2/(1+t-\tau_t) = \mathcal{O}(t^{\sigma-\beta}) + \mathcal{O}(\log^2 t\cdot t^{-\omega}) + \mathcal{O}(\epsilon_{app}) + \mathcal{O}(\epsilon(t)),$$

where $\epsilon(t) = \sum_{k=\tau_t}^{t}(2E[A_k^2]+8E[B_k^2])/(1+t-\tau_t)$.

## B.6 PROOF OF THEOREM 4: ESTIMATING THE AVERAGE REWARD FOR CONSTRAINED NATURAL ACTOR CRITIC

We will use the same notations as used in Section B.2.

*Proof of Theorem 4*

From the algorithm we have the update rule as

$$L_{t+1} = L_t + a(t)(q(t) + \sum_{k=1}^{k=N}\gamma_k(t)(h_k(t) - \alpha_k)) - L_t).$$

Unrolling the above recursion, we have

$$
\begin{aligned}
y_{t+1}^2 &= (L_{t+1} - L_{t+1}^*)^2 \\
&= \left( L_t + a(t)\left( q(t) + \sum_{k=1}^{k=N} \gamma_k(t)(h_k(t) - \alpha_k) - L_t \right) - L_{t+1}^* \right)^2 \\
&= \left( y_t + L_t^* - L_{t+1}^* + a(t)\left( q(t) + \sum_{k=1}^{k=N} \gamma_k(t)(h_k(t) - \alpha_k) - L_t \right) \right)^2 \\
&= y_t^2 + 2a(t)y_t(C_t - L_t) + 2y_t(L_t^* - L_{t+1}^*) + (L_t^* - L_{t+1}^* + a(t)(C_t - L_t))^2 \\
&\leq y_t^2 + 2a(t)y_t(C_t - L_t) + 2y_t(L_t^* - L_{t+1}^*) + 2(L_t^* - L_{t+1}^*)^2 + 2a(t)^2(C_t - L_t)^2 \\
&= y_t^2 - 2a(t)y_t^2 + 2a(t)y_t^2 + 2a(t)y_t(C_t - L_t) + 2y_t(L_t^* - L_{t+1}^*) + 2(L_t^* - L_{t+1}^*)^2 + 2a(t)^2(C_t - L_t)^2 \\
&= y_t^2 - 2a(t)y_t^2 + 2a(t)y_t(C_t - L_t + y_t) + 2y_t(L_t^* - L_{t+1}^*) + 2(L_t^* - L_{t+1}^*)^2 + 2a(t)^2(C_t - L_t)^2 \\
&= (1 - 2a(t))y_t^2 + 2a(t)y_t(C_t - L_t^*) + 2y_t(L_t^* - L_{t+1}^*) + 2(L_t^* - L_{t+1}^*)^2 + 2a(t)^2(C_t - L_t)^2,
\end{aligned}
$$

where $C_t = q(t) + \sum_{k=1}^{k=N} \gamma_k(t)(h_k(t) - \alpha_k)$. The first inequality is due to $(x + y)^2 \leq 2x^2 + 2y^2$.

Rearranging and summing from $\tau_t$ to $t$, we have

$$
\sum_{k=\tau_t}^{t} E[y_k^2] \leq \underbrace{\sum_{k=\tau_t}^{t} \frac{1}{2a(k)} E(y_k^2 - y_{k+1}^2)}_{I_1} + \underbrace{\sum_{k=\tau_t}^{t} E[\hat{\Xi}(O_k, L_k, \theta_k, \gamma(k), q(k), h(k))]}_{I_2}
$$

$$
+ \underbrace{\sum_{k=\tau_t}^{t} \frac{1}{a(k)} E[y_k(L_k^* - L_{k+1}^*)]}_{I_3} + \underbrace{\sum_{k=\tau_t}^{t} \frac{1}{a(k)} E[(L_k^* - L_{k+1}^*)^2]}_{I_4}
$$

$$
+ \underbrace{\sum_{k=\tau_t}^{t} a(k) E[(C_k - L_k)^2]}_{I_5}.
$$

After carrying out an analysis similar to that of section B.2, we obtain

$$
\sum_{k=\tau_t}^{t} E[y_k^2]/(1 + t - \tau_t) = \mathcal{O}(t^{\omega - 1}) + \mathcal{O}(\log^2 t \cdot t^{-\omega}) + \mathcal{O}(t^{-2(\sigma - \omega)}).
$$

## B.7 PROOF OF THEOREM 4: ESTIMATING THE CONVERGENCE POINT OF CRITIC FOR CONSTRAINED NATURAL ACTOR CRITIC

The update rule for the critic in Algorithm 2 is similar to the one in Algorithm 1. Hence we will get the inequality (19) for natural constrained actor critic also. After carrying out an analysis similar to Section B.3 we get

$$
\left( \sum_{k=\tau_t}^{t} E\|v_k - v^*(k)\|^2 \right)/(1 + t - \tau_t) = \mathcal{O}\left( \frac{1}{t^{1-\omega}} \right) + \mathcal{O}\left( \frac{\log t}{t^{\omega}} \right) + \mathcal{O}\left( \frac{1}{t^{2(\sigma - \omega)}} \right).
$$

## B.8 PROOF OF COROLLARY 2

We have the following result from Theorem 3:

$$
\min_{0 \leq k \leq t} E[\|\nabla L(\theta_k, \gamma(k))\|^2] = \mathcal{O}(t^{\sigma - \beta})) + \mathcal{O}((\log t)^2 t^{-\omega}) + \mathcal{O}(\epsilon_{app}) + \mathcal{O}(\varepsilon(t)), \tag{21}
$$

where,

$$\varepsilon(t) = (2 \sum_{k=\tau_t}^{t} E\|A_k\|^2 + 8 \sum_{k=\tau_t}^{t} E\|B_k\|^2)/(1 + t - \tau_t),$$
$$A_k = L_k - L(\theta_k, \gamma(k)),$$
$$B_k = v_k - v(\theta_k, \gamma(k)).$$

From the results of Theorem 4, we have

$$\varepsilon(t) = \mathcal{O}(t^{\omega-1}) + \mathcal{O}(\log t \cdot t^{-\omega}) + \mathcal{O}(t^{-2(\sigma-\omega)}).$$

Putting this back in (21), we obtain

$$\min_{0 \leq k \leq t} E[\|\nabla L(\theta_k, \gamma(k))\|^2] = \mathcal{O}(t^{\sigma-\beta}) + \mathcal{O}(\log^2 t \cdot t^{-\omega}) + \mathcal{O}(t^{\omega-1}) + \mathcal{O}(\log t \cdot t^{-\omega}) + \mathcal{O}(t^{-2(\sigma-\omega)}) + \mathcal{O}(\epsilon_{app})$$

$$= \mathcal{O}(t^{\sigma-\beta}) + \mathcal{O}(\log^2 t \cdot t^{-\omega}) + \mathcal{O}(t^{\omega-1}) + \mathcal{O}(t^{-2(\sigma-\omega)}) + \mathcal{O}(\epsilon_{app})$$

$$= \mathcal{O}(t^{\sigma-\beta}) + \mathcal{O}(\log^2 t \cdot t^{-\omega}) + \mathcal{O}(t^{-2(\sigma-\omega)}) + \mathcal{O}(\epsilon_{app})$$

The last equality above again holds because $(\sigma - \beta) > (\omega - 1)$. Optimising over the choice of $\omega$, $\sigma$ and $\beta$, we have $\omega = 0.4$, $\sigma = 0.6$ and $\beta = 1$, respectively. Hence,

$$\min_{0 \leq k \leq t} E[\|\nabla L(\theta_k, \gamma(k))\|^2] = \mathcal{O}(\log^2 t \cdot t^{-0.4}) + \mathcal{O}(\epsilon_{app}).$$

Therefore, in order to obtain an $\epsilon$-approximate (ignoring the approximation error as with Wu et al., [2020]) stationary point of the performance function ($L(\theta, \gamma)$), namely,

$$\min_{0 \leq k \leq T} E[\|\nabla L(\theta_k, \gamma(k))\|^2] = \mathcal{O}(\log^2 T \cdot T^{-0.4}) + \mathcal{O}(\epsilon_{app}) \leq \mathcal{O}(\epsilon_{app}) + \epsilon,$$

we need to set $T = \tilde{\mathcal{O}}(\epsilon^{-2.5})$.

## C  EXPERIMENTAL SETTING

For detailed information about the settings involved for the three Safety-Gym environments, Safety-PointGoal1-v0 (SPG1-v0), Safety-CarGoal1-v0 (SCG1-vo) and SafetyPointPush1-v0 (SPP1-v0), respectively, please see Safety Gymnasium. We experimentally compare C-AC and C-NAC algorithms with C-DQN on the three settings.

The C-DQN algorithm is obtained from the algorithm Deep Q-Network (DQN) Mnih et al., [2015] by (a) modifying the basic setting to incorporate the average reward framework from the discounted reward setting considered there and (b) relaxing the constraints to form a Lagrangian in a similar manner as the C-AC and C-NAC algorithms. We also update the Lagrange parameter using the same updates as C-AC and C-NAC, respectively. This ensures a fair comparison across all the algorithms. Note, however, that such an update of the Lagrange parameter had not been used previously in the context of the DQN algorithm. We have taken the threshold level to be 0.1 for all the settings.

We observe that the constraint threshold is satisfied by all the three algorithms in the three different environments. Table 3 exhibits the values of the constraint costs (both average and standard error) over ten independent runs of each algorithm. As can be seen from the table as well as the bottom row of plots in Figure 1, the constraint threshold is met by all the three algorithms on each of the settings.

Table 3: Comparision of C-AC , C-NAC and C-DQN in terms of constraint cost $\pm$ standard error.

| Algorithm | SafetyPointGoal1-v0 | SafetyCarGoal1-v0 | SafetyPointPush1-v0 |
|-----------|---------------------|-------------------|---------------------|
| C-AC | $0.038 \pm 0.026$ | $0.0195 \pm 0.027$ | $0.028 \pm 0.018$ |
| C-NAC | $0.049 \pm 0.045$ | $0.047 \pm 0.046$ | $0.0295 \pm 0.032$ |
| C-DQN | $0.039 \pm 0.023$ | $0.00872 \pm 0.0076$ | $0.035 \pm 0.022$ |