# OpenReview forum: "Finite-Time Analysis of Three-Timescale Constrained Actor-Critic and Constrained Natural Actor-Critic Algorithms."
_auai.org/UAI/2024/Conference — UAI 2024 poster_

### Official Review · Reviewer_dAfe · 2024-02-27

**Q2-1 Originality-Novelty:** 3
**Q2-2 Correctness-Technical Quality:** 3
**Q2-5 Clarity Of Writing:** 3

**Q10 Ethical Concerns:**

I do not have ethical concerns.

**Q1 Summary And Contributions:**

This paper considers actor critic and natural actor critic algorithms with function approximation for constrained Markov decision processes (C-MDP) involving inequality constraints, and carries out a non-asymptotic analysis for both of these algorithms in a non-i.i.d (Markovian) setting. The authors consider the long-run average cost criterion, where both the objective and the constraint functions are suitable policy-dependent long-run averages of certain prescribed cost functions. Specifically, the authors handle the inequality constraints using the Lagrange multiplier method. The authors show that the actor critic and natural actor critic algorithms are guaranteed to find a first-order stationary point of the Lagrange function. The authors also show the results of experiments on three different Safety-Gym environments.

**Q2-3 Extent To Which Claims Are Supported By Evidence:**

3: Good: the main claims are supported by convincing evidence (in the form of adequate experimental evaluation, proofs, (pseudo-)code, references, assumptions).

**Q2-4 Reproducibility:**

3: Good: key resources (e.g. proofs, code, data) are available and key details (e.g. proofs, experimental setup) are sufficiently well-described for competent researchers to confidently reproduce the main results.

**Q3 Main Strengths:**

1.	The studied problem, constrained MDPs with function approximation, is interesting to the RL literature, and finds applications in safety-critical scenarios such as autonomous driving and robotic surgery.
2.	The authors prove that the Lagrange multiplier-based actor critic and natural actor critic algorithms are guaranteed to find a first-order stationary point of the Lagrange function.
3.	The authors conduct experiments on three different Safety-Gym environments to validate their theoretical results.

**Q4 Main Weakness:**

1.	It seems that this paper only provides theoretical guarantees in terms of the first-order stationary point of the Lagrange function. Can the authors provide the theoretical guarantees in terms of the optimality of the taken policy and constraint violation separately, as considered in many prior constrained RL works?
2.	The baselines in experiments are limited. It seems that the authors only compare the Lagrange multiplier-based actor critic and natural actor critic algorithms to the Constrained DQN (C-DQN) algorithm. It would be better to compare more existing constrained RL algorithms.

**Q5 Detailed Comments To The Authors:**

Please see the weaknesses above.

**Q9 Complying With Reviewing Instructions:**

Yes

---

> ### Author Rebuttal · Authors · 2024-04-06
>
> Response to Main weakness 1: Yes, we show convergence to stationary points. However, it is shown in Bhatnagar and Lakshmanan (2012) that, in the case of the C-AC algorithm, upon convergence of the gradient search procedure, one indeed obtains an optimal policy that gives a local minimum of the objective function while satisfying the constraint, see Proposition 4.3 and Remarks 4.3-4.5 of that reference. This is mainly because the stationary points that are not local minima are unstable equilibria of the underlying ODE. The same is also true of the C-NAC algorithm.  We shall add a remark in the paper to that effect. As far as providing finite time bounds on the optimality gap as well as constraint violation is concerned, there are several papers which provide such bounds but all of them are in the look-up table setting and not function approximation.
> Defining optimality gap precisely in our setting is hard due to the presence of multiple local minima in the constraint set and so obtaining such bounds is not easy as we know from results in stochastic approximation theory that if the stochastic recursion enters a compact neighborhood of any of the local minima infinitely often, it will converge to it. The compact neighborhood of which of the minima the recursion enters will depend on the initial condition, noise etc. Thus, in a general function approximation framework like we have, the optimality gap metric cannot be precisely defined.
>
> Response to Main weakness 2: C-AC is an important RL algorithm and we presented the C-NAC algorithm with function approximation for the first time. Our main contribution has been to provide the first non-asymptotic analysis of three-timescale constrained AC and NAC algorithms with function approximation and we show this analysis for the average cost setting. On top of the detailed theoretical results, we have also shown the results of experiments on three different environments where we have compared the empirical performance of these constrained algorithms.For empirical performance comparisons, we did implement the  constrained DQN,
> using the same constraint structure on the DQN that we used for C-AC and C-NAC algorithms so as to make accurate performance comparisons.

---

### Official Review · Reviewer_fLUh · 2024-03-24

**Q2-1 Originality-Novelty:** 3
**Q2-2 Correctness-Technical Quality:** 4
**Q2-5 Clarity Of Writing:** 4

**Q1 Summary And Contributions:**

The authors consider Constrained Markov Decision Processes with long-run average cost and side cost constraints
and propose for this setting the finite-time sample complexity analysis of Actor-Critic and Natural Actor-Critic reinforcement learning algorithms. These algorithms have already been proposed in BL2012 but the finite-time sample complexity analysis
is far from been trivial.

**Q2-3 Extent To Which Claims Are Supported By Evidence:**

4: Excellent: all claims are supported by very convincing evidence (in the form of comprehensive experimental evaluation, rigorous mathematical proofs, detailed (pseudo-)code, precise references, well-motivated and realistic assumptions) and the authors deliver what they promise.

**Q2-4 Reproducibility:**

3: Good: key resources (e.g. proofs, code, data) are available and key details (e.g. proofs, experimental setup) are sufficiently well-described for competent researchers to confidently reproduce the main results.

**Q3 Main Strengths:**

1. It is important to have a finite-time sample complexity analysis for AC RL algorithms for
Constrained MDPs.

2. In addition to the theoretical analysis, there is also a numerical comparison with a variation
of DQN.

**Q4 Main Weakness:**

The assumptions could have been explained and justified more rigorously. (See more below.)

**Q5 Detailed Comments To The Authors:**

The assumptions could have been explained and justified more rigorously.
In particular, Assumption 5 cannot be checked in practice. The authors mention that
Assumption 4(a) implies Assumption 5. Why not to state this as a formal lemma
or there is still some degree of hand-waving?

Related questions: what \pi_\theta the authors suggest to use? What have they used
in the numerical experiments?

**Q9 Complying With Reviewing Instructions:**

Yes

---

> ### Author Rebuttal · Authors · 2024-04-06
>
> Response to comment: We agree with the reviewer and will explain the assumptions more rigorously in the paper. We listed Assumption 5 separately since we needed to make use of both Assumptions 4 and 5 in the proof. We are thankful for the reviewer's comments and will formally provide a small result proving that Assumption 5 follows from Assumption 4(a) along the lines already described in the material following Assumption 5.
>
>
>
> Response to question: For our experiments, we have parameterized the policy using neural networks as OpenAI Gym environments largely require such architectures for good performance. Our experiments with these archnitectures also showed good performance of the algorithms. In practice, one can also potentially use parameterized Softmax policies as well.

---

### Official Review · Reviewer_9xyu · 2024-03-24

**Q2-1 Originality-Novelty:** 2
**Q2-2 Correctness-Technical Quality:** 2
**Q2-5 Clarity Of Writing:** 2

**Q1 Summary And Contributions:**

The paper studies constrained reinforcement learning and proposes an actor-critic algorithm and a natural actor-critic algorithm, both of which are equipped with finite-time bounds. Numerical simulations are also provided to demonstrate the performance of the proposed algorithms.

**Q2-3 Extent To Which Claims Are Supported By Evidence:**

3: Good: the main claims are supported by convincing evidence (in the form of adequate experimental evaluation, proofs, (pseudo-)code, references, assumptions).

**Q2-4 Reproducibility:**

3: Good: key resources (e.g. proofs, code, data) are available and key details (e.g. proofs, experimental setup) are sufficiently well-described for competent researchers to confidently reproduce the main results.

**Q3 Main Strengths:**

Making reinforcement learning algorithms safe and robust is crucial for the practical potential of reinforcement learning. This paper tackles this important problem and provides algorithms with theoretical guarantees, which could be of broad interest to the reinforcement learning community.

**Q4 Main Weakness:**

(1) In the unconstrained setting, both policy gradient and natural policy gradient have global convergence guarantees. In the constraint setting, this paper only establishes convergence to stationary points, which does not imply any performance guarantees of the output policy.

(2) In the unconstrained setting, NAC has better sample complexity guarantees compared with AC due to its connection to policy iteration. In the unconstrained setting, this does not seem to be the case. Why is that?

(3) Assumption 5 imposes an assumption on the iterates generated by the algorithm, which is not ideal. It would be better if this assumption could be removed or replaced by an assumption that does not depend on the algorithm trajectory.

(3) Does the finiteness of $\epsilon_{app}$ automatically hold? If not, it should be stated in an assumption environment.

**Q5 Detailed Comments To The Authors:**

See Q4.

**Q9 Complying With Reviewing Instructions:**

Yes

---

> ### Author Rebuttal · Authors · 2024-04-06
>
> Response to Main weakness 1: We showed convergence to stationary points because we do not make strong assumptions regarding the function approximators, the constraint set etc. Thus, our objective in general will have multiple local minima within the feasible region. Establishing global convergence will require us to make additional (many times unrealistic) assumptions. Nonetheless, it is already shown in Bhatnagar and Lakshmnanan (2012)(see proposition 4.3 and remarks 4.3-4.5) that the algorithm C-AC converges to the set of local minima since the other stationary points will be unstable equilibria of the associated ODE. Some related papers do show finite time bounds on the optimality gap and the constraint violation but this is largely accomplished in the look-up table case (not the function approximation setting as we consider). In our case, defining optimality gap precisely will be a problem because of the presence of multiple local minima in the constraint set.We will add a remark concerning the local optimality of the obtained policy. Similar results as in Bhatnagar and Lakshmanan (2012) will hold for the local optimality of the converged policy in the case of natural AC algorithm as well.
>
>
> Response to Main weakness 2: No, this is not a well-established fact that NAC gives better sample complexity than AC. For instance, Table 1 in
> "Khodadadian, Sajad, Zaiwei Chen, and Siva Theja Maguluri, Finite-sample analysis of off-policy natural actor-critic algorithm,  International Conference on Machine Learning. PMLR, 2021", shows that the sample complexity of NAC algorithm is not better than AC in the (unconstrained) function approximation setting. It is shown in "Wang, L., Cai, Q., Yang, Z., \& Wang, Z. (2019), Neural policy gradient methods: Global optimality and rates of convergence,  arXiv preprint arXiv:1909.01150", that AC gives a sample complexity of $\mathcal{\tilde{O}}(\epsilon^{-6})$ while NAC gives a sample complexity of $\mathcal{\tilde{O}}(\epsilon^{-14})$ (which is clearly worse than what AC achieves). Our constrained AC and NAC algorithms on the other hand, have the same computational complexity of $\mathcal{\tilde{O}}(\epsilon^{-2.5})$.
>
>
>
> Response to Main weakness 3:  Assumption 5 essentially assumes stability of the $G_t$ recursion as well as its inverse. As mentioned in our response to Reviewer bsBb, and as indicated in the lines following Assumption 5, this assumption can be ensured to hold from Assumption 4(a) and the fact that the step-size sequence is diminishing. In place of Assumption 5, we shall provide in the paper a small result, as hinted in the aforementioned lines following Assumption 5, where we shall prove that Assumption 5 holds.
>
>
>
> Response to Main weakness 4:  In the paper we have assumed that  $\forall \theta ,\forall \gamma,  \mbox{ } \epsilon_{\text{app}}(\theta ,\gamma) \le \epsilon_{\text{app}}$, where  $\epsilon_{\text{app}}\geq0$ is some constant. We shall state it as an assumption.

---

### Official Review · Reviewer_bsBb · 2024-03-26

**Q2-1 Originality-Novelty:** 3
**Q2-2 Correctness-Technical Quality:** 3
**Q2-5 Clarity Of Writing:** 3

**Q1 Summary And Contributions:**

The paper presents the first non-asymptotic analysis of the constrained actor-critic and constrained natural actor-critic algorithms. Specifically, they show that the proposed algorithms can obtain an $\epsilon$-optimality of the Lagrange function with $\tilde{\mathcal{O}}(\epsilon^{-2.5})$ sample complexity.

**Q2-3 Extent To Which Claims Are Supported By Evidence:**

3: Good: the main claims are supported by convincing evidence (in the form of adequate experimental evaluation, proofs, (pseudo-)code, references, assumptions).

**Q2-4 Reproducibility:**

3: Good: key resources (e.g. proofs, code, data) are available and key details (e.g. proofs, experimental setup) are sufficiently well-described for competent researchers to confidently reproduce the main results.

**Q3 Main Strengths:**

1. First non-asymptotic analysis of a three time scale constrained actor critic algorithm.
2. The obtained sample complexity is same as its unconstrained counterpart.

**Q4 Main Weakness:**

1. The main problem is that the authors present first-order convergence results of the Lagrange function. Ideally, convergence results for the objective function (regret) and constraint value function (constraint violation) should be separately presented. However, the convergence of the Lagrange function is neither helpful nor interpretable for real-world applications.

2. Secondly, although the analysis and results are new for constrained actor-critic algorithms, other CMDP algorithms in the literature may obtain better sample complexities with better convergence metrics. No discussion is provided in this direction.

**Q5 Detailed Comments To The Authors:**

$\textbf{Comments:}$

1. The example of safe driving may not be appropriate for motivating CMDP with long-term cost since accident prevention must restrain the cost at each time instant.
2. The authors should provide a table that compares the sample complexities of different CMDP algorithms.
3. Assumption 4(b) is redundant since it can be derived from Assumption 4(a) using first-order Taylor expansion.

$\textbf{Questions:}$

1. Are there any non-actor-critic algorithm in the literature that has better sample complexity than the algorithms presented in this paper? If yes, why should we consider the presented result as an improvement of the state-of-the-art? If no, then why not present the result as an improvement of the state-of-the-art CMDP algorithms?
2. What is the intuition behind Assumption 2? In which situations, is this assumption expected to not hold?
3. If Assumption 5 is implied by Assumption 4(a) (stated in the main text, page 5), why enlist it as a separate Assumption?
4. Why is the linear approximation error defined in terms of the limiting point $v^*(\boldsymbol{\theta}, \gamma)$? I think it makes more sense if the error is defined as, $$\epsilon_{\mathrm{app}}(\boldsymbol{\theta}, \gamma) \triangleq \sqrt{\min_{v}\mathbb{E}\_{s\sim\mu_{\theta}}\left(f_s^{\mathrm{T}}v-V^{\pi_{\boldsymbol{\theta}}, \gamma}(s)\right)^2}$$
The above definition follows a similar definition of the approximation error $\epsilon_{\mathrm{bias}}$ used for policy gradient algorithms with general parameterization. For example, see Assumption 2 in [1]. A comparative discussion would be helpful. Please also clarify if the above definition of $\epsilon_{\mathrm{bias}}(\boldsymbol{\theta}, \gamma)$ is equivalent to the definition given in the paper (i.e., if the limiting point $v^*(\boldsymbol{\theta}, \gamma)$ is a minimizer of the linear approximation error).
5. Can the first-order convergence result be converted to a global one using similar techniques in [1] and other related papers? Specifically, see Lemma 3 in [1] which establishes a connection between the first order and global convergence.

[1] Mondal, W.U. and Aggarwal, V., 2023. Improved sample complexity analysis of natural policy gradient algorithm with general parameterization for infinite horizon discounted reward Markov decision processes. arXiv preprint arXiv:2310.11677.

**Q9 Complying With Reviewing Instructions:**

Yes

---

> ### Author Rebuttal · Authors · 2024-04-06
>
> Weakness 1: It is shown in Bhatnagar \& Lakshmanan (2012) that for C-AC, stationary-point convergence results in locally optimal policy, that gives local minimum of the objective in the constraint set, see Proposition 4.3 and Remarks 4.3-4.5 there. This is because stationary points that are not local minima are unstable attractors of the underlying ODE. The same is also true of the C-NAC algorithm. We shall add a remark in the paper to that effect. As for providing finite time bounds on the optimality gap as well as constraint violation, there are papers which provide such bounds but all of them are in the look-up table setting and not function approximation.
> Defining optimality gap precisely in our setting is hard due to the presence of multiple local minima in the constraint set and so obtaining such bounds is not easy. From results in stochastic approximation, if the stochastic recursion enters a compact neighborhood of a local minimum infinitely often, it will converge to it w.p.1. The compact neighborhood of which of the minima the recursion enters will depend on the initial condition and noise. Thus, in a general function approximation framework, the optimality gap cannot be precisely defined.
>
> Weakness 2: Our is the first work to give sample complexity of any three-timescale RL algorithm. While there are several papers which perform finite time analyses for various constrained RL algorithms when the algorithm uses a look-up table representation, such as M.Agarwal, Q.Bai and V.Aggarwal, TMLR, 12/2022 (and the references there), or where the critic uses look-up table representation even though the actor is parameterized, such as D.Ding, K.Zhang, T.Basar and M.R.Jovanovic, NeurIPS, 2020,  we did not find any paper on constrained RL where both actor and critic are parameterized like we consider. Hence, we did not show sample complexity comparisons with any constrained RL algorithm in the FA case. We agree however and will add a table with sample complexities of constrained algorithms (even though they are not for FA setting).
>
> Comment 1: We meant safe navigation of a multitude of self-driving vehicles in the long-run where the average cost constraint would be meaningful (and not the safe navigation of a particular vehicle from its source to the destination). \\
>
> Comment 2: Addressed in Weakness 2.
>
> Comment 3: Assumption 4 has been borrowed from Wu et al (2020) and other papers. We agree with the reviewer that Lipschitz continuity of $\pi_\theta(a|i)$ in 4(b) will follow from 4(a). We will remove 4(b) and instead just state this fact. We thank the reviewer for pointing this out.
>
> Question 1: Addressed in Weakness 2.
>
> Question 2: Assumption 2 is a technical requirement made to guarantee the problem’s solvability because of the long-run average cost framework, as a result of which the Bellman operator is a non-expansive map and not a contraction,  unlike discounted cost MDPs. This is akin to the relative value iteration procedure that one adopts to get a unique solution. This assumption has been used while estimating the convergence point of the critic (see the last inequality on page no. 32). This assumption has also been made in Wu et al. A slightly different technical requirement but which is also geared towards showing that the matrix A is invertible has been made in Tsitsiklis and Van Roy. Automatica, 35(11):1799-1808, 1999, which deals with the case of TD-based value function estimation for a given policy.
>
> Question 3: We agree and will formally provide a small result proving that Assumption 5 follows from 4(a) along the lines described in the material following Assumption 5.
>
> Question 4: $v^{*}(\theta,\gamma)$ is the limit point for TD(0) under the policy $\pi_{\theta}$ and $V^{\pi_{\theta},\gamma}(\cdot)$ is the state value function under policy $\pi_{\theta}$. Since $\epsilon_{\text{app}}(\theta,\gamma)$ is the approximation error of the linear function class, it is more reasonable to look at the error post critic-convergence. In [1], advantage function of the state-action tuples with compatible function approximation is used whereas we consider the value function of the state alone under general FA.
>
> Question 5: Lemma 3 of [1] gives global convergence but under conditions such as strong convexity of the compatible function approximation error obtained by assuming  existence of a uniform positive lower bound on the eigenvalues of the Fisher information matrix across all policy parameters. We do not make such assumptions and hence our algorithm is guaranteed to output only a locally optimal policy obtained upon ensuring that the cost constraints are met. Further, [1] considers the unconstrained setting (with compatible function approximators) unlike us as we work with the constrained setting and with general function approximators. Extending our work to the setting of [1] can be a good future direction for the constrained setting with function approximation. We will cite [1] in the paper.

---

### Meta-Review · Area_Chair_Q8fa · 2024-04-20

This submission addresses a significant area in reinforcement learning, offering a finite-time analysis of three-timescale average cost actor-critic algorithms with constraints. It introduces the first such analysis for constrained actor-critic and natural actor-critic algorithms, marking a noteworthy step in theoretical advancements for reinforcement learning in constrained environments.

Considering the mixed reviews, the paper presents a solid theoretical contribution that pushes forward the understanding of constrained reinforcement learning algorithms. The authors have addressed several concerns raised during the review process adequately, suggesting that they are committed to improving the paper's clarity and depth in a revision.